



# Description and evaluation of the tropospheric aerosol scheme in the Integrated Forecasting System (IFS-AER, cycle 45R1) of ECMWF

Samuel Rémy[1], Zak Kipling[2], Johannes Flemming[2], Olivier Boucher[1], Pierre Nabat[4], Martine Michou[4], Alessio Bozzo[2,5], Melanie Ades[2], Vincent Huijnen[3], Angela Benedetti[2], Richard Engelen[2], Vincent-Henri Peuch[2], and Jean-Jacques Morcrette[2]

[1]Institut Pierre-Simon Laplace, Sorbonne Université / CNRS, Paris, France
[2]European Centre for Medium Range Weather Forecasts, Reading, UK
[3]Royal Netherlands Meteorological Institute, De Bilt, Netherlands
[4]Météo-France, Toulouse, France
[5]now at European Organisation for the Exploitation of Meteorological Satellites, Darmstadt, DE

*Correspondence to:* Samuel Rémy (samuel.remy@lmd.jussieu.fr)

**Abstract.**

This article describes the IFS-AER aerosol module used operationally in the Integrated Forecasting System (IFS) cycle 45R1, operated by the European Centre for Medium Range Weather Forecasts (ECMWF) in the framework of the Copernicus Atmospheric Monitoring Services (CAMS). We describe the different parameterizations for aerosol sources, sinks and its
5   chemical production in IFS-AER, as well as how the aerosols are integrated in the larger atmospheric composition forecasting system. The focus is on the entire 45R1 code-base, including some components that are not used operationally, in which case this will be clearly specified. This paper is an update to the Morcrette et al. (2009) article that described aerosol forecasts at ECMWF, using the cycle 32R2 of the IFS. Between cycles 32R2 and 45R1, a number of source and sink processes have been reviewed and/or added, increasing notably the complexity of IFS-AER. A greater integration with the tropospheric chemistry
10  scheme of the IFS has been achieved, for the sulphur cycle as well as for nitrate production. Two new species, nitrate and ammonium, have also been included in the forecasting system. Global budgets and aerosol optical depth (AOD) fields are shown, as well as an evaluation of the simulated Particulate Matter (PM) and AOD against observations, showing an increase in skill from cycle 40R2, used in the CAMS interim Reanalysis (CAMSiRA), to cycle 45R1.

## 1   Introduction

Ambient air pollution is a major public health issue, with effects ranging from increased hospital admissions to increased risk of premature death. Globally, an estimated 4.2 million deaths are estimated to be linked to air pollution in 2016 (World Health Organisation, report on Ambient air quality and health, 2018, https://www.who.int/news-room/fact-sheets/detail/ambient-(outdoor) -air-quality-and-health, accessed on 02/04/2019), mainly from heart disease, stroke, chronic obstructive pulmonary disease,





lung cancer, and acute respiratory infections in children. A large part of this mortality is caused by exposure to small particulate matter of 2.5 microns or less in diameter ($PM_{2.5}$), which are known to cause cardiovascular and respiratory disease, as well as cancers. Particulate matter, or atmospheric aerosols, also acts on the climate system though these effects are still subject to large uncertainties. Particles released by volcanic eruptions can also impact air traffic, as happened in April 2010 with the

eruption of the Eyjafjallajökull volcano in Iceland, which led to a major disruption of European and transatlantic aviation. As a consequence, modelling and forecasting levels of particulate matter, with the highest possible level of accuracy, is a major concern of the public authorities worldwide and has focused a large effort of the research community. In this context, ECMWF has been one of the first centres to propose operational global forecasts of aerosols. Besides ECMWF, there are currently at least eight centers producing and disseminating near real-time operational global aerosol forecasting products: Japan Meteorolog-

ical Agency (JMA), NOAA National Centre for Environmental Prediction(NCEP), US Navy's Fleet Numerical Meteorology and Oceanography Centre (NREL/FNMOC), NASA Global Modelling and Assimilation Office (GMAO), UK Met Office, Météo-France, Barcelona Supercomputing Center (BSC) and the Finnish Meteorological Institute (FMI). These groups are all member of the International Cooperative for Aerosol Prediction (ICAP; Sessions et al., 2015; Xian et al., 2019), which uses data provided by these centers in the ICAP Multi Model Ensemble (ICAP-MME).

Global monitoring and forecasting of aerosols is a key objective of the Copernicus Atmospheric Monitoring Service (CAMS), operated by ECMWF on behalf of the European Commission. To achieve this, ECMWF operates and develops the Integrated Forecasting System (IFS), which combines state of the art meteorological and atmospheric composition modelling together with the data assimilation of satellite products in the framework of CAMS (2014 to present) and before that of the Monitoring Atmospheric Composition and Climate series of projects (MACC, MACC-II and III, 2010 to 2014) and of the Global and

regional Earth-system Monitoring using Satellite and in-situ data project (GEMS, 2005 to 2009; Hollingsworth et al., 2008). The MACC and CAMS projects are centered around operational Near-Real-Time (NRT) forecasts and reanalyses of global atmospheric composition: the MACC reanalysis (Inness et al., 2013), the CAMS interim ReAnalysis (CAMSiRA; Flemming et al., 2017), and the CAMS reanalysis (CAMSRA; Inness et al., 2019). The IFS is originally a numerical weather prediction system dedicated to operational meteorological forecasts. It was extended to forecast and assimilate aerosols (Morcrette

et al., 2009; Benedetti et al., 2009), greenhouse gases (Engelen et al., 2009; Agustí-Panareda et al., 2014) and reactive trace gases (Flemming et al., 2009, 2015; Huijnen et al., 2019). The atmospheric composition component of the IFS was continually updated within the MACC and CAMS project, with yearly or twice yearly upgrades of the operational forecasting system. Besides its use in the CAMS project, different versions of IFS-AER have been adapted within Météo-France's CNRM-CM system (Michou et al., 2015); it is also part of the MarcoPolo–Panda ensemble dedicated to the forecast of air quality in Eastern China

(Brasseur et al., 2019), of which ECMWF is a member.

Morcrette et al. (2009), hereafter denoted as M09, and Benedetti et al. (2009) describe the aerosol modelling and data assimilation aspects, respectively, in the cycle 32R2 of the IFS. This paper focuses on the updates in the forward model since 2009; the data assimilation aspects and the optical properties used are only briefly described.

The paper is organized as follows. Section 2 presents a general description of IFS-AER and how it is implemented in the

IFS and interacts with other components of the forecasting system. Section 3 focuses on the dynamical and prescribed aerosol





emissions and production processes. Section 4 details the aerosol sink processes: dry and wet deposition and sedimentation. The aerosol optical properties and PM formulae are presented in Section 5, while Section 6 describes the model configuration used in the operational Near-Real-Time (NRT) simulations. Section 7 presents simulation results and budgets; Section 8 is dedicated to a preliminary evaluation of simulations against Aerosol Optical Depth (AOD) observations from the AERONET

network (Holben et al., 1998) and against European and North-American PM observations.

## 2 General description of IFS and IFS-AER

### 2.1 Atmospheric composition forecasts with the Integrated Forecasting System (IFS)

General aspects of the IFS and how they relate to atmospheric composition modelling are described in Flemming et al. (2015); a more detailed technical and scientific documentation of the cycle 45R1 release of the IFS can be found at https:

//www.ecmwf.int/en/forecasts/documentation-and-support/evolution-ifs/cycles/summary-cycle-45r1 (last accessed on 9th of May 2019). The IFS is a Numerical Weather Prediction (NWP) model operated by the ECMWF to provide operational weather forecasts with extensions to represent tropospheric aerosols, chemically-interactive gases and greenhouse gases. This integrated atmospheric composition forecasting system forms the core of the global system of the Copernicus Atmosphere Monitoring Service (CAMS). At the start of the time step, the three-dimensional advection of the tracer mass mixing ratios is simulated

using a semi-Lagrangian (SL) method as described in Temperton et al. (2001) and Hortal (2002). Mass conservation of the transported tracers (aerosols and trace gases) can be an issue because the SL scheme is not formally mass conservative. Similarly to what is practiced for trace gases (Flemming et al., 2015; Diamantakis and Flemming, 2014) and for greenhouse gases (Agusti-Panareda et al., 2017), a proportional mass fixer is used in order to ensure that the total global mass of aerosol tracers is conserved during advection.

The aerosol tracers are mixed vertically by the turbulent diffusion scheme (Beljaars and Viterbo, 1998), which also simulates the injection of emissions at the surface and the application of the surface dry deposition flux as boundary conditions. The dry deposition velocity is estimated by IFS-AER depending on the land surface and meteorological conditions as outlined in section 4. The aerosol tracers are further transported and mixed vertically by the shallow and deep convection fluxes (Bechtold et al., 2014).

Since cycle 43R3, a new radiation package is in use operationally in the IFS and is described in Hogan and Bozzo (2018). The shortwave and longwave aerosol-radiation interactions (ARI) can be computed using an aerosol climatology based on the CAMS interim Reanalysis (Bozzo et al., 2019). Optionally, the prognostic aerosol mass mixing ratio from IFS-AER can be used to compute dynamically the ARI; this option is used in the operational context since cycle 45R1. This was shown to have occasionally a large impact on surface temperature (e.g. Rémy et al., 2015) when the aerosol loading is particularly high. There

is currently no representation of aerosol-cloud interactions (ACI).

  The aerosol tracers and related processes are represented only in grid-point space. The horizontal grid can be either a reduced Gaussian grid (Hortal and Simmons, 1991) or a cubic octahedral grid. The vertical distribution uses a hybrid sigma-pressure coordinate with 60 or 137 levels. In this paper, a horizontal spectral resolution of $T_L511$ (equivalent to a grid box size of





about 40 km) and a vertical resolution of 60 levels were used, which matches the resolution used operationally with cycle 45R1. The aerosol tracers in IFS-AER can either be initialized using the 4D-Var data assimilation of the IFS as described in Benedetti et al. (2009), or by the 3D fields from the previous forecast (in so-called "cycling forecast mode"). In the latter case, the meteorological fields are provided by the ECMWF IFS operational analysis.

## 2.2 Atmospheric composition in the IFS

Tropospheric and stratospheric chemistry is represented in the IFS through the IFS-CB05-BASCOE system (Flemming et al., 2015; Huijnen et al., 2016). Tropospheric chemistry in the IFS is based on a modified version of Carbon Bond 05 (CB05; Yarwood et al., 2005), which represents 55 trace gases interacting through 93 gaseous, 3 heterogeneous and 18 photolysis reactions. IFS-CB05 is described in detail in Flemming et al. (2015). Stratospheric chemistry is based on the Belgian Assimilation System for Chemical ObsErvations (BASCOE; Errera et al., 2008), which was first developed to assimilate satellite observations of stratospheric composition. The BASCOE version as adapted in the IFS includes 58 trace gases interacting through 142 gaseous, 9 heterogeneous and 52 photolysis reactions. The merging of tropospheric and stratospheric chemistry parameterizations is described in detail in Huijnen et al. (2016). The representation of stratospheric chemistry through BASCOE is not used in the operational cycle 45R1. Alternative chemistry schemes, based on IFS-MOZART and IFS-MOCAGE, have also become available recently (Huijnen et al., 2019).

## 2.3 Main characteristics of IFS-AER

IFS-AER is a bulk-bin scheme derived from the LOA/LMDZ model (Boucher et al., 2002; Reddy et al., 2005), using mass mixing ratio as prognostic variable of the aerosol tracers. The prognostic species are sea-salt, desert dust, Organic Matter (OM), Black Carbon (BC), sulfate and its gas-phase precursor sulphur dioxide. IFS-AER can be run in standalone mode, i.e. without any interaction with the chemistry, or coupled with IFS-CB05. Sea-salt is represented with three bins (radius bin limits at 80% relative humidity are 0.03, 0.5, 5 and 20 microns). As described in Reddy et al. (2005), sea salt emissions as well as sea salt particle radii are expressed at 80% relative humidity. This is different from all the other aerosol species in IFS-AER, which are expressed as dry mixing ratios (0% relative humidity). Users should bring special attention to this when dealing with diagnosed sea-salt aerosol mass mixing ratio, which needs to be divided by a factor of 4.3 to convert to dry mass mixing ratio, in order to account for the hygroscopic growth and change in density. Desert dust is also represented with three bins (radius bin limits are 0.03, 0.55, 0.9, and 20 microns). For both dust and sea-salt, there is no mass transfer between bins. Organic matter and black carbon, two components, hydrophilic and hydrophobic are considered, with the ageing processes transferring mass from hydrophobic to the hydrophilic OM and BC. Sulphate aerosols and, when not fully coupled to IFS-CB05, its precursor gas sulfur dioxide are represented by two prognostic variables. When running fully coupled with IFS-CB05, which is not the operational configuration with cycle 45R1, sulfur dioxide is represented in CB05 and thus not in IFS-AER. For the optional nitrate species, two prognostic variables represent fine mode nitrate, produced by gas-particle partitioning, and coarse mode nitrate, produced by heterogeneous reactions of dust and sea-salt particles. In all, IFS-AER is thus composed of 12





prognostic variables when running standalone and 14 when fully coupled with IFS-CB05, which allows for a relatively limited consumption of computing resources, as shown in Table 1.

**Table 1.** System Billing Unit (SBU) consumption of a 24hour forecast at $T_L511L60$. SBU is a unit of CPU consumption used at ECMWF; its precise definition can be found at https://confluence.ecmwf.int/display/UDOC/HPC+accounting

| Configuration | SBU used |
|---|---|
| IFS (NWP) | 483 |
| IFS-AER (standalone) | 704 |
| IFS-AER-CB05 (coupled) | 1030 |

### 2.4 Coupling to the chemistry

IFS-AER can run coupled with the tropospheric chemistry scheme IFS-CB05. The coupling is two-way and consists, on the chemistry side, in the use of aerosols in heterogeneous chemical reactions and in the computation of the photolysis rates, which is operational since cycle 43R3. On the aerosol side, the coupling is not used operationally and consists in the use of the gaseous precursors $HNO_3$ and $NH_3$ from IFS-CB05 for the production of nitrate and ammonium aerosols through gas-partitioning and heterogeneous reactions on dust and sea-salt particles, as described in Section 4. The updated concentrations of the precursors gases are passed back to IFS-CB05. Production rates of sulphate aerosols as estimated by IFS-CB05 can also be used in IFS-AER (this option is also not used operationally).

### 3 Aerosol sources

In IFS-AER, the sea-salt and dust emissions are computed dynamically using prognostic variables from the meteorological model. Conversion of sulfur dioxide into sulfate aerosol as well as nitrate production also uses input from the meteorological model. The other aerosol species use external emissions datasets. Aerosol emissions are released at the surface, except for emissions from biomass burning which can optionally be released at an injection height, and $SO_2$ emissions from outgassing volcanoes which can optionally be released at the altitude of the volcano. In the operational 45R1 context, emissions from biomass burning are released at the surface while $SO_2$ emissions from outgassing volcanoes are released at the altitude of the volcano.

### 3.1 Organic Matter and Black Carbon

The anthropogenic (non biomass-burning) sources of OM and BC can be taken from the MACCity (Granier et al., 2011) or the more recent CMIP6 (Gidden et al., 2019) emissions datasets; for the operational cycle 45R1 analysis and forecasts emissions from MACCity are used. These emissions inventories provide monthly emissions, updated from year to year for MACCity.





MACCity emissions of black carbon are distributed by 20% into the hydrophilic and the remaining 80% into the hydrophobic black carbon tracers. MACCity emissions provide only organic carbon emissions rather than organic matter emissions. To translate these organic carbon emissions into OM emissions a OM:OC ratio of 1.8 is used. This is in the middle range of the OM:OC ratio provided by Canagaratna et al. (2015) and Philip et al. (2014). The OM emissions are then divided evenly

between hydrophilic and hydrophobic OM. Table 2 reports the average yearly global anthropogenic emissions for the year 2014 from the three inventories. Biomass-burning emissions from the Global Fire Assimilation System (GFAS) are also shown. The sulfur dioxide emissions are remarkably consistent between the three datasets. This is less the case for OM and BC.

**Table 2.** Global emissions in 2014 of Organic Matter, Black Carbon and Sulfur dioxide in $Tg\,yr^{-1}$. Anthropogenic (non biomass burning) sources from MACCity and CMIP6 and biomass burning sources from GFAS are shown.

|  | Anthropogenic | | Biomass-burning |
| --- | --- | --- | --- |
| Species | MACCity | CMIP6 | GFAS |
| Organic matter | 21.3 | 29.7 | 101.5 |
| Black carbon | 4.97 | 7.97 | 6.57 |
| Sulfur dioxide | 108.7 | 111.1 | 2.3 |

Biomass burning sources of OM and BC are provided by GFAS (Kaiser et al., 2012), which estimates these emissions (along with those of trace gases) using active fire product from the Moderate Resolution Imaging Spectroradiometer (MODIS)

instrument onboard the Aqua and Terra satellites. Kaiser et al. (2012) compared cycling forecast simulations of biomass burning aerosols with simulations using data assimilation and concluded that a scaling factor of 3.4 should be applied to GFAS biomass-burning sources when used in the IFS. The same method was used in Rémy et al. (2017) to derive distinct scaling factors for the OM and BC species; with scaling factors varying from 2.7 and 5 with an average of 3.2 for the former, and from 4.9 to 7 with a 6.1 average for the latter. The use of scaling factors for biomass burning emissions is frequent; for example

a value of 1.7 is used in the Met Office Unified Model limited area model configuration over South America that was used for the South American Biomass Burning Analysis (SAMBBA) campaign (Kolusu et al., 2015); values of 1.8 to 4.5 are used in GEOS-5 (Colarco, 2011). Some models such as CAM5 (Tosca et al., 2013) also use regional scaling factors (Lynch et al., 2016). The reasons why scaling factors are required are not fully elucidated but relate at least in part to the condensation of gases onto the particles as the aerosol plume ages.

Biomass burning emissions are by default released at the surface. This can be unrealistic: a large fraction of fires release smoke constituents in the planetary boundary layer (PBL) and a minority of very large fires emit large quantities of aerosols and trace gases in the free troposphere and even, for extreme cases, in the stratosphere (Freitas et al., 2005). The fraction of fires that emit aerosols and trace gases in the free troposphere was evaluated at 5–15% by various authors (Kahn et al., 2008; Val Martin et al., 2010; Sofiev et al., 2012). The GFAS dataset also includes daily injection heights that are computed using two

different methods: the IS4FIRE approach (Sofiev et al., 2013) and the Plume Rise Model (PRM; Freitas et al., 2010) approach.





Injection heights from GFAS as estimated using the PRM can optionally be used for biomass burning emissions. Biomass burning aerosols are emitted at the mean height of maximum injection, which is defined as the average of the plume heights at which detrainment is above half the maximum value. The daily injection heights in GFAS are representative of the maximum value reached during daytime (see Rémy et al., 2017), so using these at night when the atmosphere is stable could lead to errors
in the vertical distribution and transport of biomass burning aerosol plumes. To prevent this, injection heights are used only if the mean height of maximum injection is above 200 m and if the diagnosed PBL height is above 1500 m. Otherwise, the smoke constituents are released in the first three model levels above surface.

The treatment of Secondary Organic Aerosols (SOA) is very simplistic in IFS-AER. SOA are part of the organic matter species, and are emitted at the surface. The biogenic component of SOA emissions is taken from the EDGAR emissions
inventory (Dentener et al., 2006) which estimates SOA emissions as a 15% fraction of natural terpene emissions; biogenic SOA emissions stand at 19.1 Tg yr$^{-1}$. Additionally and optionally, since cycle 43R1, the anthropogenic component of SOA production is represented in a very simple way as a fraction of CO emissions from MACCity, following Spracklen et al. (2011). This option has been used operationally since cycle 43R1. Anthropogenic SOA emissions estimated using this method amount to 144 Tg yr$^{-1}$, which is consistent with the estimate provided by Spracklen et al. (2011) and by Hodzic et al. (2016), for which
best estimates of global SOA production stand at 132 Tg/Yr. A recent intercomparison (Tsigaridis et al., 2014) showed that most models underestimate the production of SOA. Most speciated observations indicate that SOA compose a large fraction of surface aerosols, and this very simple representation helps in addressing a persistent underestimation of anthropogenic aerosols in the IFS. This new source of anthropogenic aerosols also had adverse impacts on PM simulations, leading to a large overestimation especially over China (as noted in Brasseur et al., 2019). Work is ongoing to address this through establishing
a coupling with precursor organic chemistry; as a temporary solution in the operational cycle 45R1, anthropogenic SOA emissions have been capped at 0.25 µg m$^{-2}$ s$^{-1}$. Figure 1 shows that the SOA emissions estimated with this method exhibit a strong seasonality and are concentrated in highly populated areas: China, India, Nigeria, Europe and Eastern United States.

## 3.2 Sea-salt

Sea-salt is by far the most abundant aerosol species. In the IFS, two parameterizations of sea-salt emissions are present: the
Monahan et al. (1986) scheme which was already present in cycle 32R2 and has been described in M09, and a new scheme following Grythe et al. (2014), which was implemented and became operational in cycle 45R1. The two schemes are denoted hereafter M86 and G14, respectively.

The two schemes M86 and G14 both use mean wind speed as an input. Gustiness is accounted for in mean wind speed by adding a free convection velocity scale based on surface fluxes on sensible and latent heat to the horizontal velocity, following
Beljaars and Viterbo (1998):

$$U_{10} = (U^2 + V^2 + w_*^2)^{1/2}, \tag{1}$$

$$w_* = (z_i g/\theta_v (\overline{w'\theta'_{v0}} + \overline{w'q'_{v0}}))^{1/3}, \tag{2}$$

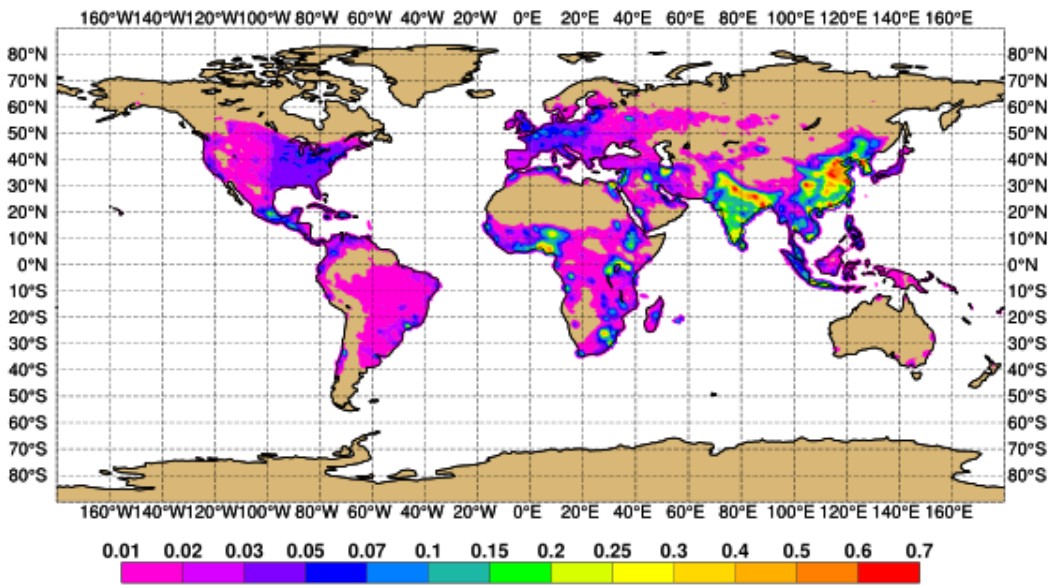

**Figure 1.** Emissions of anthropogenic Secondary Organic Aerosols (SOA) in 2017 in $\mathrm{kg\,m^{-2}\,s^{-1}}$, as used in cycle 45R1 of the IFS.

where $U$ and $V$ are the longitudinal and latitudinal wind speed at the lowest model level, $z_i$ is the PBL height, which is not a very critical input of this formula according to Beljaars and Viterbo (1998) and is taken as 1000 m in this expression, $g$ is the gravitational constant, $\theta_v$ is the virtual potential temperature as defined in Stull (1988) and $\overline{w'\theta'_{v0}}$ and $\overline{w'q'_{v0}}$ are the surface fluxes of sensible and latent heat, respectively.

### 3.2.1  Monahan et al. (1986)

Monahan and Muircheartaigh (1980) suggested that the fraction of sea surface that is covered in white cap follows a wind speed dependency in the form of:

$$W(U_{10}) = 3.84\,10^{-6}\,U_{10}^{3.41} \tag{3}$$

From this, the production flux of sea-salt aerosol is estimated by the following formula (Monahan et al., 1986):

$$\frac{dF}{D_p} = W(U_{10}) \times 3.6 \times 10^5 \times D_p^{-3} \times (1 + 0.057 \times D_p^{1.05}) \times 10^{1.19\exp(-B^2)} \tag{4}$$

where

$$B = \frac{0.38 - \log(D_p)}{0.65} \tag{5}$$

and $D_p$ is the particle diameter.





### 3.2.2 Grythe et al. (2014)

The more recent G14 parameterization has been implemented in the operational cycle 45R1. It combines emissions in different modes: 0.1, 3 and 30 µm dry diameter. A dependency of sea-salt aerosol emissions on sea-surface temperature following Jaeglé et al. (2011) is introduced, which increases emissions over tropics and regions with warmer waters. This important increase of sea-salt aerosol production with temperature is consistent with the conclusions of Sofiev et al. (2011) that modeled marine aerosol optical depth are generally too low in the tropics. Because of the scarcity and heterogeneity of the observational data there are large uncertainties in the temperature dependence of sea-salt aerosol production (Grythe et al., 2014). The production of sea-salt aerosol in G14 can be summarized as:

$$\frac{dF}{D_p} = T_W(T) \left( 235\, U_{10}^{3.5} \exp\left( -0.55(\ln\frac{D_p}{0.1})\right)^2 \right) \tag{6}$$

$$+ T_W(T) \left( 0.2\, U_{10}^{3.5} \exp\left( 1.5(\ln\frac{D_p}{3})\right)^2 \right) \tag{7}$$

$$+ T_W(T) \left( 6.8\, U_{10}^{3} \exp\left( -(\ln\frac{D_p}{30})\right)^2 \right) \tag{8}$$

where the temperature depency factor is:

$$T_W(T) = 0.3 + 0.1\,T - 0.0076\,T^2 + 0.00021\,T^3 \tag{9}$$

Ocean salinity is not an input of the scheme, which is different from other schemes such as Sofiev et al. (2011). Ocean salinity varies a lot regionally, from 10–15 ‰ in the Baltic to more than 38 ‰ in the Mediterranean for example. Cold water tank experiments carried out by Zábori et al. (2012) indicated a dependency of sea-salt aerosol production with salinity for salinity values up to 18 ‰. This means that overall the dependency of sea-salt aerosol production on salinity can be considered as weak, except regionally where salinity values are lower than 18 ‰.

Table 3 show the 2014 emissions in Tg yr$^{-1}$ as estimated by the three scheme for the three sea-salt bins. Since the bin limits are specific to each model, the total emissions of sea-salt particles with a diameter below 10 µm is also shown so as to compared to other models as well as retrievals of emissions. The main difference between the three schemes concerns super-coarse sea-salt, for which emissions are much higher with G14 as compared to M86. This shifts notably the size distribution of sea-salt at emissions towards larger particles with G14. Grythe et al. (2014) provide a best estimate of global emissions derived from NOAA and EMEP PM$_{10}$ observations of 10.2 Pg yr$^{-1}$. Based on this, the G14 emissions are clearly closer to estimates.

Figure 2 shows the 2017 emissions of super coarse sea-salt estimated by the two schemes. The annual production ranges from 0.005-0.02 kg m$^{-2}$ yr$^{-1}$ with G86. Production in the mid-latitudes with G14 is much higher than with M86, with values ranging from 0.1 to 0.4 kg m$^{-2}$ yr$^{-1}$. G14 stands out however more for the large increase of sea-salt production in the tropics, caused by the newly introduced dependency on sea surface temperature (SST). Production in the tropics range from 0.001 to 0.01 kg m$^{-2}$ yr$^{-1}$ for M86 and from 0.05 to 0.2 kg m$^{-2}$ yr$^{-1}$ for G14. It should be noted that over the Great Lakes area production of sea-salt aerosol is not zero for all schemes, which is clearly an artifact of the land-sea mask. This was corrected in later cycles.





**Table 3.** Global emissions in 2014 in $\mathrm{Pg\,yr^{-1}}$ of fine, coarse and super-coarse sea-salt as estimated by the M86 and G14 schemes. The total emissions of particles with a diameter under 10 μm is also shown.

| Sea-salt bin | M86 | G14 |
|---|---|---|
| Fine | 0.022 | 0.024 |
| Coarse | 1.933 | 1.03 |
| Super-coarse | 2.34 | 25.9 |
| Particles with diameter <= 10μm | 2.73 | 9.69 |

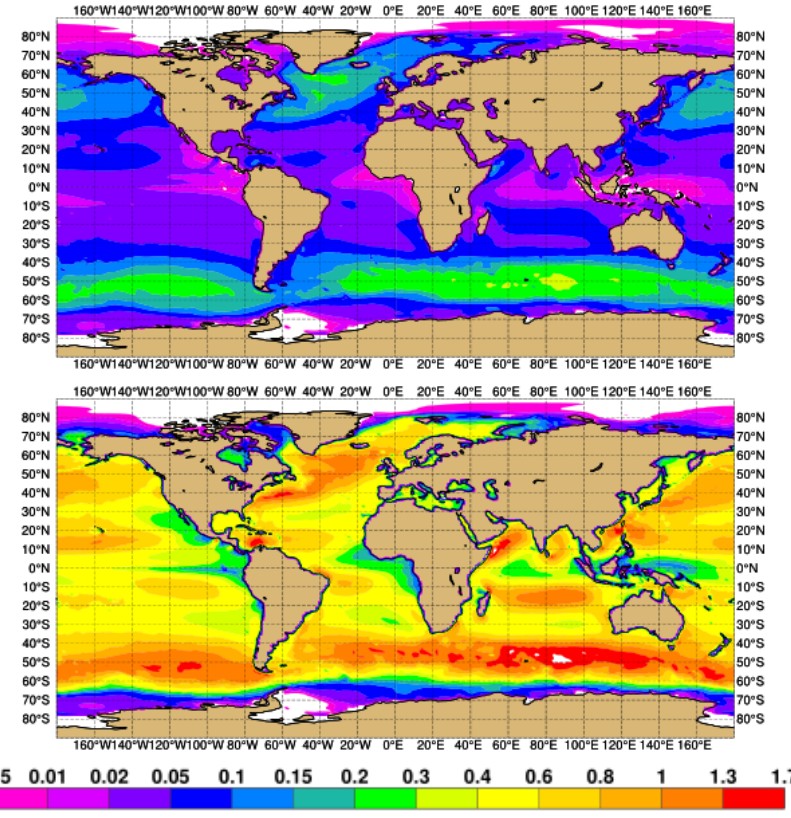

**Figure 2.** 2017 total emissions of sea-salt aerosol from M86 (top) and G14 (bottom), in $\mathrm{kg\,m^{-2}\,yr^{-1}}$.

Figure 3 shows observed and simulated AOD at the AERONET station of Ragged Point in the Antillas, which is one of the few stations that is mostly impacted by sea-salt. Transatlantic transport of dust emitted in the Sahara also occasionally reaches the station. The G14 scheme increases simulated AOD to values that are generally closer to AERONET observations. IFS-AER with M86 generally underestimates AOD over oceans.



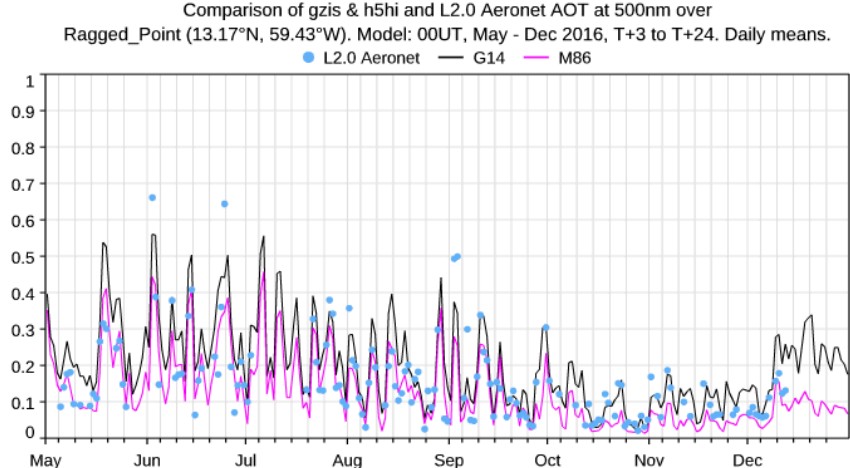

**Figure 3.** 2014 daily AOD at 500 nm at the Ragged Point AERONET station, observations from L2.0 AERONET (blue points), simulated by cycling forecast only IFS-AER using the M86 scheme (violet) and by cycling forecast IFS-AER using the G14 scheme (black)

### 3.3 Dust

The parameterization of dust emissions has been left unchanged since M09. Only the distribution of dust emissions into the three dust bins has been modified. The formulation of Ginoux et al. (2001) is used. The areas likely to produce dust are first diagnosed using a combination of masks: potential dust producing grid-cells must satisfy the following criteria:

– Surface albedo is under 0.52,

– The grid cell is entirely composed of land,

– The snow cover is null,

– The fraction of bare soil is above 0.1,

– There is no ice and no wet skin,

– The fraction of low vegetation is under 0.5,

– There is no high vegetation,

– The standard deviation of subgrid orography is under 50 m.

For a potential dust producing grid-cell, the total dust flux is computed by

$$F(U_{10\,\text{gust}}) = S U_{10\,\text{gust}}^2 (U_{10\,\text{gust}} - U_t) \text{ if } U_{10\,\text{gust}} > U_t \tag{10}$$

$$= 0 \text{ otherwise} \tag{11}$$





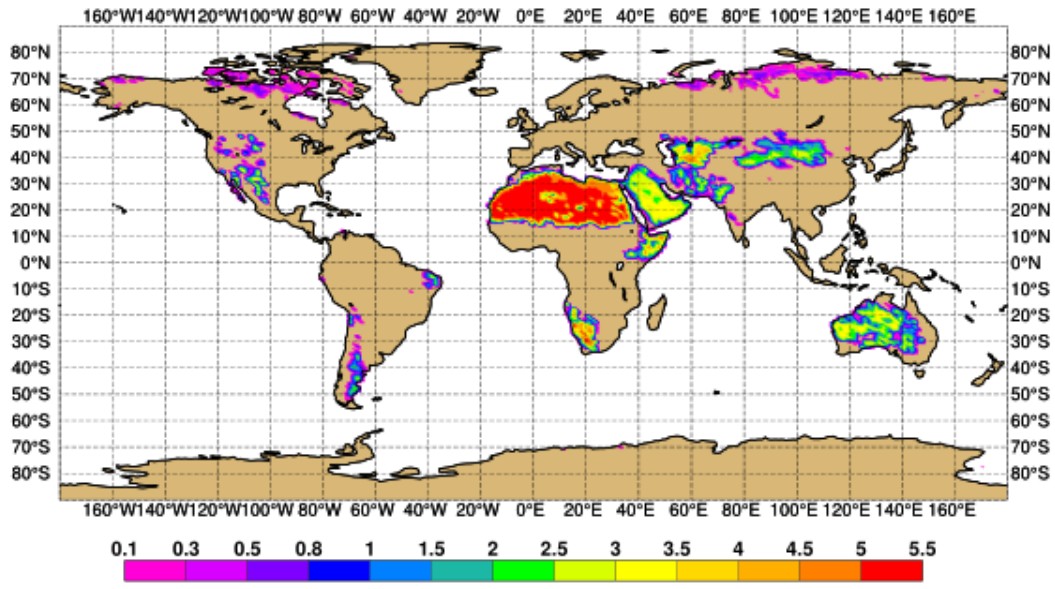

**Figure 4.** 2017 lifting threshold speed in $\mathrm{m\,s^{-1}}$.

where $S$ is a dust source function, $U_t$ is the lifting threshold speed and $U_{10\,\mathrm{gust}}$ are the 3 second wind gusts computed using the mean wind including gustiness effect $U_{10}$ from Eq. 1 (Bechtold and Bidlot, 2009):

$$U_{10\,\mathrm{gust}} = U_{10} + 7.71\, u_* \left(1 + f(\frac{z}{L})\right) \tag{12}$$

where $z$ is the PBL height, taken as 1000 m here, $u_*$ is the surface friction velocity and $L$ is the Monin-Obukhov length-scale

defined as a function of surface fluxes of sensible and latent heat. This follows the parameterization of wind gusts in the IFS until cycle 33R1. The function $f$ can be expressed as:

$$f\left(\frac{z}{L}\right) = 1 + \left(\frac{0.5}{12}\frac{z}{L}\right)^{1/3} \tag{13}$$

The lifting threshold speed is computed as a function of assumed, fixed particle size at emissions and of soil moisture and emission capacity. The lifting threshold speed is similar for each dust bin and is shown in Figure 4. Values are highest over the

Sahara, above $5\ \mathrm{m\,s^{-1}}$, while areas of very low values ($0.1$–$1\ \mathrm{m\,s^{-1}}$) can be found in some boreal regions. The relatively low values over the Taklimakan and Gobi deserts can explain the high emissions over these regions.

The dust source function $S$ is proportional to surface albedo. Equation 13 provides an estimate of the total emitted dust flux, which has to be distributed into the three dust bins. Until cycle 43R1, the distribution was 8% of emissions into fine dust, 31% into coarse dust and 61% into super-coarse dust. Comparing these values to the observed size distribution of dust aerosols

at emission provided by Kok (2011) showed that the relative fraction of super-coarse particles was too low, and the relative fraction of fine particles too high. In the CAMS reanalysis (Inness et al., 2019) and in the operational cycles 43R1 onward, the

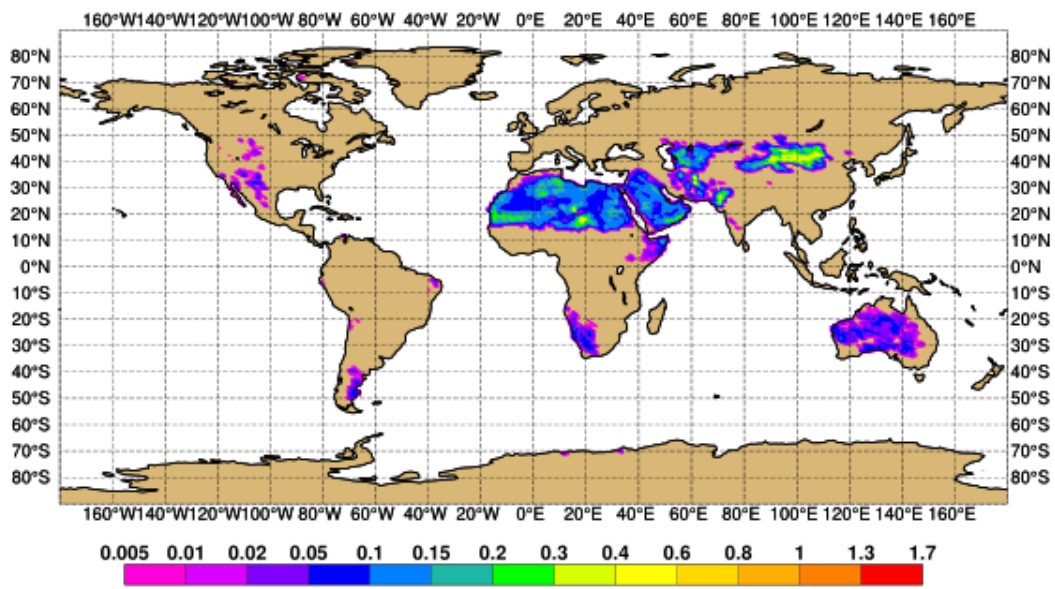

**Figure 5.** 2017 total emissions of dust aerosol in $kg\,m^{-2}\,yr^{-1}$.

distribution of total emissions into the dust bins was revised as follows: 5% into fine dust, 12% into coarse dust and 83% into super-coarse dust. Even though the total emissions are left unchanged, this change of distribution led to a significant decrease in the simulated burden and AOD of dust aerosols because the lifetime of super-coarse dust is shorter than for the other two bins as it is subject to a large sedimentation rate.

Figure 5 shows the 2017 emissions of super-coarse dust. The highest emissions, at $0.2$–$0.3\,kg\,m^{-1}\,yr^{-1}$ occur in the Gobi and Taklimakan deserts, which were impacted by severe dust storms in particular in May 2017. The Sahara, the Arabian Peninsula and parts of Iran and of Turkestan are also prominent. The emissions are very widespread in these regions, which is probably not realistic. Maps of probability of occurrence of observed AOD by MODIS dust AOD above different thresholds were computed using data kindly provided by Paul Ginoux (personal communication), which can serve as dust source functions. These show
much higher maxima and lower minima in the Sahara and Arabian peninsula, which shows that the current operational approach could be refined.

### 3.4 Sulfur dioxide and sulfate

When running standalone (i.e., without the chemistry), sulfur dioxide is included in the tracers of IFS-AER; when running coupled with chemistry, sulfur dioxide is a prognostic species of the chemistry scheme and oxidation rates provided by IFS-
CB05 are used instead. Here we describe emissions and sources of sulfur dioxide in the standalone case. Similarly to OM and BC, emissions from MACCity, CMIP6 and other inventories can be used; the global averages are shown in Table 1. The emissions are the same as used in IFS-CB05. Optionally, these anthropogenic sources can be divided in "low sources", which





take 20% of anthropogenic emissions, and "high sources", which take the remaining 80%. If this option is activated, then "high" sources are released in the first four model levels. This option was not used in operational forecasts except in the CAMS reanalysis. A known issue of the CAMS reanalysis is too high amount of sulfate aerosols above outgassing volcanoes such as the Kilauea in Hawaii or the Popocatepetl in Mexico (Inness et al., 2019). To prevent this, emissions of sulfur dioxide above

volcanoes can optionally be distinguished from the general case: if sulfur dioxide emissions occur above a volcano, then the emissions are distributed between the four model levels that are above the real altitude of the volcano instead of being emitted at surface. Biomass-burning sources of sulfur dioxide are provided by GFAS.

In cycle 38R2 a source of sulfur dioxide from oceanic dimethylsulfide (DMS) was introduced. This new source is parameterized following Liss and Merlivat (1986):

$$F_{\mathrm{DMS}} = 0.5\, Z_l\, [\mathrm{DMS}] M_{\mathrm{SO_2}} \tag{14}$$

where $[\mathrm{DMS}]$ is the concentration of DMS at the surface of the ocean in $\mathrm{nmol\,l^{-1}}$, as provided by an ancillary file, $M_{SO_2}$ is the molar mass of sulfur dioxide and $Z_l$ is a transfer speed computed as a function of wind speed, sea-surface temperature and sea-ice fraction following Curran and Jones (2000):

$$Z_l = 0.17 C_i U_{10} S_c^{0.6667} \text{ if } U_{10} <= 3.6 \tag{15}$$

$$= C_i(2.85 U_{10} - 9.65) S_c^{0.5} \text{ if } U_{10} > 3.6 \text{ and } <= 13 \tag{16}$$

$$= C_i(5.9 U_{10} - 49.3) S_c^{0.5} \text{ if } U_{10} > 13 \tag{17}$$

$$C_i = 1 - \frac{S_i - 0.6}{0.4} \tag{18}$$

where $Si$ is the sea-ice fraction and $S_c$ is the dimensionless Schmidt number which is used to characterize flows in which viscosity and mass transfer are involved. $S_c$ is computed as a function of ocean skin temperature in degree Celsius $T_{sk}$:

$$Sc = \frac{600}{2674 - T_{sk}(147.12 - T_{sk}(3.726 - 0.038 T_{sk}))} \tag{19}$$

In all, the source of sulfur dioxide from oceanic DMS stands at $30\ \mathrm{Tg\,yr^{-1}}$ on average.

Conversion of sulfur dioxide into particulate sulfate is treated in a very simple way following Huneeus (2007). In cycles CY43R1 and before, conversion was parameterized only as a function of latitude, as a proxy to the abundance of the OH radical. The conversion rate (per s) can be written as:

$$C_0 = \frac{\exp\left(-\frac{\delta t}{(C_1 - C_2 \cos\theta)}\right)}{\delta t} \tag{20}$$

where $\delta t$ is the time step, $\theta$ is the angular latitude and $C_1$ and $C_2$ are e-folding times in days representing the lifetime at the pole and the equator set to 8 and 5 days, respectively, for operational cycle up to 43R1. For the CAMS reanalysis and for operational cycles 43R3 and later, the values of $C_1$ and $C_2$ were set to 4 and 3.5 days, respectively, leading to a higher production over most of the Globe.





For the CAMS reanalysis and in operational cycles 43R3 and after, a diurnal cycle, simple dependency on temperature following Eatough et al. (1994) and on relative humidity were introduced and the new conversion rate is expressed as:

$$C = C_0 D(lt) \exp\left(32.37 - \frac{9000}{T}\right) I_{RH} \tag{21}$$

where $D(lt)$ is a cosine diurnal cycle function of local time, with a maximum value of 2 at midday local time and a minimum

value of 0 at midnight local time. $I_{RH}$ is an increment factor set to 2 when $RH$ is above or equal to 98% and set to one otherwise. The difference arises from the fact that where $RH$ is above 98% the grid cell is supposed to be at least partly saturated, which leads to more active conversion from sulfur dioxide to sulphate aerosol.

As shown in Table 4, these modifications led to a significant increase in conversion of sulfur dioxide into particulate sulfate. Sulfur oxidation rates provided by IFS-CB05, which are used when IFS-AER is run coupled with the chemistry, are also shown

and stand between the older and newer value using the conversion scheme of IFS-AER. The mean and median of the conversion process from the AEROCOM phase III (Bian et al., 2017) is also shown, for the year 2008. Accounting for the fact that sulfur dioxide emissions were higher in 2008 than in 2014 by around 7% in the MACCity inventory, the value for IFS-AER cycles 43R3 and later is quite close to the AEROCOM median.

The changes in the sulphate conversion implemented for the CAMS Reanalysis and in the operational cycles 43R3 and

beyond (conversion constants, temperature and relative humidity dependency) are meant to help address the problem of too high sulphate burden in the CAMS interim Reanalysis (Flemming et al., 2017) and also in the operational NRT runs before cycle 43R3. They are meant to reduce the concentrations of sulphate in the mid- and upper-troposphere and shorten the lifetime of both sulphate and sulfur dioxide. With a faster life cycle and reduced concentrations above the planetary boundary layer, the fraction of the mass mixing ratio increments distributed to sulphate during the data assimilation stage has been generally

reduced in the CAMS reanalysis and in cycle 43R3 and beyond, leading to an important decrease of the total burden of sulphate in the CAMS Reanalysis as compared to the CAMS interim Reanalysis, and in the operational cycles CY43R3 and beyond.

**Table 4.** Global conversion of sulfur dioxide into particulate sulfate in 2014 in $\mathrm{Tg\,SO_4\,yr^{-1}}$.

|  | $SO_2$ to $SO_4$ conversion flux |
| --- | --- |
| IFS-AER up to CY43R1 | 69.4 |
| IFS-AER CY43R3 and later | 119.3 |
| IFS-CB05 | 98.3 |
| AEROCOM Phase III Mean/Median for 2008 | 151/139 |

### 3.5 Nitrate and Ammonium

With the important decrease of anthropogenic emissions of sulfur dioxide in recent years, the relative importance of nitrate and ammonium has increased (Bellouin et al., 2011). The production of nitrate and ammonium aerosols in IFS-AER when running





coupled with IFS-CB05 has been introduced in cycle 45R1 but is not used operationally. The parameterization of production of fine mode nitrate and ammonium from gas-to-particle partitioning and of coarse mode nitrate from heterogeneous reactions over dust and sea-salt particles follows the approach of Hauglustaine et al. (2014) which is summarized below. The precursors gases $HNO_3$ and $NH_3$ are prognostic variables of IFS-CB05; their treatment is described in Flemming et al. (2015), while $SO_2$

and $HNO_3$ are evaluated in Huijnen et al. (2019).

### 3.5.1 Gas-to-particle partitioning

The most abundant acids in the troposphere are sulfuric acid ($H_2SO_4$) and nitric acid ($HNO_3$). $NH_3$ acts as the main neutralizing agent for these two species. As a first step, ammonium sulfate is formed from $H_2SO_4$ and $NH_3$, only limited by the less abundant of the two species. This reaction takes priority over the formation of ammonium nitrate ($NH_4NO_3$) because of the low vapor

pressure of sulfuric acid. The main reaction pathways are:

$$NH_3 + H_2SO_4 \longrightarrow (NH_4)HSO_4 \tag{1}$$

$$3NH_3 + 2H_2SO_4 \longrightarrow (NH_4)_3H(SO_4)_2 \tag{2}$$

$$2NH_3 + H_2SO_4 \longrightarrow (NH_4)_2HSO_4 \tag{3}$$

Following Metzger et al. (2002), depending on the relative concentrations of ammonia and sulfate, three domains are considered to characterize how ammonium sulfate is formed. The total ammonia, sulfate and nitrate concentrations are defined as:

$$T_A = [NH_3] + [NH_4{}^+]$$

$$T_S = [SO_4{}^=]$$

$$T_N = [HNO_3] + [NO_3{}^-]$$

For ammonia rich conditions ($T_A > 2T_S$) reaction (3) is considered; for sulfate rich conditions ($T_A <= 2T_S$ and $T_A > T_S$) reaction (2) is considered and finally for sulfate very rich conditions ($T_A <= T_S$) reaction (1) is considered.

As a second step, if $NH_3$ is still present after reactions (1), (2) or (3) then it is used for the neutralization of $HNO_3$ by the

following reaction:

$$NH_3 + HNO_3 \leftrightarrow NH_4NO_3 \tag{4}$$

The equilibrium constant $K_p$ of reaction (4) depends strongly on relative humidity and temperature. The parameterization of Mozurkewich (1993) are used to represent this dependence. Total ammonia that remains after reactions (1), (2) or (3) is written




as:

$$T_A^* = T_A - \Gamma T_S$$

where the value of $\Gamma$ is 1, 1.5 or 2 depending on whether reaction (1), (2) or (3) took place, respectively. If $T_N.T_A^* > K_p$ then ammonium nitrate is formed and its concentration is calculated by:

$$[\text{NH}_4\text{NO}_3] = \frac{1}{2}\left[T_A^* + T_N - \sqrt{(T_A^* + T_N)^2 - 4(T_N T_A^* - K_p)}\right] \tag{5}$$

Otherwise, ammonium nitrate dissociates and

$$[\text{NH}_4\text{NO}_3] = 0 \tag{6}$$

Reaction (5) also allows to compute the concentration of NH$_3$ at equilibrium; the concentration of particulate NH$_4$ is then given by

$$[\text{NH}_4] = T_A - [\text{NH}_3] \tag{7}$$

Finally, the updated concentrations of the precursor gases, [NH$_3$] and [HNO$_3$] are passed back to IFS-CB05.

### 3.5.2 Heterogeneous production

Gaseous HNO$_3$ can also condense on large particles. The formation of smaller nitrate and ammonium particles through gas-to-particle partitioning is solved first because the equilibrium is reached faster (Hauglustaine et al., 2014). After the smaller particles are in equilibrium, the condensation of HNO$_3$ on larger particles is treated. The heterogeneous reactions of HNO$_3$ with calcite (a component of dust aerosol) and sea-salt particles is accounted through the following reactions:

$$\text{HNO}_3 + \text{NaCl} \longrightarrow \text{NaNO}_3 + \text{HCl} \tag{8}$$

$$2\text{HNO}_3 + \text{CaCO}_3 \longrightarrow \text{Ca(NO}_3)_2 + \text{H}_2\text{CO}_3 \tag{9}$$

While the NaCl species is similar to sea-salt aerosols, calcite (CaCO$_3$) is one of the many components of dust aerosol. In Fairlie et al. (2010) and Hauglustaine et al. (2014), the concentration of calcite is taken as 3 or 5% of the total concentration of dust aerosol. An experimental version of IFS-AER that simulates a simplified dust mineralogy was used to compute a climatology of airborne calcite, using as an input the dataset of Journet et al. (2014) which provides an estimate of the calcite content in the clay and silt fraction of soils. Figure 6 shows the vertically integrated fraction of airborne calcite over coarse and super-coarse dust. The regional differences are great, especially for calcite emitted from clay surfaces as compared to coarse dust.



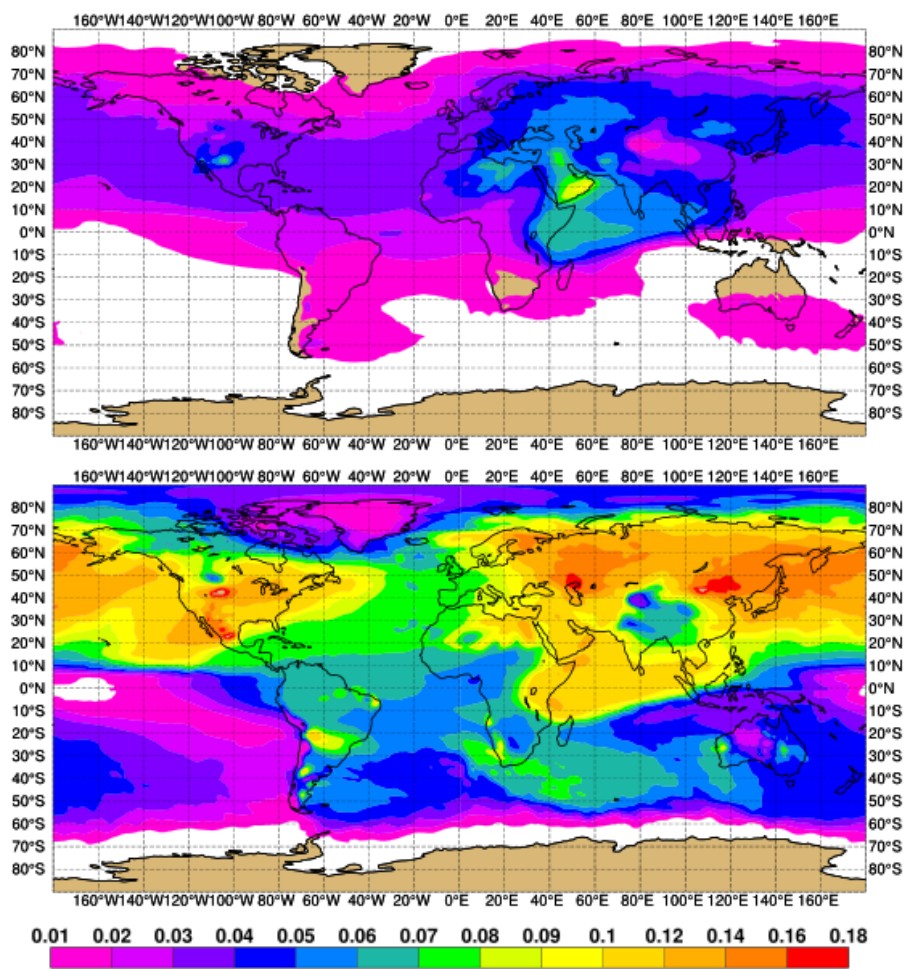

**Figure 6.** Average fraction of calcite over super coarse (top) and coarse (bottom) dust.

A first order update parameterization is used to represent the uptake of HNO$_3$ over sea-salt and calcite particles. The rate constants of reactions (8) and (9) are computed in a simplified way as compared to the original scheme of Hauglustaine et al. (2014) for each sea-salt (SS) and desert-dust (DD) bin $i$:

$$K_6 = 4\pi D_{\text{SS}i}^2 N_{\text{SS}i} \left( \frac{D_{\text{SS}i}}{2D_g} + \frac{4}{\nu\gamma} \right)^{-1} \tag{10}$$

$$K_7 = 4\pi D_{\text{DD}i}^2 N_{\text{DD}i} \left( \frac{D_{\text{DD}i}}{2D_g} + \frac{4}{\nu\gamma} \right)^{-1} \tag{11}$$

where $D_{\text{SS}i}$ is the mass median diameter of sea-salt bin $i$, $D_{\text{DD}i}$ is the mass median diameter of desert dust bin $i$, $N_{\text{SS}i}$ and $N_{\text{DD}i}$ are the number concentration for sea-salt and desert dust bin $i$, respectively, computed using the mass concentration and the mass median diameter. $D_g$ is the pressure and temperature-dependent estimated molecular diffusion coefficient, $\nu$ is





the temperature-dependent estimated mean molecular speed and $\gamma$ is the reactive uptake coefficient. For sea-salt, as in Fairlie et al. (2010), a dependence of the uptake coefficient on relative humidity is used. Similarly to the gas-to particle partitioning reactions, the updated concentration of $HNO_3$ is passed back to IFS-CB05. The concentration of the desert-dust and sea-salt bins are also updated depending on the amount of coarse mode nitrate that is produced.

The production rate of fine mode nitrate in 2014 is estimated at $2.16\,\mathrm{Tg\,N\,yr^{-1}}$, which is significantly below the $3.24\,\mathrm{Tg\,N\,yr^{-1}}$ for the year 2000 reported in Hauglustaine et al. (2014). For coarse mode nitrate, $12.36\,\mathrm{Tg\,N\,yr^{-1}}$ were produced, which is higher than the $11.16\,\mathrm{Tg\,N\,yr^{-1}}$ reported in Hauglustaine et al. (2014). In both cases, different concentration of the precursors gases as well as dust and sea-salt aerosols are a large source of difference between the original implementation and the adaptation in IFS-AER. Figure 7 shows the 2014 average of fine mode, coarse mode nitrate and ammonium mass mixing ratios. Fine

mode nitrate higher surface concentrations are collocated with heavily populated areas and regions with a high agricultural activity, reaching 3 to $5\,\mathrm{\mu g\,m^{-3}}$ over Europe and U.S. and up to $9\text{-}12\,\mathrm{\mu g\,m^{-3}}$ over parts of India and China. Coarse mode nitrate is produced primarily over oceans close to heavily populated areas such as the Eastern and Western extremities of the Atlantic and Pacific oceans. Ammonium surface concentration show similar patterns to fine mode nitrate, with values between 1 and $2\,\mathrm{\mu g\,m^{-3}}$ over Europe, slightly less over U.S. and 3 to $4\,\mathrm{\mu g\,m^{-3}}$ over the heavily populated parts of India and China.

## 3.6    Hygroscopic growth

Hygroscopic growth is the process whereby, for some aerosol species, water is mixed in the aerosol particle, increasing its mass and size and decreasing its density. This process is treated implicitly in IFS-AER, since size is not resolved. It plays an important role however in the computation of optical properties and also for the sinks that are size and/or density dependent, in particular dry deposition. The species subjected to hygroscopic growth in IFS-AER are sea-salt, the hydrophilic components

of OM and BC, sulfate, nitrate and ammonium. The amount of water that is mixed in the aerosol particle depends on particle size. Table 5 details the changes in size for the concerned species. The values are drawn from Tang and Munkelwitz (1994) for sea-salt, Tang et al. (1997) for sulfate and ammonium, from Chin et al. (2002) for BC, and from Svenningsson et al. (2006) for nitrate. For OM, the values are derived from Water Soluble Organic (WASO) of the OPAC (Optical Properties of Aerosols and Clouds) database (Hess et al., 1998)

# 4    Removal processes

Removal processes consist of dry and wet deposition and sedimentation or gravitational settling. Wet deposition and sedimentation are similar to M09 but they are described again here for completeness.

## 4.1    Dry deposition

Two schemes to compute the dry deposition velocities coexist in IFS-AER: the scheme from Reddy et al. (2005) or R05 that

was used in the CAMS reanalysis and in operational cycles up to 43R3, and the newly implemented Zhang et al. (2001) or ZH01 scheme that computes the dry deposition velocities online. Until the operational cycle 45R1, the dry deposition velocity



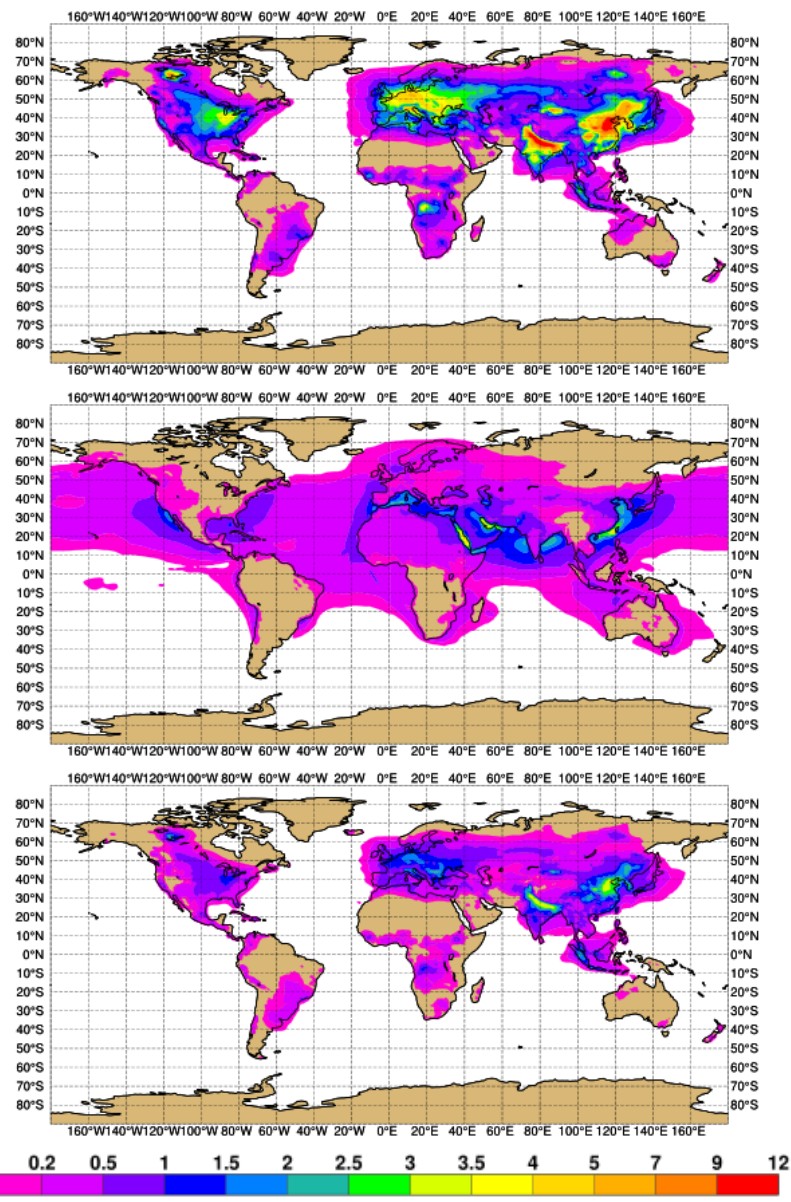

**Figure 7.** 2014 average of surface fine mode (top), coarse mode (middle) nitrate mass and ammonium (bottom) surface concentration in $\mu g\,m^{-3}$.

was used to directly compute a dry deposition flux:

$$F_{DD} = C\,\rho\,V_{DD} \tag{26}$$





**Table 5.** Hygroscopic growth factor depending on ambient relative humidity.

| RH/% | Sea-salt | OM | BC | Sulfate and Ammonium | Nitrate |
|---|---|---|---|---|---|
| 0–40 | 1 | 1 | 1 | 1 | 1 |
| 40–50 | 1.442 | 1.169 | 1 | 1.169 | 1.1 |
| 50–60 | 1.555 | 1.2 | 1 | 1.220 | 1.2 |
| 60–70 | 1.666 | 1.3 | 1 | 1.282 | 1.25 |
| 70–80 | 1.799 | 1.4 | 1 | 1.363 | 1.3 |
| 80–85 | 1.988 | 1.5 | 1.2 | 1.485 | 1.35 |
| 85–90 | 2.131 | 1.55 | 1.3 | 1.581 | 1.5 |
| 90–95 | 2.361 | 1.6 | 1.4 | 1.732 | 1.7 |
| 95–100 | 2.876 | 1.8 | 1.5 | 2.085 | 2.1 |

where $C$ is the aerosol mass mixing ratio at the lowest model level, $\rho$ is the air density and $V_{DD}$ is the dry deposition velocity. Since the operational cycle 45R1, the dry deposition velocity is passed through to the vertical turbulent diffusion scheme, with the surface flux unchanged. The difference between the two approaches was shown to be extremely small.

Also, since cycle 43R3, the dry deposition of sulfur dioxide is represented and is described below.

### 4.1.1 R05 dry deposition velocities

In the R05 scheme, dry deposition velocities are fixed for each aerosol tracer over continents and oceans. The values used in IFS-AER are shown in table 6: the values over oceans and land differ only for sea-salt aerosols. Since operational cycle 45R1, a cosine function of local time is applied as a diurnal cycle modulation of the fixed velocities, with a maximum of 1.7 at midday local time and a minimum of 0.3 at midnight. This is to account for the fact that dry deposition velocities display a marked diurnal cycle (Zhang et al., 2003) because of lower aerodynamic and canopy resistance. Also, over ice and snow surfaces, dry deposition velocities cannot exceed 0.3 mm s$^{-1}$.

### 4.1.2 ZH01 dry deposition velocities

The ZH01 scheme is itself based on the dry deposition model of Slinn (1982). The deposition velocity at surface is evaluated for all aerosol prognostic variables as:

$$V_{\text{DD}} = \frac{1}{A_r + S_r} \tag{27}$$

As compared to the original implementation of this scheme in Zhang et al. (2001), gravitational settling is not included in this equation as it is taken care of in another routine. $A_r$ is the aerodynamic resistance, independent of the particle type, computed





**Table 6.** Dry deposition velocities in the R05 scheme, in $\text{cm s}^{-1}$.

| Species | Values over continents | values over oceans |
|---|---|---|
| fine mode sea-salt | 1.1 | 1.1 |
| coarse mode sea-salt | 1.2 | 1.15 |
| super-coarse sea-salt | 1.5 | 1.2 |
| fine mode dust | 0.02 | 0.02 |
| coarse mode dust | 0.1 | 0.1 |
| super-coarse mode dust | 1.2 | 1.2 |
| OM | 0.1 | 0.1 |
| BC | 0.1 | 0.1 |
| Sulfate | 0.25 | 0.15 |
| Nitrate and ammonium | 0.15 | 0.15 |

by

$$A_r = \frac{\ln\left(\frac{z}{z_0}\right)}{ku_*} \tag{28}$$

Where $k$ is the von Karman constant, $z_0$ is the roughness length provided by the IFS, $z$ the height of the first model level and $u_*$ the surface friction velocity. $S_r$ in Equation (27) is the surface resistance:

5   $$S_r = \frac{1}{3u_*(E_\text{B} + E_\text{IM} + E_\text{IN})} \tag{29}$$

where $E_\text{B}$, $E_\text{IM}$ and $E_\text{IN}$ are the collection efficiencies for Brownian diffusion, impaction and interception, respectively.

$$E_\text{B} = Sc^{-Y_\text{R}} \tag{30}$$

where $Sc$ is the particle Schmidt number computed by $\frac{\nu}{D}$ where $\nu$ is the kinematic viscosity of air and $D$ is the particle diffusion coefficient, $Y_\text{R}$ is a surface-dependent constant with values provided in Table 3 of Zhang et al. (2001).

10   $$E_\text{IM} = \left(\frac{St}{\alpha + St}\right)^2 \tag{31}$$

where $St$ is the Stokes number for smooth and rough flow regime:

$$St = V_\text{g}\frac{u_*^2}{D_\text{visc}} \quad \text{smooth surface:} z_0 < 1mm, \tag{32}$$

$$St = V_\text{g}\frac{u_*}{(gC_\text{R})} \quad \text{rough surface:} z_0 > 1mm \tag{33}$$





where $D_{visc}$ is the dynamic viscosity of air, computed as a function of temperature only, and $V_{\mathrm{g}}$ is the gravitational velocity computed as

$$V_{\mathrm{g}} = 2\rho \frac{D_{\mathrm{p}}^2 g C_{\mathrm{F}}}{(18 D_{\mathrm{visc}})} \tag{34}$$

$C_{\mathrm{F}}$ is the Cunningham slip correction to account for the viscosity dependency on air pressure and temperature. $\rho$ and $D_{\mathrm{p}}$ are

the particle density and diameter respectively. For $D_{\mathrm{p}}$, the mass median diameter (MMD) of each aerosol prognostic variable is used, and hygroscopic growth is accounted for the relevant species. The Cunningham slip correction is defined differently from the original Zhang et al. (2001) implementation:

$$C_{\mathrm{F}} = \exp(16\,\sigma) + 1.246 \exp(3.5\ln(2\sigma)) \times 2\frac{\lambda}{D_{\mathrm{p}}} \tag{35}$$

where $\lambda$ is the mean free path of air molecules, and $\sigma$ the standard deviation of the assumed log-normal distribution of the

considered particle. The impact of the different formulation of $C_{\mathrm{F}}$ has been shown to be extremely small, $\alpha$ and $C_{\mathrm{R}}$ are surface-dependent constants, whose values are provided in Table 3 of ZH01. Finally,

$$E_{\mathrm{IN}} = 0.5\frac{D_{\mathrm{p}}}{C_{\mathrm{R}}} \tag{36}$$

The IFS surface model distinguishes 9 surface classes, which are given as fractions (tiles) for each grid box. The two vegetation tiles ("high" and "low" vegetation) are further classified according to 20 vegetation tiles. For the low and high

vegetation tiles the IFS vegetation types were mapped to the 15 land classes of the ZH01 surface classes.

The dry deposition velocity computed with this algorithm is computed three times for the three dominant tile fractions of each grid cell if they are defined, which gives a component of sub-grid variability to the ZH01 scheme as it is implemented in IFS-AER. The final dry deposition velocity is the average of these three dry deposition velocities weighted by the relative fraction of the three dominant tile fractions. Table 7 provide a comparison of the global dry deposition velocities in 2014

computed with the R05 and ZH01 methods. Values are on average generally lower with ZH01 as compared to R05 for fine particles. For coarse and super-coarse particles on the other hand, values estimated with ZH01 are on average higher. Figure 8 shows monthly averages of dry deposition velocities of super coarse sea-salt for January and July 2014. Values with R05 differ only for regions where snow or sea-ice is present because of the $3\,\mathrm{mm\,s^{-1}}$ threshold over these areas. Elsewhere, values are very close between oceans and continents as the prescribed value is very close for both surfaces: 1.2 and $1.5\,\mathrm{cm\,s^{-1}}$, respectively.

Values with ZH01 also show this dichotomy between regions free of ice and snow and the rest; however the dry deposition velocities vary a lot more elsewhere also, with generally higher values over continents than over oceans because of rougher surfaces. This is more marked for dry deposition velocities of super-coarse sea-salt or dust, for which values over continents are 2 to 4 times larger than over oceans.

The impact of using the ZH01 or the R05 dry deposition schemes is important on simulations of aerosol optical depth and

even more so on simulations of surface concentration and of PM. There are still some issues with the ZH01, and notably the too high values of dry deposition velocities and fluxes over mountainous terrain. These have been addressed in later versions of IFS-AER.



**Table 7.** 2014 global average of the dry deposition velocities computed with R05 and ZH01, in $\mathrm{cm\,s^{-1}}$.

| Species | R05 | ZH01 |
|---|---|---|
| fine mode sea-salt | 0.89 | 0.057 |
| coarse mode sea-salt | 0.94 | 0.95 |
| super-coarse sea-salt (includes sedimentation) | 1.035 | 1.2 |
| fine mode dust | 0.017 | 0.075 |
| coarse mode dust | 0.084 | 0.061 |
| super-coarse mode dust (includes sedimentation) | 0.98 | 1.14 |
| OM | 0.084 | 0.066 |
| BC | 0.084 | 0.079 |
| Sulfate | 0.14 | 0.21 |

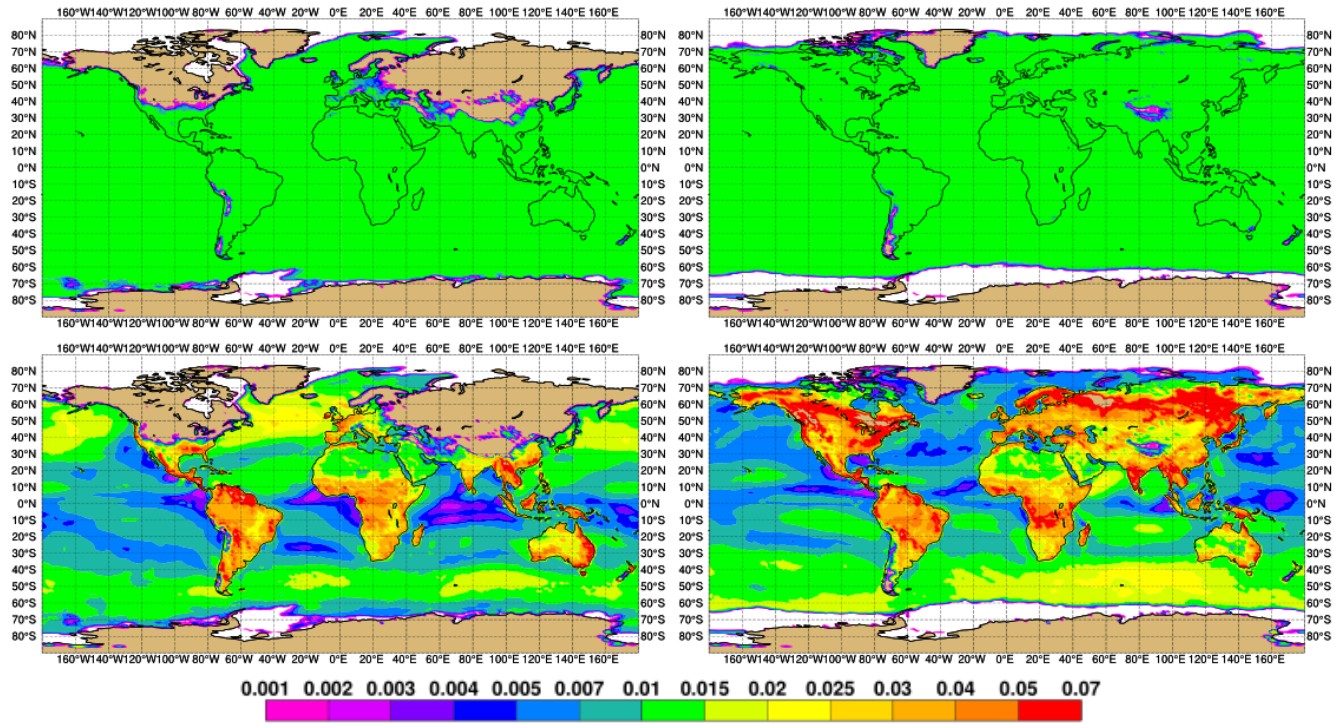

**Figure 8.** January (left) and July (right) 2014 average of dry deposition velocity of super coarse sea-salt computed with R05 (top) and ZH01 (bottom), in $\mathrm{m\,s^{-1}}$.



### 4.1.3 Dry deposition of sulfur dioxide

Dry deposition is an important sink for gaseous sulfur dioxide. For IFS-AER in standalone mode, this process has been represented since cycle 43R3 and in the CAMS Reanalysis. The approach is different from the other aerosol tracers and is similar to what is done for sulfur dioxide in IFS-CB05. Monthly sulfur dioxide dry deposition velocities have been computed offline

using the approach described in Michou et al. (2004). These dry deposition velocities are applied to the sulfur dioxide tracer in IFS-AER, modulated by the same diurnal cycle as used for the R05 dry deposition velocities. Dry deposition of sulfur dioxide in 2017 was estimated at 51 Tg yr$^{-1}$, to be compared with 138 Tg yr$^{-1}$ emissions.

### 4.2 Sedimentation

Sedimentation was also left broadly unchanged as compared to M09. It is applied only for super-coarse dust and sea-salt, for

which it is an important sink. The change in mass mixing ratio from sedimentation follows the approach of Tompkins (2005) for ice sedimentation. The change in mass concentration caused by a transport in flux form at velocity $V_s$ is given by:

$$\frac{dC}{dt} = \frac{1}{\rho}\frac{d(\rho V_s C)}{dz} \tag{37}$$

where $\rho$ is the air density. The integration of this gives for each level $k$ and time step $j$:

$$C_{k+1}^{j} = \frac{\dfrac{\rho j - 1 V_s C_{k+1}^{j-1}}{\rho^j \Delta Z}\Delta t + C_k^j}{1 + \dfrac{\rho^j V_s}{\rho^j \Delta Z}\Delta t} \tag{38}$$

which is solved from top to bottom. The gravitational velocity $V_s$ is constant in time and space for the two sedimented species and is computed using Stokes' law:

$$V_s = \frac{2\rho_p g}{9\mu}\, r^2\, C_F \tag{39}$$

where $\rho_p$ is the particle density, $g$ the gravitational constant, $\mu$ the air viscosity and $C_F$ the Cunningham correction factor.

### 4.3 Wet deposition

Wet deposition has been modified very little compared to M09. All aerosol tracers are subjected to wet deposition except hydrophobic OM and BC as well as sulfur dioxide. Both in-cloud (or rainout) and below-cloud (or washout) processes are represented.

### 4.3.1 In-cloud wet deposition (rainout)

The in-cloud scavenging rate in s$^{-1}$ at model level $k$ of an aerosol $i$ is written as follows:

$W_{i,k}^{I} = \beta_k f_k D_i$           (40)





**Table 8.** Value of the parameter $D$, representing the fraction of aerosol included in cloud droplet.

| Species | $D$ value |
|---|---|
| Sea-salt | 0.9 |
| Dust | 0.7 |
| OM hydrophilic | 0.7 |
| BC hydrophilic | 0.7 |
| Sulfate | 0.7 |
| Nitrate and ammonium | 0.4 |

where $D_i$ is the fraction of aerosol $i$ that is included in cloud droplets and $f_k$ is the cloud fraction at level $k$. The value of the parameter $D_i$ is indicated in Table 8. Following Giorgi and Chameides (1986), $\beta_k$ is the rate of conversion of cloud water to rain water, computed by comparing the precipitation flux at levels $k$ and $k+1$ is written as follows:

$$\beta_k = \frac{P_{k+1} - P_k}{\rho_k \, \Delta z_k \, f_k \, q_k} \tag{41}$$

where $P_k$ is the sum of rain and snow precipitation fluxes at level $k$, $q_k$ the sum of the liquid and ice mass mixing ratio and $\Delta z_k$ is the layer thickness at level $k$. This means that, as in M09, no distinction is made between rain and snow.

### 4.3.2 Below-cloud wet deposition (washout)

The below cloud scavenging rate at model $k$ of an aerosol $i$ is given by:

$$W_{i,k}^{\mathrm{B}} = \frac{3}{4} \left( \frac{P_{k\mathrm{l}} \alpha_\mathrm{l}}{R_{r\mathrm{l}} \rho_\mathrm{l}} + \frac{P_{k\mathrm{i}} \alpha_\mathrm{i}}{R_{r\mathrm{i}} \rho_\mathrm{i}} \right) \tag{42}$$

Where $P_{k\mathrm{l}}$ and $P_{k\mathrm{i}}$ are the mean liquid and solid precipitation fluxes respectively, $\rho_\mathrm{l}$ and $\rho_\mathrm{i}$ the water and ice density, $\alpha_\mathrm{l}$ and $\alpha_\mathrm{i}$ the efficiency with which aerosol variables are washed out by rain and snow, respectively, which account for Brownian diffusion, interception and inertial impaction. The values used in IFS-AER for $\alpha_\mathrm{l}$ and $\alpha_\mathrm{i}$ are 0.001 and 0.01, respectively.

### 4.4 Ageing

For OM and BC, once emitted, the hydrophobic component is transformed into a hydrophilic one with an exponential lifetime
of 1.16 days, shorter than in Reddy et al. (2005) where the lifetime is 1.63 days. This is closer to more recent measurements of black carbon ageing, which ranges from 8 to 23 hours over Beijing and Houston, respectively (Wang et al., 2018).





## 5   Optical properties and PM formula

### 5.1   Optical properties

In Cycle 45R1, the aerosol optical property diagnostics consist of total and fine-mode aerosol optical depth (AOD), absorption AOD (AAOD), single scattering albedo (SSA) and the asymmetry factor, which are computed as column properties over 20

wavelengths between 340 nm and 10 μm. A lidar emulator has been implemented in IFS-AER which also takes into account Rayleigh scattering and gaseous scattering to provide profiles of the attenuated backscattering signal from ground or from satellite, at 355, 532 and 1064 nm. The profile of total aerosol extinction coefficient is also output at these three wavelengths. These diagnostics are using values of mass extinction, SSA, asymmetry and lidar ratio for each aerosol species that have been pre-computed with a standard code for Mie scattering based on Wiscombe (1980). Spherical shape is assumed for all species,

with a number size distribution described by a mono-modal or bi-modal log-normal functions. More details on the specifics of the computation of the aerosol optical properties can be found in Bozzo et al. (2019)

Table 9 lists the relevant parameters of the distribution for each species used in the computations of the optical diagnostics. For the hydrophilic types the optical properties change with the relative humidity due to the swelling of the water soluble component in wetter environments. The growth factors applied are detailed in Table 5. A summary of the refractive index

associated to each aerosol type is given in the following paragraphs.

### 5.1.1   Organic matter

The optical properties are based on the "continental" mixtures described in Hess et al. (1998). We use a combination of 13% in mass of insoluble soil and organic particles, 84% of water soluble particles originated from gas to particle conversion containing sulfates, nitrates and organic substances and a 3% of soot particles. The combination gives optical properties representing an

average of biomass and anthropogenic organic carbon aerosols. The refractive indices and the parameters used in the particle size distribution of each component are as described in Hess et al. (1998). The hydrophobic organic matter type uses the same set of optical properties but for a fixed relative humidity of 20%.

### 5.1.2   Black carbon

The refractive index used in the Mie computations is based on the OPAC SOOT model. At the moment the hydrophilic type of

the black carbon species is not implemented and both types are treated as independent from the relative humidity. The single particle properties are integrated with a log-normal particle size distribution for sizes between 0.005 and 0.5 μm.

### 5.1.3   Sulfate

The refractive index is taken from the Global Aerosol Climatology Project (GACP, http://gacp.giss.nasa.gov/data_sets/) and it is representative of dry ammonium sulfate.





### 5.1.4 Mineral dust

The large uncertainty in mineral dust composition (e.g., Colarco et al., 2014) means that it is difficult to represent the radiative properties of this species with a single refractive index fitting different parts of the world. The refractive indexes of Woodward (2001) are used, which were estimated by combining measurements from different locations and which provides the largest absorption in the visible range as compared to other estimates of dust refractive indexes such as Fouquart et al. (1987) or Dubovik et al. (2002), with an imaginary refractive index at 500 nm of $n_{i,500} = 0.0057$. The optical properties are computed individually for each of the three size intervals in the CAMS mineral dust model, using a log-normal size distribution with limits in particle radius of 0.03, 0.55, 0.9 and 20 μm.

### 5.1.5 Sea salt

The refractive index for sea water is as in the OPAC database and the optical properties are integrated across the three size ranges in the CAMS model, using bimodal lognormal distributions with limits of particle radius set at 0.03, 0.05, 5 and 20 μm as in Reddy et al. (2005).

### 5.2 PM formulae

Particulate Matter smaller than 1, 2.5 and 10 μm are important outputs of IFS-AER. They are computed in cycle 45R1 with the following formulae that uses the mass mixing ratio from each aerosol tracer as an input:

$$PM_1 = \rho\Big(\frac{[SS_1]}{4.3} + 0.97[DD_1] + 0.6[OM] + 0.6[BC]\Big)$$

$$PM_{2.5} = \rho\Big(\frac{[SS_1]}{4.3} + 0.5\frac{[SS_2]}{4.3} + [DD_1] + [DD_2]$$

$$+ 0.7[OM] + [BC] + 0.7[SU] + 0.7[NI_1]$$

$$+ 0.25[NI_2] + 0.7[AM]\Big)$$

$$PM_{10} = \rho\Big(\frac{[SS_1]}{4.3} + \frac{[SS_2]}{4.3} + [DD_1] + [DD_2]$$

$$+ 0.4[DD_3] + [OM] + [BC] + [SU] + [NI_1]$$

$$+ [NI_2] + [AM]\Big)$$

where $\rho$ is the air density. The sea-salt aerosol tracers are divided by 4.3 so as to transform the mass mixing ratio at 80% ambiant relative humidity to dry mass mixing ratio.

**Table 9.** Refractive index and parameters of the size distribution associated to each aerosol type in the CAMS model ($r_{\mathrm{mod}}$ =mode radius, $\rho$=particle density, $\sigma$=geometric standard deviation). Values are for the dry aerosol a part from sea salt which is given at 80% RH. The organic matter type is represented by a mixture of three OPAC types similar to the average continental mixture, as described in Hess et al. (1998).

| Aerosol type | Size bin limits (sphere radius, $\mu$m) | Refractive index source | $\rho$ (kg m$^{-3}$) | $r_{\mathrm{mod}}$ ($\mu$m) | $\sigma$ |
|---|---|---|---|---|---|
| Sea Salt* (80% RH) | 0.03-0.5 / 0.5-5.0 / 5.0-20 | OPAC | 1183 | 0.1992,1.992 | 1.9,2.0 |
| Dust | 0.03-0.55 / 0.55-0.9 / 0.9-20 | Woodward (2001) | 2610 | 0.29 | 2.0 |
| Black carbon | 0.005-0.5 | OPAC (SOOT) | 1000 | 0.0118 | 2.0 |
| Sulfates | 0.005-20 | Lacis et al. (GACP) | 1760 | 0.0355 | 2.0 |
| Organic matter[+] | 0.005-20 | WASO+ | 1800 | 0.0212 | 2.24 |
| | | OPAC INSO+ | 2000 | 0.471 | 2.51 |
| | | SOOT | 1000 | 0.0118 | 2.00 |

*Sea salt is described by a bi-modal log-normal distribution with fixed number concentrations of 70 cm$^{-3}$ and 3 cm$^{-3}$ for the small and the large mode, respectively.

[+]The species are mixed by number concentration. The individual number concentrations are 12000 cm$^{-3}$ (WASO), 0.1 cm$^{-3}$ (INSO), 8300 cm$^{-3}$ (SOOT) The hydrophobic component of organic matter uses the same optical properties but for a fixed relative humidity of 20%.

## 6 Operational configuration

IFS-AER cycle 45R1 is operated by ECMWF to provide operational Near-Real-Time aerosol products in the framework of the Copernicus Atmospheric Monitoring Services. The model is run in assimilation mode, using AOD observations from MODIS collection 6 (Levy et al., 2013) and from the Polar Multi Angle Product (PMAP; Popp et al., 2016). Before cycle
5 45R1, only MODIS AOD was assimilated. IFS-AER cycle 36R1, 40R2 and 42R1 were used in assimilation to produce the MACC reanalysis (Inness et al., 2013), the CAMS interim reanalysis (Flemming et al., 2017) and the CAMS reanalysis (Inness et al., 2019). The operational configuration and the changes brought by successive cycles are presented in https://atmosphere. copernicus.eu/node/326. A summary of the operational configurations of the latest versions of the NRT system during the CAMS and MACC projects, as well as the three reanalysis is shown in table 10.
10 The horizontal resolution was updated in June 2016, increasing from $T_L 255$ (approximately 80 km grid size) to $T_L 511$ (40 km). The vertical resolution is planned to increase from 60 to 137 levels in the future upgrade to cycle 46R1. Also, the CAMS reanalysis as well as the operational cycle 45R1 are run with interactive aerosols as an input of the radiative scheme to





compute aerosol radiative interaction. The specific treatment of $SO_2$ emissions over outgassing volcanoes has been introduced in cycle 45R1 also. The oceanic DMS source of sulfur dioxide was implemented in cycle 37R3 in April 2013.

**Table 10.** IFS versions and options for the aerosols used operationally for Near-Real-Time global CAMS products. MF stands for Mass fixer, DDEP for dry deposition and SCON for sulfate conversion. G01bis is for the Ginoux et al. (2001) dust emission scheme with modified distribution of the emissions into the dust bins. R05bis is for the updated simple sulphate conversion scheme with temperature and relative humidity dependency

| Model Version | Date | Resolution | Emissions | | | | | MF | DDEP | SCON |
|---|---|---|---|---|---|---|---|---|---|---|
| | | | Sea-salt | Dust | OM | BC | SO2 | | | |
| CY37R3 | 04/2013 | T255L60 | M86 | G01 | EDGAR | EDGAR | EDGAR | No | R05 | R05 |
| CY40R2 | 09/2014 | T255L60 | M86 | G01 | EDGAR | EDGAR | EDGAR | No | R05 | R05 |
| CY41R1 | 09/2015 | T255L60 | M86 | G01 | EDGAR | EDGAR | EDGAR | No | R05 | R05 |
| CY41R1 | 06/2016 | T511L60 | M86 | G01 | EDGAR | EDGAR | EDGAR | No | R05 | R05 |
| CY43R1 | 01/2017 | T511L60 | M86 | G01bis | MACCity +SOA | MACCity | MACCity | Yes | R05 | R05 |
| CY43R3 | 09/2017 | T511L60 | M86 | G01bis | MACCity +SOA | MACCity | MACCity | Yes | R05+SO$_2$ | R05bis |
| CY45R1 | 06/2018 | T511L60 | G14 | G01bis | MACCity +SOA | MACCity | MACCity | Yes | ZH01+SO$_2$ | R05bis |
| MACCRA | 2013 | T255L60 | M86 | G01 | EDGAR | EDGAR | EDGAR | No | R05 | R05 |
| CAMSiRA | 2016 | T159L60 | M86 | G01 | EDGAR | EDGAR | EDGAR | No | R05 | R05 |
| CAMSRA | 2018 | T255L60 | M86 | G01bis | MACCity +SOA | MACCity | MACCity | Yes | R05 | R05bis |

## 7 Budgets and simulated fields

### 7.1 Configuration

5  IFS-AER was run standalone in cycling forecast mode, without data assimilation or coupling with the chemistry, from May 2016 to May 2018 inclusive at a resolution of $T_L511L60$, using emissions and model options similar to the operational NRT run. Budgets are shown for June 2016 to May 2018 to allow for a month of spin-up time. The simulated AOD and PM are shown for 2017.

### 7.2 Budgets

10  Budgets are presented in Table 11, with a comparison to values from GEOS-CHEM version 9-01-03 found in Croft et al. (2014) where applicable. For both sea-salt and dust, the particle size has an important impact on lifetime: the larger particles have a much shorter lifetime. Particle size also matters for the repartition of sinks between wet and dry deposition; for larger particles,





**Table 11.** IFS-AER budgets for the June 2016 to May 2018 period. Fluxes are expressed in Tg yr$^{-1}$, burdens in Tg, and lifetimes in days. Equivalent from GEOS-CHEM from Croft et al. (2014) when available and comparable are indicated in hyphens

| Species | Source | Dry dep + sedim | Wet dep | Chemical conv | Burden | Lifetime |
|---|---|---|---|---|---|---|
| fine mode sea-salt | 44.1 | 3.85 | 40.2 | 0 | 0.146 | 1.2 |
| coarse mode sea-salt | 1908 | 987 | 921 | 0 | 3.37 | 1.2 |
| super-coarse sea-salt | 48044 | 39810 | 8234 | 0 | 24.9 | 0.2 |
| fine mode dust | 86.7 | 21.4 | 65.3 | 0 | 1.6 | 6.8 |
| coarse mode dust | 289.1 | 65.8 | 223.3 | 0 | 5.6 | 7.1 |
| super-coarse mode dust | 20228 | 1743 | 279 | 0 | 8.2 | 1.5 |
| Hydrophobic OM | 112.4 | 19.1 | 1.5 | -92.4 | 0.4 | 7.1 |
| Hydrophilic OM | 112.4 | 38.5 | 166.2 | 92.4 | 2.2 (0.61 total) | 3.9 (5.9 total) |
| Hydrophobic BC | 8.7 | 1.4 | 0.1 | -7.2 | 0.032 | 7.8 |
| Hydrophilic BC | 2.2 | 1.4 (1.23) | 8(5.65) | 7.2 | 0.12 (0.12 total) | 4.7 (6.1) |
| SO$_2$ | 138.3 | 51.4 | 0 | -86.1 | 0.28 | 0.74 |
| Sulfate | 0 | 19.6 (4.2) | 109.9 (152) | 129.2 | 1.1 (1.1) | 3.1 (2.6) |

dry deposition becomes preponderant. This is also because for super-coarse particles, sedimentation is also included in the dry deposition process in these tables.

These numbers can be compared to the detailed budgets of GEOS-CHEM presented in Croft et al. (2014). For dust, sea-salt and sulfates, the lifetime values are comparable, though slightly shorter for sulphate in GEOS-CHEM with a lifetime of 2.6 days against 3.1 days in IFS-AER. For OM and BC, lifetime values are significantly shorter than in GEOS-CHEM (6.1 and 5.9 days for BC and OM, respectively). The OM burden is much smaller with GEOS-CHEM, which comes from the different treatment of secondary organics between the two simulations. Also, the distribution between dry and wet deposition appears to be relatively more in favour of dry deposition in IFS-AER for sulphate as compared to GEOS-CHEM. While wet deposition of sulphate is lower with IFS-AER (109.9 Tg yr$^{-1}$ against 152 Tg yr$^{-1}$), wet deposition of hydrophilic BC is much higher wityh IFS-AER, at 8 Tg yr$^{-1}$ against 5.65 Tg yr$^{-1}$. A possible explanation is that rainout from large scale precipitation is lower with IFS-AER as compared to GEOS-CHEM, while the contribution from convective precipitation is higher.

## 7.3 Simulated AOD and PM

Figure 9 shows total and speciated AOD at 550 nm for 2017. The highest values are found in the dust producing regions of North Africa, the middle East and the Gobi–Taklimakan desert, in the heavily populated regions of the Indian subcontinent and eastern China, and in the most active seasonal biomass burning region in the world which is equatorial Africa. Sea-salt AOD is quite evenly spread between the mid-latitude regions where mean winds are high, and the tropics where trade winds are





on average less intense, but with a relatively more active sea-salt production thanks to the dependency of sea-salt production on SST. Transatlantic transport of dust produced in the western Sahara is a prominent feature, which can be compared to simulations from other models (Schepanski et al., 2009). Spatial differences in the Sahara are not very pronounced, and very active dust producing regions such as the Bodele depression do not appear, which could be due to a dust source function that does not discriminate enough. OM is a species that combine anthropogenic and biomass burning sources: AOD are highest over parts of China and India, mostly from secondary organics, and equatorial Africa, from biomass burning. BC sources are also a combination of anthropogenic and biomass burning origin, the patterns are close to what is simulated for OM. Sulfate AOD is concentrated over heavily populated areas, and a few outgassing volcanoes such as Popocatepetl in Mexico and Kilauea in Hawaii. Oceanic DMS sources bring a "background" of sulfate AOD over most oceans.

Figure 10 shows the global simulated $PM_{2.5}$ and $PM_{10}$ for 2017. Mean values over 70 to 100 $\mu g \, m^{-3}$ for $PM_{2.5}$ and 150 to 300 $\mu g \, m^{-3}$ occur mainly over desert areas. The transatlantic transport of dust particles from the Sahara is a prominent feature, with mean $PM_{10}$ values of 20–25 $\mu g \, m^{-3}$ in the Caribbean islands, while in the Pacific Ocean West of Panama average $PM_{10}$ values are below 10 $\mu g \, m^{-3}$. Seasonal biomass burning regions such as Indonesia, Brazil and Equatorial Africa, and few extreme fires over Eastern Siberia, United States and Canada reach 70 to 100 $\mu g \, m^{-3}$ for both $PM_{2.5}$ and $PM_{10}$; the difference between the two is negligible since OM and BC contribute their whole mass to both $PM_{2.5}$ and $PM_{10}$. Heavily populated areas with high pollution such as China and the Indian subcontinent also show high values, reaching average values between 40 to 70 $\mu g \, m^{-3}$ for $PM_{2.5}$ and up to 100–150 $\mu g \, m^{-3}$ for $PM_{10}$. The difference between $PM_{2.5}$ and $PM_{10}$ for these areas can be explained by dust sources from the Gobi and Taklimakan deserts for China and from the Thar desert in India and Pakistan. Over most of oceans and outside of the influence of other species, $PM_{2.5}$ and $PM_{10}$ from marine aerosol reach average concentrations of 5–10 $\mu g \, m^{-3}$ and 10–15 $\mu g \, m^{-3}$, respectively. Over a large part of Europe and United States, average values of $PM_{2.5}$ and $PM_{10}$ are comprised between 5 and 15 $\mu g \, m^{-3}$ and 10 to 25 $\mu g \, m^{-3}$, respectively.

## 8   Evaluation

In this section, a short evaluation of the simulated AOD against observations from the Aerosol Robotic Network (AERONET; Holben et al., 1998) is shown, and of $PM_{2.5}$ and $PM_{10}$ against observations from the Airnow and Airbase networks in the Unites States and Europe, respectively. The simulations evaluated here consist of 24h cycling forecasts with meteorological initial conditions provided by an analysis, and aerosol and chemical initial conditions provided by the previous forecast. An evaluation of such a simulation with cycle 45R1 using the same configuration and resolution ($T_L 511L60$) as the operational forecasts is presented first. A comparison of the skill scores between standalone and coupled with IFS-CB05 simulations with cycle 45R1 is then made. Finally, the skill scores of simulations with cycle 40R2 and 45R1 are compared, with simulations using similar resolution ($T_L 159L60$) and emissions. Cycle 40R2 was chosen because it was used in the CAMS interim Reanalysis and because the changes between cycle 32R2, described in Morcrette et al. (2009), and cycle 40R2, are limited as far as aerosols are concerned. Because of upgrades in the ECMWF high performance computing facility, it is not possible anymore to run

**Figure 9.** From left to right and top to bottom: 2017 total, sea-salt, dust, OM, BC and sulfate AOD at 550 nm, simulated by IFS-AER CY45R1 in cycling forecast mode.

simulations of the original Cycle 32R2. This is not intended as a full evaluation, which would require a much more thorough validation of the output of IFS-AER, but rather to show that the model performs relatively well for the headline CAMS products.

## 8.1 Summary

Table 12 shows a summary of global and regional scores for AOD at 500 nm and PM for a year of simulation for the four
5 experiments described above. For all regions and for AOD at 500 nm, $PM_{2.5}$ and $PM_{10}$, the RMSE is improved by CY45R1 as compared to CY40R2 at a similar resolution, sometimes by a large margin: global RMSE on AOD at 500 nm is decreased

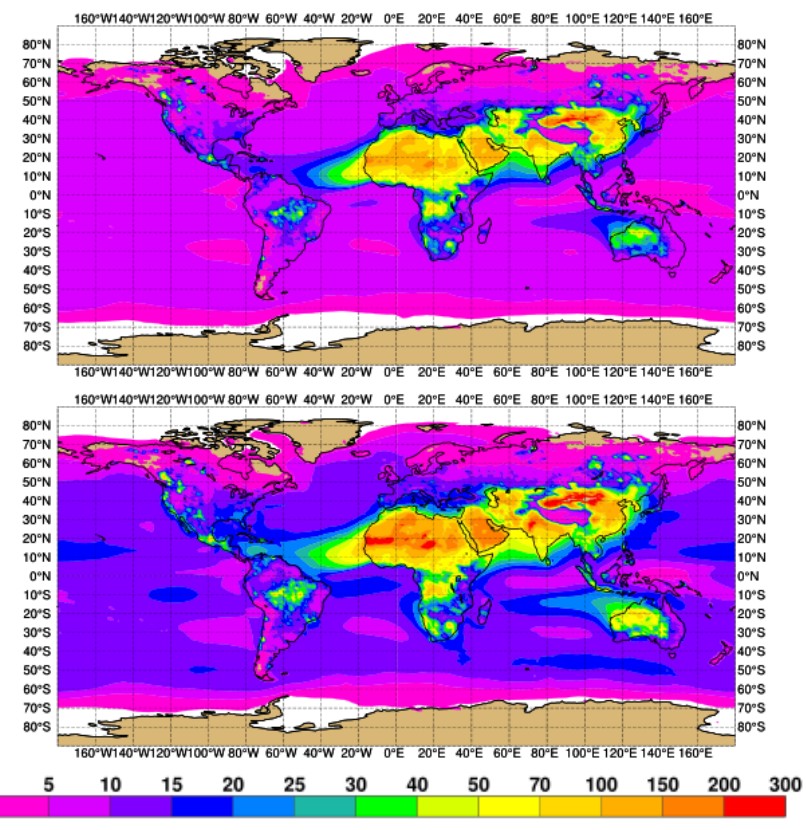

**Figure 10.** Global 2017 near-surface PM$_{2.5}$ (left) and PM$_{10}$ (right) in µg m$^{-3}$ simulated by IFS-AER CY45R1 in cycling forecast mode.

by about 20% . Bias is improved nearly everywhere except over Europe for AOD and North America for PM$_{2.5}$. Interestingly, the CY45R1 simulation using the operational resolution of T$_L$511L60 shows an improved bias for AOD as compared to the CY45R1 simulation at T$_L$159L60, a improvement for most scores for European PM simulations, and a degradation of the scores for North-American PM. Simulation with 45R1 coupled with IFS-CB05 show mixed scores as compared to standalone CY45R1 at a global scale. However, regional AOD scores are notably improved with the coupled simulation, especially over Europe and North America, where RMSE is reduced from 0.11 to 0.093, and where the negative bias over Europe is nearly eliminated.

## 8.2 Evaluation against AERONET

Figures 11 give an indication of how the model compared to AERONET observations for weekly AOD forecasts. The Root Mean Square Error (RMSE) on weekly AOD is in the range of 0.1 to 0.2 on average, while the bias is generally lower than 0.05. The global correlation factor between simulated and observed instantaneous AOD reaches 0.7. There is no important seasonal

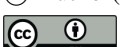



**Table 12.** Average over the 1/5/2016 to 1/5/2017 period of the bias/RMSE of daily AOD at 500nm and PM from the experiments described in this section. AOD observations are from AERONET level 2; European PM observations are from 65 $PM_{2.5}$ and 138 $PM_{10}$ background rural Airbase stations; North American PM observations are from 1006 $PM_{2.5}$ stations and 336 $PM_{10}$ stations.

| Experiment | Global | Europe | N.America | S.America | Africa | SE Asia |
|---|---|---|---|---|---|---|
| AOD CY40R2 ($T_L$159L60) | −0.031 / 0.22 | −0.0071 / 0.14 | −0.015 / 0.13 | −0.043 / 0.12 | 0.028 / 0.20 | −0.15 / 0.36 |
| AOD CY45R1 ($T_L$159L60) | −0.020 / 0.17 | −0.033 / 0.11 | −0.0049 / 0.093 | −0.023 / 0.10 | 0.013 / 0.16 | −0.074 / 0.27 |
| AOD CY45R1 ($T_L$511L60) | −0.012 / 0.18 | −0.037 / 0.11 | 0.0015 / 0.11 | −0.013 / 0.12 | 0.017 / 0.17 | −0.058 / 0.27 |
| AOD CY45R1 coupled ($T_L$511L60) | −0.020 / 0.17 | −0.0021 / 0.093 | 0.0025 / 0.093 | −0.020 / 0.11 | 0.013 / 0.16 | −0.081 / 0.27 |
| $PM_{2.5}$ CY40R2 ($T_L$159L60) | — | 3.2 / 18 | 1.3 / 31 | — | — | — |
| $PM_{2.5}$ CY45R1 ($T_L$159L60) | — | 1.8 / 11 | 4.2 / 18 | — | — | — |
| $PM_{2.5}$ CY45R1 ($T_L$511L60) | — | 0.44 / 10 | 5.1 / 25 | — | — | — |
| $PM_{2.5}$ CY45R1 coupled ($T_L$511L60) | — | 2.9 / 10 | 5.1 / 24.9 | — | — | — |
| $PM_{10}$ CY40R2 ($T_L$159L60) | — | 7.7 / 32 | −11 / 32 | — | — | — |
| $PM_{10}$ CY45R1 ($T_L$159L60) | — | 0.61 / 18 | −5.4 / 30 | — | — | — |
| $PM_{10}$ CY45R1 ($T_L$511L60) | — | −1.3 / 15 | −0.88 / 36 | — | — | — |
| $PM_{10}$ CY45R1 coupled ($T_L$511L60) | — | 2.4 / 15 | −2.7 / 33 | — | — | — |

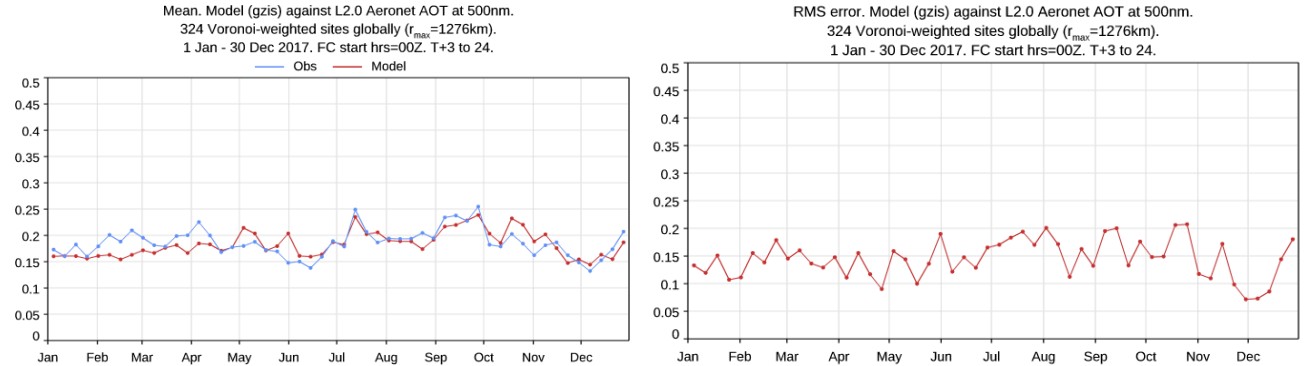

**Figure 11.** Mean value (left) and RMSE (right) of weekly AOD at 500 nm simulated by CY45R1 (red) against global observations from AERONET (blue) in 2017.

cycle for skill scores; however it should be noted that because of the location of the AERONET stations, many important dust and biomass-burning events don't impact these scores.



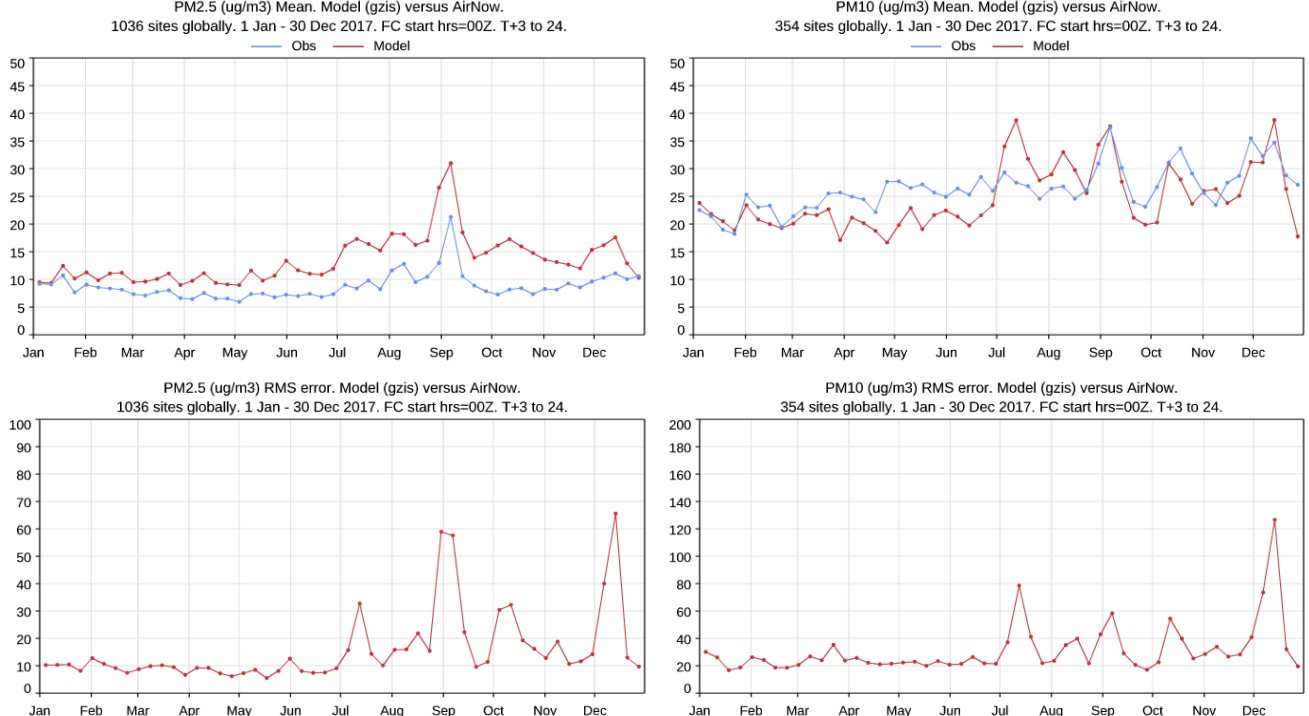

**Figure 12.** Mean value (above) and RMSE (below) of weekly near-surface PM$_{2.5}$ (left) and PM$_{10}$ (right) in $\mu$g m$^{-3}$ simulated by CY45R1 (red) against North American observations from the Airnow network (blue).

## 8.3 Evaluation against PM observations

In this section, simulations of near-surface PM$_{2.5}$ and PM$_{10}$ are evaluated against observations from two regional networks: the Airnow network which gathers observations mostly over the United States and Canada, and the AirBase, the European Air Quality Database operated by the European Environment Agency (EEA), which gathers observations over Europe. As metadata was not available for Airnow observations, data from all sites (1036 for PM$_{2.5}$ and 354 for PM$_{10}$ in 2017) are used, which means that urban, suburban and rural stations are included in the scores. Similarly, the observations may be representative of background pollution or closer to traffic sources. For the Airbase observations, for which site information was available, it was chosen to focus on scores vs background rural stations (62 stations in all). Because of the relatively coarse horizontal resolution of the simulation (about 40 km), urban areas cannot be realistically represented and this will lead to a low bias as compared to observations, caused not from issues in the model or in emissions, but mostly from resolution. Figures 12 and 13 show the result of the evaluation against the Airnow and the AirBase observations.

Observed PM$_{2.5}$ at all Airnow stations show higher values during the summer months, reaching 30 $\mu$g m$^{-3}$ in early September. These high values were caused by a series of extreme fire events: the summer 2017 was saw a very active biomass-burning





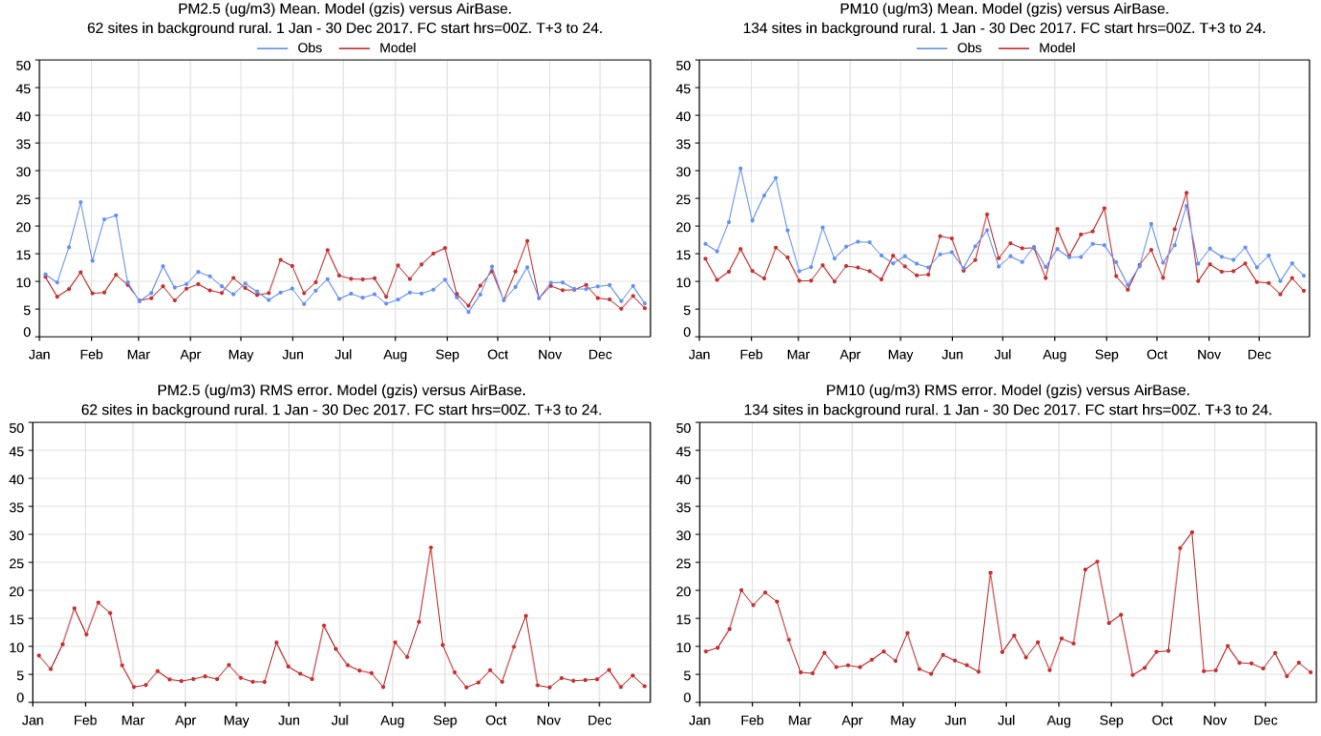

**Figure 13.** Mean value (above) and RMSE (below) of weekly near-surface PM$_{2.5}$ (left) and PM$_{10}$ (right) in $\mu$g m$^{-3}$ simulated by CY45R1 (red) against European observations at background rural stations from the AirBase network (blue).

season in the United States and Canada. IFS-AER generally overestimates PM$_{2.5}$ significantly, by 5 to 10 $\mu$g m$^{-3}$ in general and by more than 10 $\mu$g m$^{-3}$ during fire events. This overestimation is mainly caused by the SOA emissions at surface, which contribute to a large fraction of PM$_{2.5}$, and to the fact that emissions from biomass-burning are released at the surface, which can increase PM$_{2.5}$ forecasts to very high values during fire events.

5    As compared to AirBase observations, PM$_{2.5}$ and PM$_{10}$ are biased low during most of January and February 2017 by up to 10–15 $\mu$g m$^{-3}$. For the rest of the year, the low bias persists for PM$_{2.5}$ but with value generally lower than 5 $\mu$g m$^{-3}$, while for PM$_{10}$ the bias can be negative (in March–April) or positive (in August). The error is on average significantly larger for PM$_{10}$ (between 5 and 15 $\mu$g m$^{-3}$, excluding larger spikes) than for PM$_{2.5}$ (generally below 5 $\mu$g m$^{-3}$, except for spikes). Biomass-burning events are associated with spikes in RMS, which otherwise is generally below 10 $\mu$g m$^{-3}$. The impact of fire events is

10    less marked in PM$_{10}$ observations, especially since dust is a significant component of PM$_{10}$ in many North American stations. PM$_{10}$ simulated by IFS-AER show generally a low bias except during large fire events. RMSE is generally at 20–25 $\mu$g m$^{-3}$ except during large fire events.



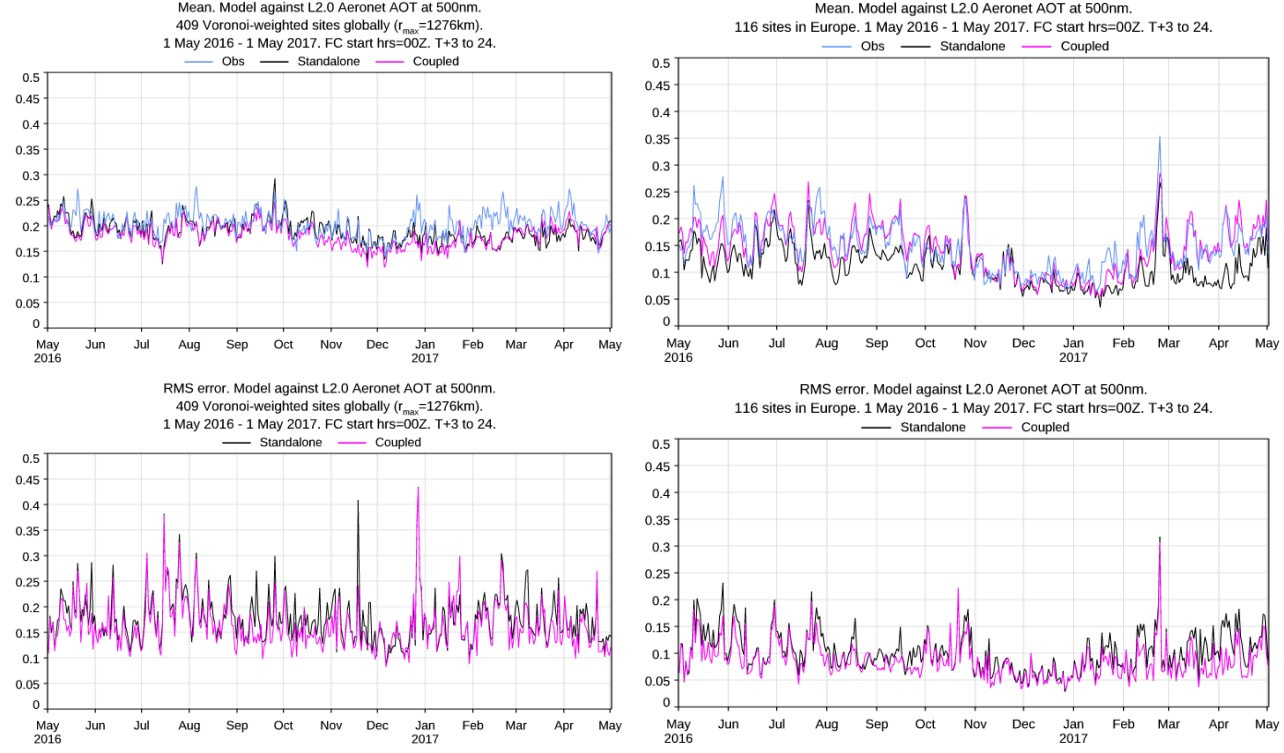

**Figure 14.** Mean value (above) and RMSE (below) of daily AOD at 500 nm simulated by standalone (black) and coupled (violet) IFS-AER cycle 45R1 against global (left) and European (right) observations from AERONET (blue).

## 8.4 Skill of IFS-AER standalone and coupled with chemistry

In this section, the AOD and European PM forecasts of cycle 45R1 IFS-AER in its standalone and coupled to the chemistry configurations are compared. When coupled to the chemistry, sulfate oxidation rates are provided by IFS-CB05, and are generally lower than the rates computed by the simple scheme of IFS-AER. Lower sulfate burden is compensated by contribution from nitrate and ammonium. The simulations were carried out at the resolution used operationally: $T_L511$ with 60 levels. It should be noted that the results presented here are only valid for cycle 45R1: since IFS-CB05 is constantly evolving, both the sulfur dioxide oxidation rates and the concentration of the nitric acid and ammonia precursor gases of nitrate and ammonium can vary from cycle to cycle. A selection of skill scores against observations of Global and European AOD are shown in Figure 14 and against European observations of $PM_{2.5}$ and $PM_{10}$ in Figure 15. The coupled IFS-AER show a slightly degraded global bias, as the lower sulfate is not entirely compensated by nitrate and ammonium. However, the global error is on average lower with the coupled system. Over Europe, the impact of the coupling with IFS-CB05 is much more positive, with a clear decrease in both bias (by up to 0.05) and error (by up to 25%).



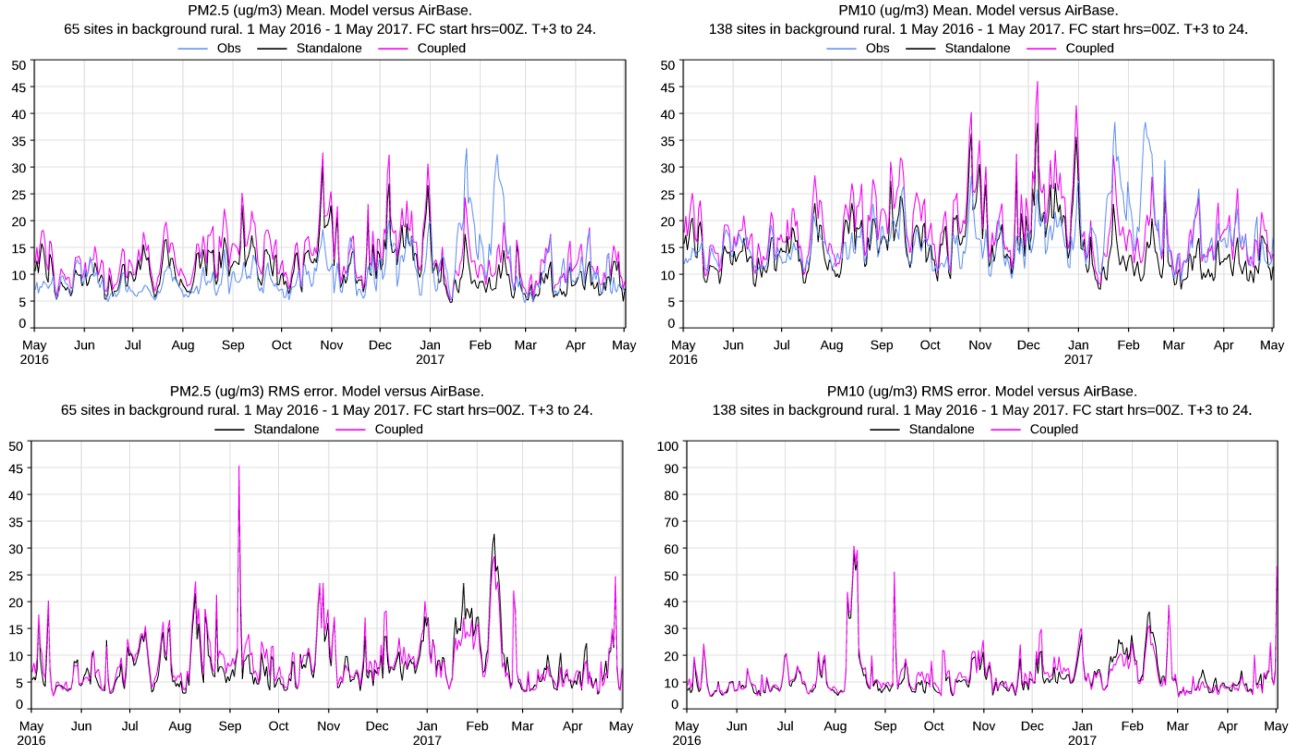

**Figure 15.** Mean value (above) and RMSE (below) of daily near-surface $PM_{2.5}$ (left) and $PM_{10}$ (right) in $\mu g \, m^{-3}$ simulated by standalone (black) and coupled (violet) IFS-AER cycle 45R1 against European observations at rural background stations from AirBase (blue).

The impact on PM forecasts is however generally negative over Europe. The coupling with IFS-CB05 appear to have mostly negative impact on $PM_{2.5}$, with an increase by 2–4 $\mu g \, m^{-3}$ of the positive bias, and a small increase of RMSE. The difference is slightly more important for $PM_{10}$, which is simulated 3–5 $\mu g \, m^{-3}$ higher when running coupled with IFS-CB05. This slightly larger difference could be caused by coarse mode nitrate formed on sea-salt particles, which is taken into account in $PM_{10}$ but contributes only 25% of its mass to $PM_{2.5}$. RMSE of simulated $PM_{10}$ is very close for the two simulations, slightly degraded for the coupled run.

## 8.5 Skill of IFS-AER cycle 40R2 and 45R1

This section aims to give an initial evaluation of the impact of the model upgrades between CY40R2 and CY45R1. To achieve this, the control run (i.e., without data assimilation) of the CAMS interim reanalysis Flemming et al. (2017) is used, and a simulation with cycle 45R1 in standalone mode, using the same fixed emissions (except for the new SOA emissions) and the same horizontal resolution ($T_L 159$) was carried out. The latter simulation covers the May–December 2016 period. Skill scores of daily AOD vs AERONET observations, and of daily near-surface $PM_{2.5}$ and $PM_{10}$ vs observations from the AirBase and



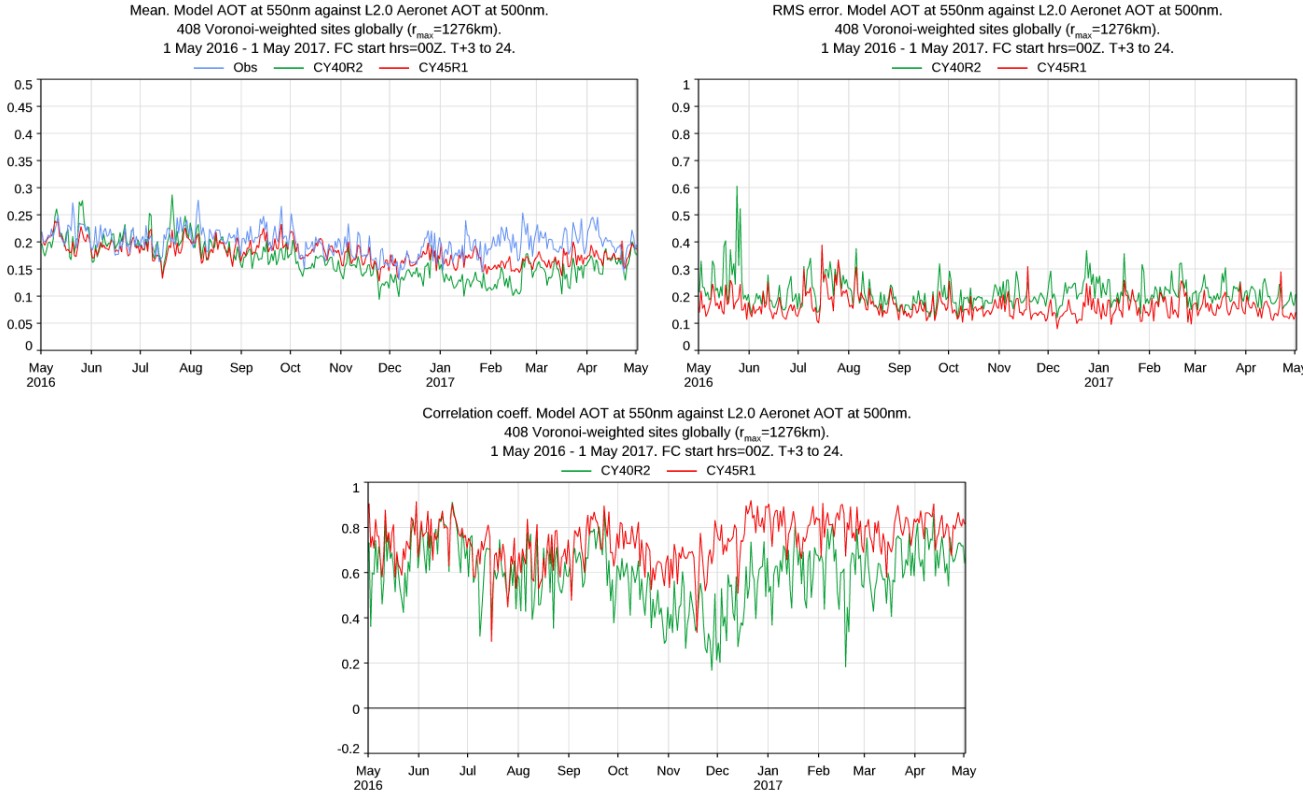

**Figure 16.** Mean value (left), RMSE (right) and spatial correlation (below) of daily AOD at 500-550 nm simulated by CY40R2 (green) and CY45R1 (red) against global observations from AERONET (blue).

Airnow networks were then computed and are shown in Figures 16, 17 and 18. As compared to CY40R2, CY45R1 brings an important improvement in the skill of AOD forecasts, as measured by bias, RMSE and correlation. The bias is generally reduced to under 0.05 for CY45R1. The global RMSE is significantly reduced, and is sometimes more than halved by CY45R1 as compared to CY40R2. The spatial correlation is also generally higher with CY45R1 as compared to CY40R2.

5    Skill scores of the global PM simulations are also generally improved, except for $PM_{2.5}$ over North America, for which the bias and error are both degraded by CY45R1 as compared to CY40R2. The correlation is nonetheless improved. This degradation is caused by the new source of SOA, which is associated with high PM values over North America and an unrealistic diurnal cycle. $PM_{10}$ is generally underestimated over North America, but less so with CY45R1. This is associated with a lower error and a much higher correlation of simulations with observations. PM forecasts over Europe are significantly improved by

10   CY45R1 as compared to CY40R2. The improvement is especially marked for $PM_{10}$, for which the RMSE is divided by more than three.



**Figure 17.** Mean value (top), RMSE (middle) and spatial correlation (bottom) of daily near-surface $PM_{2.5}$ (left) and $PM_{10}$ (right) in $\mu g\,m^{-3}$ simulated by CY40R2 (green) and CY45R1 (red) against observations in the United States from the Airnow network (blue).

## 9 Conclusions

IFS-AER is a simple and low-cost scheme that aims to represent the major atmospheric aerosol species. Because of this simplicity, many processes such as internal mixing, coagulation and nucleation are not explicitly represented. Despite this, IFS-AER achieves a reasonable skill in forecasting AOD and PM. The improvement in skill for the headline products of the



**Figure 18.** Mean value (top), RMSE (middle) and spatial correlation (bottom) of daily near-surface PM$_{2.5}$ (left) and PM$_{10}$ (right) in $\mu$g m$^{-3}$ simulated by CY40R2 (green) and CY45R1 (red) against observations of background rural stations in Europe from the Airbase network (blue).

operational CAMS system, AOD and PM, were improved considerably (except for PM$_{2.5}$ over North America) from cycle 40R2 to 45R1. The tendency has been towards a system that is more integrated with the other components of the IFS and particularly atmospheric chemistry, with the possibility to run with integrated sulfur and ammonia cycles. The coupling with IFS-CB05



generally improves the skill of the forecasts of IFS-AER, especially over Europe. This also makes the skill of IFS-AER more dependent on the evolution of IFS-CB05.

As compared to the original implementation as described in Morcrette et al. (2009), many components of IFS-AER have been reviewed, and two new species (nitrate and ammonium) were added. Some parameterizations on the other hand such as

dust production and wet scavenging are essentially similar in cycle 45R1 as in M09. Work is planned on these two items which should undergo a major upgrade in the future cycle 46R1. The treatment of secondary organics is also very simplistic and lead to unrealistic diurnal cycle in PM simulations regionally. Future upgrades should include the possibility to use SOA production rates provided by the chemistry scheme. Also a coupling to alternative chemistry precursor schemes will be established. Also, the use of GFAS injection heights to release smoke constituents is being considered.

*Code availability.*

Model codes developed at ECMWF are the intellectual property of ECMWF and its member states, and therefore the IFS code is not publicly available. ECMWF member-state weather services and their approved partners will get access granted. Access to a reduced version of the IFS code not including the aerosol component may be obtained from ECMWF under an OpenIFS licence.

*Author contributions.*  SR drafter the paper; SR, ZK and JF maintain and carry out developments on IFS-AER; OB developed the LMD-LOA model from which IFS-AER was adapted; JJM carried out developments on IFS-AER up to cycle 38R2, PN and MM contributed to the desert dust and sea-salt parameterizations, ABo to the optical properties part, VH to the coupling to IFS-CB05. MA, ABe, RE, VHP contributed to drafting and revising this article.

*Acknowledgements.*  This work is supported by the Copernicus Atmospheric Monitoring Services (CAMS) programme managed by ECMWF

on behalf of the European Commission. The authors would also like to thank NASA, the European Environment Agency, the Airnow network and the United States Environmental Protection Agency (EPA) for making the MODIS, AirBase, Airnow and CASTNET data publicly available.



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
