# Peer review of "Description and evaluation of the tropospheric aerosol scheme in the Integrated Forecasting System (IFS-AER, cycle 45R1) of ECMWF"

_Geoscientific Model Development, 2019_

## Referee Comment (RC1) · Anonymous Referee #1 · 4 Jul 2019

**General comments**

This work describes the aerosol module IFS-AER used within the Integrated Forecasting System (IFS) from ECMWF, with changes described up until cycle 45R1. Since this model is used as part of the Copernicus Atmospheric Monitoring Service (CAMS) to forecast and reanalyze atmospheric composition, the model description and evaluation is of clear importance to the scientific community. The paper provides details behind the parameterizations used in the aerosol module, and compares model results to observations of particulate matter and aerosol optical depth (AOD). The improvements added in cycle 45R1 lead to better agreement with the observations compared

with the model version from cycle 40R2. The paper presents a thorough description of the IFS-AER model, and would be suitable for publication in GMD. However, I do have some comments regarding the clarity of the manuscript and the comparison with the observations. Specifically, the motivation behind the code revisions/additions that were chosen could be more clearly discussed in the introduction of the paper. As well, the authors should try to identify which modification was responsible for the improvement between cycles 40R2 and 45R1. These changes would make it easier for the reader to follow the logic of the paper.

**Specific comments**

P1L13: The code cycle 40R2 was not mentioned before in the abstract, and therefore it is unclear why the version 45R1 is compared to 40R2, and not 32R2.

P5L1: When running coupled with IFS-CB05, $SO_2$ no longer is a prognostic variable in IFS-AER. Why there are more prognostic variables in the coupled version?

Why are certain code changes not included in operational cycle? It is clear to me that certain options would be computationally expensive (for example: P4L13-stratospheric chemistry; P4L29 - coupling with IFS-CB05) but I do not understand why others are non-operational (e.g. P5L17 - height of emissions for biomass burning and $SO_2$ from volcanoes, especially considering statement on P6L20-24). Perhaps a general statement could be added for why certain code segments are operational or non-operational.

P6L7: "remarkably consistent between the three datasets": I would say two datasets, since only the anthropogenic emission inventories are compared.

P6L10-14: In cycle 45R1, do the scaling factors come from Kaiser et al. (2012) or Rémy et al. (2017)?

P7L21 and Figure 1: Are the units $\mu$g m$^{-2}$ s$^{-1}$ correct? Figure caption reads kg m$^{-2}$ s$^{-1}$, and also the there are higher values shown than 0.25.

P10L3: It is mentioned that G14 is closer to the AOD observations, but this is not immediately clear to me from Figure 3, where for some months (e.g. May-June, Oct-Nov) it seems like M86 is closer to the AOD observations. Can you back this statement up with any quantitative measures (e.g overall bias?)

P13L8: "Maps of probability of occurrence of observed AOD by MODIS dust AOD above different thresholds"- this phrase is not clear to me.

P14L1: I do not understand how and why the anthropogenic $SO_2$ emissions are divided into these categories. Does "high"/"low" refer to altitude or the magnitude of emissions?

P14L28: What is the rationale behind the change in $SO_2$ lifetimes in the later cycles?

P21L7: In Table 6 the deposition velocities of sulfate aerosol also differ between continents/oceans

P23L20-21: This pattern of ZH01 showing lower deposition velocities for fine particles and higher deposition velocities for coarse particles is not clear from Table 7. For example, fine mode sea salt is indeed slower in ZH01, but fine mode dust is faster.

Why is the particle radius not considered in the wet deposition scheme? Aerosol radius can play a role in both in-cloud and below-cloud removal processes (e.g. Seinfeld and Pandis, 2006, Chapter 20). I am curious why a radius dependence was included for dry deposition but not for wet deposition.

P28L15: Is there a reference/derivation for the coefficients in the PM formulae? It can play an important role in the model evaluation, as mentioned in P39L3-5.

Table 9: Are these listed size distribution parameters used also for the dry deposition/sedimentation schemes? For sea salt and dust the size bin limits were mentioned previously in the text, but for OM/BC/Sulfate/Nitrate this is is the first mention of the assumed size distribution. Perhaps the assumed size distributions for these particulate species can also be mentioned in Section 2 or 3, when the different particle species are introduced.

Section 6 and Table 10 clarified some confusion that I experienced earlier in the paper, since they introduce the discussed code versions and highlight updates in the new CY45R1 code. I wonder whether it would be better to move this section and table earlier in the paper, since the reader would then be able to refer to this table when the code versions are mentioned.

P31L10-11: I don't fully understand the logic why this would affect the wet/dry deposition ratio of BC compared to sulfate. It could also be related to the choice of deposition parametrizations and the size distributions of particle types.

Model evaluation metrics: The RMSE and bias metrics will be biased towards detecting deviations between the model and observations when AOD/PM are high, since AOD and surface PM vary by a few orders of magnitude. The model's skill at matching lower AOD/PM locations could be overlooked using RMSE/bias. Did you consider using normalized metrics for the model evaluation, for example mean fractional bias? I think normalization would anyways help with comparing the bias and RMSE between regions in Table 12, since the mean AOD/PM also varies between regions.

P40L1-5: What is the reason for the improvement in model skill between CY45R1 and CY40R2? Can you point to a change in Table 10 that is responsible?

P40L6-8: Does the seasonal cycle of the North American bias in PM2.5 provide evidence for the hypothesis that the SOA emissions are responsible? Since this bias is larger for May-Nov?

P43L1: This statement could be made more accurate, to say that the coupling improves the *error* of the forecasts, especially over Europe. The *bias* seems to be worse everywhere except Europe and Africa in the coupled version.

P43L8-9: The final two concluding sentences could be made stronger, since these ideas were not discussed very much in the model evaluation. Perhaps these future upgrades could be linked to deficiencies that were found in the results?

**Technical corrections**

P2L10: space missing before (NCEP)

P4L18: "the" missing before prognostic variable

P4L26: Two components are considered of organic matter and black carbon, hydrophilic and hydrophobic fractions, with . . .

P7L29: fluxes of sensible

P9L19: two schemes

P9L20-21: is also shown to compare

Figure 5: Caption should specify that super-coarse dust is shown.

P14L23: CY is used before the cycle name for the first time. This should be made consistent throughout the paper.

P15L1: a diurnal cycle and a simple dependency on temperature

P15L23: "important decrease"- significant decrease -> but also could specify where - in North America and Europe $SO_2$ emissions have decreased strongly, in East Asia they have increased since 2000 (e.g. Hoesly et al., 2018)

P16L11: the equation numbering restarts from 1 here. Reactions should be numbered R1, R2, etc.

P17L24: fraction of airborne calcite in coarse

P23L10: the first comma should be a period?

P23L19: provides

P23L30: ZH01, notably

P23L31: will be addressed

P25Eq38: j-1 should be in superscript in numerator

Table 11 caption: in parentheses, not in hypens

P31L9: with

P32L5: combines

Figure 10 caption: should read (top) / (bottom) not (left) / (right)

P33L5: acronym RMSE should be defined the first time it is mentioned

Figure 12: titles of plot refer to global sites, but the caption and text refer to North American sites

P38L4: by a higher contribution from nitrate and ammonium

**References**

Hoesly, R. M., Smith, S. J., Feng, L., Klimont, Z., Janssens-Maenhout, G., Pitkanen, T., Seibert, J. J., Vu, L., Andres, R. J., and Bolt, R. M.: Historical (1750–2014) anthropogenic emissions of reactive gases and aerosols from the Community Emissions Data System (CEDS), Geoscientific Model Development, 11, 369–408, 2018.

Seinfeld, J. H. and Pandis, S.N: Wet deposition, in: Atmospheric chemistry and physics: from air pollution to climate change, John Wiley Sons, 932–979, 2006.

---

## Referee Comment (RC2) · Anonymous Referee #2 · 6 Aug 2019

The paper describes the latest developments of the AER-IFS global aerosol model development by ECMWF in the framework of the Copernicus Atmosphere Monitoring Service. The model description is very clear and comprehensive and accompanied by various detailed evaluation experiments. Therefore, I support publication in GMD once the following minor points are addressed.

The code is however not available publicly, which is usually an issue for GMD publication which I leave up to the editors to decide. I note that I did not request access to the code to perform the review.

In the introduction and perspective, I am surprised that the added value of implement-

ing coupled aerosol model to improve the weather forecasts is not mentioned in the context of ECMWF activities. This may be due to the low priority given to aerosol-cloud interactions (P3L29), but more explanation should be given in the text on this choice to understand if the impact is thought to be marginal, or if the work is planned in the future.

My only general concern regards the lack of context about state of the art in global aerosol modelling, as it would be useful to explain in more details where the AER-IFS lies in comparison with other major similar models.

Specific comments:

P1L17: these mortality numbers are only for ambient air pollution (excluding indoor air).

P2L14: a link to the ICAP-MME service should be provided as a web search did not allow me to reach those alternative global aerosol products.

P3L29: why mentioning only the impact of aerosols on surface temperature as photolysis rates are also sensitive as acknowledged P5L5.

P4L13: what would be the implication of including stratospheric chemistry for AER-IFS given the importance of heterogeneous processes in the stratosphere?

P4L30: is $SO_2$ or rather sulphate aerosol production rate (as stated P5L9) provided by IFS to AER-IFS when run in coupled mode?

P5L1-8: why not mentioning here the new CAMS-GLO-AP emissions that should be included in the global forecast soon (if not already)?

P6L8-19: the scaling factor for biomass burning emissions seems very ad-hoc. How is it handled for instance by regional models? What are the perspectives to represent explicitly condensation to cope with this issue?

P5L19: suggest adding SOA in the title of the section. Given the importance of SOA

Interactive
comment

acknowledged P7L16. More information on the state of the art on SOA modelling in global aerosol models should be provided.

P9L26: typo: M86 instead of G86

P10L3: quantitative scores (RMSE, correlation and bias) should also be provided as it seems that the new formulation sometimes leads to high overestimations of low AOD levels.

P13L5: the figure legend says total emission are presented whereas only super-coarse are mentioned in the text.

P21L3: where was it shown that differences where small? Is it only true locally or just on average?

P24: use the same unit in the table & figure

P26L13: I find it confusing to present here ageing, while hygroscopic growth is presented in 3.6

P28L7: what is referred to as CAMS mineral dust model? AER-IFS?

P28L15-24: the notations are not defined (SS1, SS2, etc.)

P29 Table 9: what is referred to as CAMS model? AER-IFS?

P29L11: confirm if the upgrade to 137 levels has been completed by now

P30 Table 10: what is referred to as IFS versions? AER-IFS?

P31L9: typo "wityh"

P32L24: typo: "Unites"

P37L10: the sentence is inconsistent. Fire and dusts are mixed, and my understanding is that Europe was discussed in that paragraph.

P43L1: whereas coupling improves AOD over Europe, it is not the case for PM and

should be reminded in the conclusion.

---

## Referee Comment (RC3) · Anonymous Referee #3 · 9 Aug 2019

The paper presents a fairly comprehensive description of the aerosol scheme at use in the IFS-AER modeling system, some presentation of the system budget, and a basic level of validation against AERONET AOD observations and surface observations. The paper is generally well organized and complete. I have two substantive comments on the paper. The first is that many of the figures are not really publication quality in that they are titled/labeled with a lot of jargon that means something to the writers but not to anyone else. I noted those occasions below and suggest redoing those figures to remove the title text (or else writing it in a way that is sensible to the general reader). My second substantive comment is that there is a great deal of text written that does not pertain specifically to the IFS-AER operational configuration. I understand this from

the point of view of explaining the possible permutations of the system, but it is hard in the end to reconstruct in my head the actual configuration used in the operational system, which seems to be the point. Table 10 is not sufficiently comprehensive in this sense. I suggest to please add a table, and maybe do it up front, that writes down the details of the IFS-AER operational configuration as a reference (chemistry or lack of, BB emissions altitude, and so forth, sea salt emission scheme, etc,.). Otherwise I was getting lost in the details.

Otherwise I have mainly minor comments listed below.

Page 3, line 28: implication of using online ARI Page 4, line 1: A point of clarification: is IFS "the" ECMWF forecasting system? And if so, isn't it run at a considerably higher than global 40 km resolution? So is IFS-AER run run at the same resolution as the operational NWP system or not? Page 4, line 32: I don't follow the counting of species, please clarify. 12 species when running standalone I guess excludes the "optional" nitrate. When running coupled do you move the SO2 to the chemistry but still count the sulfate species? Are you now counting the nitrate? Did you not note an ammonium tracer, implied in section 2.4? Assuming all of that my count is 3 dust + 3 sea salt + 4 carbon + 1 sulfate + 2 nitrate + 1 ammonium = 14. Please clarify. And what is the condition for running the optional nitrate? Here and elsewhere in the paper it is useful to state explicitly what is in IFS-AER cycle 45R1 which is the main point of the paper. The operational configuration would seem to be IFS-AER standalone. Page 6, line 1: Any citation for the partitioning of hydrophobic and hydrophilic carbon? Page 6, line 18: I think the reason given here for the scaling factors is not credible. What you are really doing here is using a scaling factor to tune the emissions to give best agreement with simulated AOD, and that in fact implicates all of the model physics (especially sink processes) as well as whatever optical parameters are used to go from dry aerosol mass to hydrated extinction efficiency. So what you are doing here is finding the model "perceived emissions" needed to converge on observed AOD. If it were simply a gas to particle factor it is not clear why this would be model dependent since you are not

doing that chemistry. I suggest rewriting this sentence in the spirit of the "perceived" emissions as I describe above. Page 7, line 21: Figure 1 does not show anything about seasonality. Page 8: there is something missing (like a "x") before the 10ˆ-6 in equation 3. Page 9, line 19: do you mean "two schemes"? Page 9, line 20: "mode" instead of "model" Page 9, line 21: two schemes? Why do you write "three"? Section 3.2: The presentation suggests there are no global tuning factors. Experience tells me this is usually not the case, and that models at least need a scaling factor as a function of spatial resolution. I'm also skeptical of the utility of comparing the predicted emissions to the G14 estimate. I won't dismiss G14's estimate of emissions, I haven't read the paper, but generally my sense is that non-inventory emissions are not usually observationally constrained sufficiently and so (like for biomass burning and dust later) some tuning is done to get emissions that in your model improve agreement with AOD, which is observed. Figure 3: The title is best cropped from this figure as it uses jargon not discussed in the paper ("gzis", etc.) Page 12, line 8: Please explain better the dynamic nature of the lifting threshold velocity. What does "emission capacity" mean? Can you please provide an equation? page 12, line 12: I think you mean equations 10/11 and not 13. Page 21, line 3: Please explain the statement about the connection to the vertical turbulent flux scheme. Are emissions just added to the model grid boxes (flux*time = mass changed to cell)? What does it mean surface fluxes are unchanged here? Page 23, line 31: How have these been addressed? Page 25, line 15: Do you mean the settling velocity is horizontal invariant? Constant in space is too broad, because you do have vertical variability (don't you?) Page 26, line 1: What is the origin of the numbers in Table 8 for Di? Page 26, line 10: What are the "R" values in equation 42? Page 27, line 24: Please define OPAC and provide citation. This is first time you use the term OPAC. Page 31, Table 11: You mean parentheses and not hyphen at the end of the caption. Page 23, figure 10: you mean top and bottom in the caption Page 32, Section 8: Is aerosol data assimilation invoked in this analysis? Page 35, Figure 11: suggest again removing the title from the figure as it is inside jargon Page 36, figure 12: suggest again removing the title from the figure as it is inside jargon

Page 36, line 12: Observed values of PM2.5 seem to reach 20 ug m-3, not 30. Page 37, Figure 13: suggest again removing the title from the figure as it is inside jargon Page 37, line 5: I am confused about the text here. i don't see a persistent low bias in the model throughout the year ( red line = model higher than blue line = observations generally May - November) Page 38, Figure 14: same comment about figure title Page 39, Figure 15: same Page 40, Figure 16: same Page 41, Figure 17: same Page 42, Figure 18: same

---

## Author Response (AR1)

**Answer to reviews of GMD-2019-142 « Description and evaluation of the tropospheric aerosol scheme in the Integrated Forecasting System (IFS-AER, cycle 45R1) of ECMWF "**

First, we wish to thank the reviewers for their comments, which raised many good points and helped improve the quality of the paper.

**Anonymous Referee #1**

**General comment**

*The motivation behind the code revisions/additions that were chosen could be more clearly discussed in the introduction of the paper. As well, the authors should try to identify which modification was responsible for the improvement between cycles 40R2 and 45R1. These changes would make it easier for the reader to follow the logic of the paper.*

This is a good point, thank you. The code revisions that are integrated into the operational version of IFS-AER must satisfy the two conditions (one qualitative, one quantitative) that they bring the model closer to "physical" reality (ie that more processes and/or species are represented), and that they improve the skill scores vs observations. A motivation behind several developments is to try to reduce the low bias of the AOD simulations against AERONET and MODIS. This was the main objective of the new SOA source and of the new sea-salt scheme. Debiasing the model vs MODIS allows the assimilation step to be more efficient in reducing error at analysis. A paragraph was added in the introduction to detail this, and in section 8.5 to highlight the developments that were responsible for most of the improvement between cycles 40R2 and 45R1.

**Specific comments**

*P1L13: The code cycle 40R2 was not mentioned before in the abstract, and therefore it is unclear why the version 45R1 is compared to 40R2, and not 32R2.*

Because it is a very old version, cycle 32R2 is now quite impossible to use with the ECMWF HPC. 40R2 was used because simulations were available as this cycle is the one used for CAMSiRA. Apart from the changes in the meteorological part of the IFS from cycle 32R2 to 40R2, there has been relatively few changes of IFS-AER from cycle 32R2 to 40R2. This is detailed in the first paragraph of section 8 "Evaluation".

*P5L1: When running coupled with IFS-CB05, $SO_2$ no longer is a prognostic variable in IFS-AER. Why there are more prognostic variables in the coupled version?*

This was taking into account nitrates (two variables) and ammonium; the sentence was modified to clarify this.

*Why are certain code changes not included in operational cycle? It is clear to me that certain options would be computationally expensive (for example: P4L13-stratospheric chemistry; P4L29 - coupling with IFS-CB05) but I do not understand why others are non-operational (e.g.*

*P5L17 - height of emissions for biomass burning and SO2 from volcanoes, especially considering statement on P6L20-24). Perhaps a general statement could be added for why certain code segments are operational or nonoperational.*

Biomass burning injection heights are not used operationally in 45R1 (they are now in 46R1, but this is outside of the scope of this paper), in order to keep consistency between the aerosol and trace gases emissions: while the use of injection heights was shown to be mostly positive for aerosol simulations, it was not evaluated yet for simulations of trace gases (CO in particular).

*P6L7: "remarkably consistent between the three datasets": I would say two datasets, since only the anthropogenic emission inventories are compared.*

This is corrected, thank you.

*P6L10-14: In cycle 45R1, do the scaling factors come from Kaiser et al. (2012) or Rémy et al. (2017)?*

The scaling factors from Kaiser et al. (2012), ie a constant value of 3.4, is used. This has been clarified in the text.

*P7L21 and Figure 1: Are the units correct? Figure caption reads kg m2s1, and also the there are higher values shown than 0.25.*

The units of Figure 1 were wrong and have been corrected, thank you. Also, the cap of 0.25 µg/m²/s was not used in the data used to generate the emissions plot; this has also been corrected.

*P10L3: It is mentioned that G14 is closer to the AOD observations, but this is not immediately clear to me from Figure 3, where for some months (e.g. May-June, Oct- Nov) it seems like M86 is closer to the AOD observations. Can you back this statement up with any quantitative measures (e.g overall bias?)*

Figure 2 was modified to add the global bias against MODIS/Aqua AOD (collection 6.1), which shows clearly how a negative bias over most of oceans with M86 is reduced with G14. I agree that against AERONET the signal is less clear; the statement was modified.

*P13L8: "Maps of probability of occurrence of observed AOD by MODIS dust AOD above different thresholds"- this phrase is not clear to me.*

Indeed this was mashed up. This paragraph was rewritten.

*P14L1: I do not understand how and why the anthropogenic SO2 emissions are divided into these categories. Does "high"/"low" refer to altitude or the magnitude of emissions?*

This is an option (added in the text). "high/low" refers to the altitude of emissions: this option (not commonly used) is meant to distinguish between SO2 emissions that are released at the surface and those that are emitted higher up (ship emissions for example). The 80/20 ratio between "high" and "low" emissions is arbitrary and should be refined using emissions by sectors. The text has been modified to clarify this option.

*P14L28: What is the rationale behind the change in SO2 lifetimes in the later cycles?*

Conversion of SO2 was made faster (and SO2 dry deposition was introduced) in order to shorten the lifetime of sulphate aerosol and also its burden, which was found to be much too high in CAMSiRA (using data assimilation). Thanks to these changes, sulphate aerosol is no more too dominant in CAMSRA.

*P21L7: In Table 6 the deposition velocities of sulfate aerosol also differ between continents/ oceans*

The text was corrected; thank you for spotting this error.

*P23L20-21: This pattern of ZH01 showing lower deposition velocities for fine particles and higher deposition velocities for coarse particles is not clear from Table 7. For example, fine mode sea salt is indeed slower in ZH01, but fine mode dust is faster.*

This is correct: with the ZH01 scheme, dry deposition velocity as a function of particle diameter decreases from 0.01 to 1 micron and increases from 1 to 10 micron. The sentence was corrected and a sentence was added to explain this.

*Why is the particle radius not considered in the wet deposition scheme? Aerosol radius can play a role in both in-cloud and below-cloud removal processes (e.g. Seinfeld and Pandis, 2006, Chapter 20). I am curious why a radius dependence was included for dry deposition but not for wet deposition.*

In Seinfeld and Pandis 2006, the process that is most dependent on particle size is below cloud scavenging, which, in our simulations, is usually dominated by in-cloud scavenging. In contrast to dry deposition, wet deposition has not been updated in cycle 45R1 IFS-AER as compared what was described in Morcrette et al. (2009). This is beyond the scope of this manuscript, but below cloud wet deposition has indeed been updated in cycle 46R1, with a better representation of the impact of particle size.

*P28L15: Is there a reference/derivation for the coefficients in the PM formulae? It can play an important role in the model evaluation, as mentioned in P39L3-5.*

The coefficients in the PM formulae were computed using the assumed size distribution; this was added in the text.

*Table 9: Are these listed size distribution parameters used also for the dry deposition/ sedimentation schemes? For sea salt and dust the size bin limits were mentioned previously in the text, but for OM/BC/Sulfate/Nitrate this is is the first mention of the assumed size distribution. Perhaps the assumed size distributions for these particulate species can also be mentioned in Section 2 or 3, when the different particle species are introduced.*

The sedimentation velocities for super-coarse sea-salt aerosol and dust (the other species are not sedimented) have been computed with these size distribution parameters. For dry deposition with the ZH01 scheme, the Mass Median Diameter (MMD) was used.

Thanks for the useful suggestion: a new table was added in section 2.3 (main characteristics of IFS-AER) with these information. Table 9 was correspondingly reduced to the refractive indexes used.

*Section 6 and Table 10 clarified some confusion that I experienced earlier in the paper, since they introduce the discussed code versions and highlight updates in the new CY45R1 code. I wonder whether it would be better to move this section and table earlier in the paper, since the reader would then be able to refer to this table when the code versions are mentioned.*

Again, thanks for the suggestion; section 6 has been moved to a new section 3.

*P31L10-11: I don't fully understand the logic why this would affect the wet/dry deposition ratio of BC compared to sulfate. It could also be related to the choice of deposition parametrizations and the size distributions of particle types.*

Since this possible explanation is a pure guess (as we don't have access to the stratiform/convective distribution of precipitation in the simulations presented in Croft et al. 2014), this part has been removed. The underlying idea was that, in IFS-AER simulations at least, BC is relatively more abundant in regions where convective precipitation are either dominant or important, while sulphate is abundant in regions where stratiform precipitation are usually dominant. Yes, the size distribution certainly plays a role, particularly in GEOS-CHEM wet deposition.

*Model evaluation metrics: The RMSE and bias metrics will be biased towards detecting deviations between the model and observations when AOD/PM are high, since AOD and surface PM vary by a few orders of magnitude. The model's skill at matching lower AOD/PM locations could be overlooked using RMSE/bias. Did you consider using normalized metrics for the model evaluation, for example mean fractional bias? I think normalization would anyways help with comparing the bias and RMSE between regions in Table 12, since the mean AOD/PM also varies between regions.*

Absolutely; in routine evaluations we use Modified Normalized Mean Bias (MNMB) and Fractional Gross Error (FGE), in which lower AOD/PM locations have a much higher weight than bias and RMSE. There was a lot of hesitation between the two options (bias/RMSE and MNMB/FGE); the bias/RMSE plots have now all been replaced with MNMB/FGE plots, as well as the data in Table 14.

*P40L1-5: What is the reason for the improvement in model skill between CY45R1 and CY40R2? Can you point to a change in Table 10 that is responsible?*

A paragraph was added in this subsection: The development that had the most impact on skill scores against AERONET is the implementation of a new SOA source in cycle 43R1, which led to a significant improvement of both bias and RMSE. Using the ZH01 dry deposition scheme also improved scores. The new G14 sea-salt aerosol scheme had little impact on AOD scores against AERONET or on PM skill scores, but improved notable the bias and RMSE versus MODIS AOD

*P40L6-8: Does the seasonal cycle of the North American bias in PM2.5 provide evidence for the hypothesis that the SOA emissions are responsible? Since this bias is larger for May-Nov?*

This is a problem the whole year long. The evidence comes from looking at PM diurnal cycle (not shown), which shows very high values at night when SOA is emitted and not vertically diffused, and from comparisons with simulations without the new SOA source.

*P43L1: This statement could be made more accurate, to say that the coupling improves the error of the forecasts, especially over Europe. The bias seems to be worse everywhere except Europe and Africa in the coupled version.*

This is a good suggestion, thank you.

*P43L8-9: The final two concluding sentences could be made stronger, since these ideas were not discussed very much in the model evaluation. Perhaps these future upgrades could be linked to deficiencies that were found in the results?*

Yes; actually since cycle 46R1 is now out, the new upgrades are actually known and consist in a upgrades of the dust emission scheme, of wet deposition and the activation of the use of biomass burning injection height. IFS-AER now runs coupled with the chemistry in the operational context. The conclusion was modified to reflect this.

**Technical corrections**

*P2L10: space missing before (NCEP)*

*P4L18: "the" missing before prognostic variable*

*P4L26: Two components are considered of organic matter and black carbon, hydrophilic and hydrophobic fractions, with : : :*

*P7L29: fluxes of sensible*

*P9L19: two schemes*

*P9L20-21: is also shown to compare*

*Figure 5: Caption should specify that super-coarse dust is shown.*

*P14L23: CY is used before the cycle name for the first time. This should be made consistent throughout the paper.*

*P15L1: a diurnal cycle and a simple dependency on temperature*

*P15L23: "important decrease"- significant decrease -> but also could specify where - in North America and Europe SO2 emissions have decreased strongly, in East Asia they have increased since 2000 (e.g. Hoesly et al., 2018)*

This sentence just means to compare sulphate burden (and AOD) for the same period between the CAMS interim reanalysis and the CAMS reanalysis: for the latter sulphate burden is much lower, which corrected a clear positive bias in CAMSiRA. This was detailed in an additional paragraph in the same section.

*P16L11: the equation numbering restarts from 1 here. Reactions should be numbered R1, R2, etc.*

*P17L24: fraction of airborne calcite in coarse*

*P23L10: the first comma should be a period?*

*P23L19: provides*

*P23L30: ZH01, notably*

*P23L31: will be addressed*

*P25Eq38: j-1 should be in superscript in numerator*

*Table 11 caption: in parentheses, not in hypens*

*P31L9: with*

*P32L5: combines*

*Figure 10 caption: should read (top) / (bottom) not (left) / (right)*

*P33L5: acronym RMSE should be defined the first time it is mentioned*

*Figure 12: titles of plot refer to global sites, but the caption and text refer to North American sites*

*P38L4: by a higher contribution from nitrate and ammonium*

All corrected, many thanks for the careful checking!

**Anonymous Referee #2**

**General comment:**

*"In the introduction and perspective, I am surprised that the added value of implementing coupled aerosol model to improve the weather forecasts is not mentioned in the context of ECMWF activities. This may be due to the low priority given to aerosol-cloud interactions (P3L29), but more explanation should be given in the text on this choice to understand if the impact is thought to be marginal, or if the work is planned in the future."*

Two paragraphs were added on this subject in the introduction, mentioning in particular how IFS-AER was used in the context of sub-seasonal forecasts at ECMWF (Benedetti et al. 2018).

*"My only general concern regards the lack of context about state of the art in global aerosol modelling, as it would be useful to explain in more details where the AER-IFS lies in comparison with other major similar models."*

A paragraph was added in the introduction on the subject. Also, the budgets of IFS-AER are compared, as much as possible, with those of GEOS-CHEM as reported in Croft et al. (2014).

**Specific comments:**

*P1L17: these mortality numbers are only for ambient air pollution (excluding indoor air).*

Absolutely. This has been precised in this sentence.

*P2L14: a link to the ICAP-MME service should be provided as a web search did not allow me to reach those alternative global aerosol products.*

A link to the ICAP-MME download ftp have been added

*P3L29: why mentioning only the impact of aerosols on surface temperature as photolysis rates are also sensitive as acknowledged P5L5.*

This paragraph refers only to the impact of using interactive aerosols in the radiation scheme, while the first paragraph of page 5 describes the coupling of the aerosol and chemistry schemes. Photolysis rates depend on temperature and thus can be impacted by the use of interactive aerosols in the radiation; a sentence was added in the same paragraph.

*P4L13: what would be the implication of including stratospheric chemistry for AER-IFS given the importance of heterogeneous processes in the stratosphere?*

The computational cost would be significant, and IFS-AER deals only with tropospheric aerosols so the potential benefits of including stratospheric chemistry appear to be not so great. However, work is ongoing (outside of the scope of this manuscript) to build a full tropospheric-stratospheric aerosol chemistry system using a modal aerosol model, IFS-GLOMAP.

*P4L30: is SO2 or rather sulphate aerosol production rate (as stated P5L9) provided by IFS to AER-IFS when run in coupled mode?*

When running in coupled mode, the sulphate production rate is provided by the chemical model integrated in the IFS, CB05.

*P5L1-8: why not mentioning here the new CAMS-GLO-AP emissions that should be included in the global forecast soon (if not already)?*

Thank you for the suggestion, indeed the new CAMS_GLOB emissions are used operationally since July 2019. This was added

*P6L8-19: the scaling factor for biomass burning emissions seems very ad-hoc. How is it handled for instance by regional models? What are the perspectives to represent explicitly condensation to cope with this issue?*

Indeed, but as explained in the manuscript this is a common problem when using fire emission datasets in global model. The exact cause of the discrepancy between simulated and observed AOD when not using scaling factors for biomass burning emissions is not known yet. There are plans to better represent the ageing of biomass burning aerosols but it is far from ensures that this will bring a solution to this problem.

*P5L19: suggest adding SOA in the title of the section. Given the importance of SOA acknowledged P7L16. More information on the state of the art on SOA modelling in global aerosol models should be provided.*

Since SOA is not yet a species of its own (in CY45R1 it is treated as a component of the OM species), we prefer not to include SOA in the title so as not to confuse readers. A paragraph on SOA modelling in global models have been included.

*P9L26: typo: M86 instead of G86*

Corrected, thank you.

*P10L3: quantitative scores (RMSE, correlation and bias) should also be provided as it seems that the new formulation sometimes leads to high overestimations of low AOD levels.*

Thanks for the suggestion; global RMSE and Bias vs MODIS/Aqua have been included; the new sea-salt scheme generally improves this skill scores: sea-salt AOD was significantly underestimated with the M86 scheme.

*P13L5: the figure legend says total emission are presented whereas only super-coarse are mentioned in the text.*

Thanks; the text has been corrected.

*P21L3: where was it shown that differences where small? Is it only true locally or just*

*on average?*

It is true both locally and on average; experiments were ran (not shown) with the two approaches for dry deposition: adding it to surface fluxes, or passed through the turbulent

diffusion scheme that gave these results. Since the sentence is confusing it has been reworded.

*P24: use the same unit in the table & figure*

Corrected, thank you.

*P26L13: I find it confusing to present here ageing, while hygroscopic growth is presented in 3.6*

The content of the ageing section was moved in section 3.6

*P28L7: what is referred to as CAMS mineral dust model? AER-IFS?*

Yes, this refers to the dust scheme of IFS-AER; this has been corrected for more clarity.

*P28L15-24: the notations are not defined (SS1, SS2, etc.)*

A definition was added for the notations of the PM formulae.

*P29 Table 9: what is referred to as CAMS model? AER-IFS?*

Yes, this refers to IFS-AER; this has been corrected for more clarity.

*P29L11: confirm if the upgrade to 137 levels has been completed by now*

It has been a part of the upgrade to cycle 46R1 on 9$^{th}$ of July 2019. This has been added in the text.

*P30 Table 10: what is referred to as IFS versions? AER-IFS?*

Yes; the caption of table 10 was reworded for more clarity

*P31L9: typo "wityh"*

Corrected, thank you.

*P32L24: typo: "Unites"*

Corrected, thank you.

*P37L10: the sentence is inconsistent. Fire and dusts are mixed, and my understanding is that Europe was discussed in that paragraph.*

This paragraph and the previous one were rewritten as they were indeed confusing.

*P43L1: whereas coupling improves AOD over Europe, it is not the case for PM and should be reminded in the conclusion.*

This was mentioned in the conclusion.

**Anonymous Referee #3**

**General comments:**

*I have two substantive comments on the paper. The first is that many of the figures are not really publication quality in that they are titled/labeled with a lot of jargon that means something to the writers but not to anyone else. I noted those occasions below and suggest redoing those figures to remove the title text (or else writing it in a way that is sensible to the general reader).*

Thanks for the suggestion; these titles and labels have been removed from the plots.

*My second substantive comment is that there is a great deal of text written that does not pertain specifically to the IFS-AER operational configuration. I understand this from the point of view of explaining the possible permutations of the system, but it is hard in the end to reconstruct in my head the actual configuration used in the operational system, which seems to be the point. Table 10 is not sufficiently comprehensive in this sense. I suggest to please add a table, and maybe do it up front, that writes down the details of the IFS-AER operational configuration as a reference (chemistry or lack of, BB emissions altitude, and so forth, sea salt emission scheme, etc,.). Otherwise I was getting lost in the details.*

This is a good point, and was the subject of discussion between the authors as to exact scope of the paper (operational 45R1 only, or to present also options not used operationally). In order to clarify the operational configuration, the "operational configuration section" has been moved into a new section 3, before the detailed description of the IFS-AER components. More details have been included in this new section in order to complement table 3 (ex table 10).

**Specific comments :**

*Page 3, line 28: implication of using online ARI*

The impact is generally small on aerosol burden itself; locally and for extreme dust events, the impact on temperature and winds can create a feedback on aerosols themselves. This paragraph has been completed to mention this.

*Page 4, line 1: A point of clarification: is IFS "the" ECMWF forecasting system? And if so, isn't it run at a considerably higher than global 40 km resolution? So is IFS-AER run at the same resolution as the operational NWP system or not?*

Yes, IFS is the ECMWF forecasting system; in the CAMS project IFS with aerosol (IFS-AER) and chemistry (IFS-CB05) extensions is used. A lower resolution is used as compared to the

operational high-res NWP because of the computational cost: CB05 adds 56 prognostic variables and AER 12 or 14 depending on the configuration. In an operational context there are tight constraints on time for the model to run, which effectively limits the horizontal and vertical resolution used with IFS-AER. A sentence was added to this paragraph to explain this.

*Page 4, line 32: I don't follow the counting of species, please clarify. 12 species when running standalone I guess excludes the "optional" nitrate. When running coupled do you move the SO2 to the chemistry but still count the sulfate species? Are you now counting the nitrate? Did you not note an ammonium tracer, implied in section 2.4? Assuming all of that my count is 3 dust + 3 sea salt + 4 carbon + 1 sulfate + 2 nitrate + 1 ammonium = 14. Please clarify. And what is the condition for running the optional nitrate? Here and elsewhere in the paper it is useful to state explicitly what is in IFS-AER cycle 45R1 which is the main point of the paper. The operational configuration would seem to be IFS-AER standalone.*

Yes, when running coupled to the chemistry, SO2 is not anymore a species used in IFS-AER, but sulfate aerosols are still simulated (using oxidation rates provided by IFS-CB05). The total does come to 12 in standalone mode, and 14 when running coupled. Nitrates need the gaseous precursors provided by IFS-CB05, so can be used only when running in coupled configuration. In the new section 3 "operational configuration", it is stated that the operational configuration for cycle 45R1 is standalone, and coupled for operational cycle 46R1.

*Page 6, line 1: Any citation for the partitioning of hydrophobic and hydrophilic carbon?*

Yes, a reference was added.

*Page 6, line 18: I think the reason given here for the scaling factors is not credible. What you are really doing here is using a scaling factor to tune the emissions to give best agreement with simulated AOD, and that in fact implicates all of the model physics (especially sink processes) as well as whatever optical parameters are used to go from dry aerosol mass to hydrated extinction efficiency. So what you are doing here is finding the model "perceived emissions" needed to converge on observed AOD. If it were simply a gas to particle factor it is not clear why this would be model dependent since you are not doing that chemistry. I suggest rewriting this sentence in the spirit of the "perceived" emissions as I describe above.*

This is exactly that, as outline in Kaiser et al. (2012): tuning/scaling the emissions so as to match MODIS AOD. The exact reason why this is needed is not yet clear, and a variety of causes can participate to this: purely model factors such as optical parameters and dry/hydrated extinction efficiency as you mention, model sinks/transport etc. The different resolution between MODIS native FRP, GFAS gridded products and the model resolution certainly plays a role. The fact is that scaling factors are applied in several global atmospheric composition models. In any case, the concept of "perceived emission" certainly fits here. This paragraph was modified accordingly.

*Page 7, line 21: Figure 1 does not show anything about seasonality.*

Yes, this was an error, now corrected.

*Page 8: there is something missing (like a "x") before the 10ˆ-6 in equation 3.*

Added, thank you.

*Page 9, line 19: do you mean "two schemes"?*

Corrected, thank you.

*Page 9, line 20: "mode" instead of "model"*

This part of the sentence was deleted as it was not really useful.

*Page 9, line 21: two schemes? Why do you write "three"?*

This is indeed two schemes, corrected.

*Section 3.2: The presentation suggests there are no global tuning factors. Experience tells me this is usually not the case, and that models at least need a scaling factor as a function of spatial resolution. I'm also skeptical of the utility of comparing the predicted emissions to the G14 estimate. I won't dismiss G14's estimate of emissions, I haven't read the paper, but generally my sense is that non-inventory emissions are not usually observationally constrained sufficiently and so (like for biomass burning and dust later) some tuning is done to get emissions that in your model improve agreement with AOD, which is observed.*

No global tuning factors were used, but the G14 formula is itself the result of a fit to observations (same for the SST dependency), which might explain why no tuning was needed there.

*Figure 3: The title is best cropped from this figure as it uses jargon not discussed in the paper ("gzis", etc.)*

Absolutely; all such titles have been cropped.

*Page 12, line 8: Please explain better the dynamic nature of the lifting threshold velocity. What does "emission capacity" mean? Can you please provide an equation?*

This paragraph has been expanded and a new equation inserted to explain the simple way lifting threshold velocity is estimated in IFS-AER.

*page 12, line 12: I think you mean equations 10/11 and not 13.*

Corrected, thank you.

*Page 21, line 3: Please explain the statement about the connection to the vertical turbulent flux scheme. Are emissions just added to the model grid boxes (flux\*time = mass changed to cell)? What does it mean surface fluxes are unchanged here?*

This part has been clarified: the vertical diffusion scheme updates the surface concentration using the deposition velocity. Alternatively, before cycle 45R1 the deposition velocity can be used to update the surface fluxes. The impact of choosing either approach is very small.

*Page 23, line 31: How have these been addressed?*

Since this is part of cycle 46R1 it is a bit out of scope for this article. The median mass diameter of some aerosol species used in the ZH01 was too high and was reduced, which greatly reduced dry deposition of these species over mountainous areas.

*Page 25, line 15: Do you mean the settling velocity is horizontal invariant? Constant in space is too broad, because you do have vertical variability (don't you?)*

Not at all, the same settling velocity is used everywhere and at every model level. This applies only to super coarse dust and sea-salt, which don't usually get transported too high in the free troposphere, but still, updating this could be done in next cycles.

*Page 26, line 1: What is the origin of the numbers in Table 8 for Di?*

They come from Reddy et al. (2005). This has been specified.

*Page 26, line 10: What are the "R" values in equation 42?*

It is the assumed radius of rain drops/snow flakes, set to 1mm. This has been inserted in the text.

*Page 27, line 24: Please define OPAC and provide citation. This is first time you use the term OPAC.*

OPAC has been defined, with the appropriate citation.

*Page 31, Table 11: You mean parentheses and not hyphen at the end of the caption.*

Yes, this has been corrected, thank you.

*Page 23, figure 10: you mean top and bottom in the caption*

Corrected, thank you.

*Page 32, Section 8: Is aerosol data assimilation invoked in this analysis?*

No aerosol data assimilation was used in the simulations that have been evaluated. This has been explicated at the head of the section.

*Page 35, Figure 11: suggest again removing the title from the figure as it is inside jargon*

The title has been removed

*Page 36, figure 12: suggest again removing the title from the figure as it is inside jargon*

The title has been removed

*Page 36, line 12: Observed values of PM2.5 seem to reach 20 ug m-3, not 30.*

*Page 37, Figure 13: suggest again removing the title from the figure as it is inside jargon*

The title has been removed

*Page 37, line 5: I am confused about the text here. i don't see a persistent low bias in the model throughout the year ( red line = model higher than blue line = observations generally May - November)*

It was meant only for January and February 2017: it is true that for other months the bias is more mixed. This whole section has been rewritten using MNMB and FGE instead of bias and RMSE but for MNMB the signal is the same.

*Page 38, Figure 14: same comment about figure title*

The title has been removed

*Page 39, Figure 15: same Page 40, Figure 16: same Page 41, Figure 17: same Page 42,*

The title has been removed

*Figure 18: same*

The title has been removed

**Description and evaluation of the tropospheric aerosol scheme in the Integrated Forecasting System (IFS-AER, cycle 45R1) of ECMWF**

Samuel Rémy[1,6], Zak Kipling[2], Johannes Flemming[2], Olivier Boucher[1], Pierre Nabat[4], Martine Michou[4], Alessio Bozzo[2,5], Melanie Ades[2], Vincent Huijnen[3], Angela Benedetti[2], Richard Engelen[2], Vincent-Henri Peuch[2], and Jean-Jacques Morcrette[2]

[1]Institut Pierre-Simon Laplace, Sorbonne Université / CNRS, Paris, France
[2]European Centre for Medium Range Weather Forecasts, Reading, UK
[3]Royal Netherlands Meteorological Institute, De Bilt, Netherlands
[4]Météo-France, Toulouse, France
[5]European Organisation for the Exploitation of Meteorological Satellites, Darmstadt, DE
[6]HYGEOS, Lille, France

*Correspondence to:* Samuel Rémy (sr@hygeos.com)

**Abstract.**

This article describes the IFS-AER aerosol module used operationally in the Integrated Forecasting System (IFS) cycle 45R1, operated by the European Centre for Medium Range Weather Forecasts (ECMWF) in the framework of the Copernicus Atmospheric Monitoring Services (CAMS). We describe the different parameterizations for aerosol sources, sinks and its chemical production in IFS-AER, as well as how the aerosols are integrated in the larger atmospheric composition forecasting system. The focus is on the entire 45R1 code-base, including some components that are not used operationally, in which case this will be clearly specified. This paper is an update to the Morcrette et al. (2009) article that described aerosol forecasts at ECMWF, using the cycle 32R2 of the IFS. Between cycles 32R2 and 45R1, a number of source and sink processes have been reviewed and/or added, increasing notably the complexity of IFS-AER. A greater integration with the tropospheric chemistry scheme of the IFS has been achieved, for the sulphur cycle as well as for nitrate production. Two new species, nitrate and ammonium, have also been included in the forecasting system. Global budgets and aerosol optical depth (AOD) fields are shown, as well as an evaluation of the simulated Particulate Matter (PM) and AOD against observations, showing an increase in skill from cycle 40R2, used in the CAMS interim Reanalysis (CAMSiRA), to cycle 45R1.

**1 Introduction**

Ambient air pollution is a major public health issue, with effects ranging from increased hospital admissions to increased risk of premature death. Globally, an estimated 4.2 million deaths are estimated to be linked to outdoor air pollution in 2016 (World Health Organisation, report on Ambient air quality and health, 2018, https://www.who.int/news-room/fact-sheets/

detail/ambient-(outdoor)-air-quality-and-health, accessed on 02/04/2019), mainly from heart disease, stroke, chronic obstructive pulmonary disease, lung cancer, and acute respiratory infections in children. A large part of this mortality is caused by exposure to small particulate matter of 2.5 microns or less in diameter ($PM_{2.5}$), which are known to cause cardiovascular and respiratory disease, as well as cancers.

5  Aerosols also impact meteorological and climate processes and predictions, directly by scattering and absorbing the incoming short-wave and long-wave radiation through the aerosol-radiation interaction (ARI, Bellouin et al. (2005)) and indirectly through aerosol-cloud interaction (ACI, Fan et al. (2016) for example). Most climate models represent the impact of aerosols (Bellouin et al. (2011a)). Meteorological forecasts provided by the IFS have been shown to be improved through the

10 use of more realistic aerosol climatologies (Rodwell and Jung (2008), Bozzo et al. (2019)). Mulcahy et al. (2014) corrected a significant bias in the outgoing long-wave radiative fluxes over the Sahara by including interactive aerosol direct and indirect effects in the Met Office Unified Model (MetUM).

Particles released by volcanic eruptions can also impact air traffic, as happened in April 2010 with the eruption of the Eyjafjallajökull volcano in Iceland, which led to a major disruption of European and transatlantic aviation. As a consequence,

15 modelling and forecasting levels of particulate matter, with the highest possible level of accuracy, is a major concern of the public authorities worldwide and has focused a large effort of the research community. In this context, ECMWF has been one of the first centres to propose operational global forecasts of aerosols. Besides ECMWF, there are currently at least eight centers producing and disseminating near real-time operational global aerosol forecasting products: Japan Meteorological Agency (JMA), NOAA National Centre for Environmental Prediction (NCEP), US Navy's Fleet Numerical Meteorology and

20 Oceanography Centre (NREL/FNMOC), NASA Global Modelling and Assimilation Office (GMAO), UK Met Office, Météo-France, Barcelona Supercomputing Center (BSC) and the Finnish Meteorological Institute (FMI). These groups are all member of the International Cooperative for Aerosol Prediction (ICAP; Sessions et al., 2015; Xian et al., 2019), which uses data provided by these centers in the ICAP Multi Model Ensemble (ICAP-MME).The ICAP-MME dataset is updated daily and is available at https://www.usgodae.org/ftp/outgoing/nrl/ICAP-MME. The World Meteorological Organisation Sand and Dust

25 Storm Warning Advisory and Assessment System (SDS-WAS; Terradellas (2016)) focuses on prediction of dust aerosol and provides near-real time analysis, forecasts and evaluation at https://sds-was.aemet.es/. Ensemble prediction of aerosols is also a promising approach (Rubin et al. (2016)). Numerous regional aerosol model have been developed; a detailed enumeration and description can be found in Kukkonen et al. (2011) and Baklanov et al. (2014).

Global monitoring and forecasting of aerosols is a key objective of the Copernicus Atmospheric Monitoring Service (CAMS),

30 operated by ECMWF on behalf of the European Commission. To achieve this, ECMWF operates and develops the Integrated Forecasting System (IFS), which combines state of the art meteorological and atmospheric composition modelling together with the data assimilation of satellite products in the framework of CAMS (2014 to present) and before that of the Monitoring Atmospheric Composition and Climate series of projects (MACC, MACC-II and III, 2010 to 2014) and of the Global and regional Earth-system Monitoring using Satellite and in-situ data project (GEMS, 2005 to 2009; Hollingsworth et al., 2008).

35 The MACC and CAMS projects are centered around operational Near-Real-Time (NRT) forecasts and reanalyses of global

atmospheric composition: the MACC reanalysis (Inness et al., 2013), the CAMS interim ReAnalysis (CAMSiRA; Flemming et al., 2017), and the CAMS reanalysis (CAMSRA; Inness et al., 2019). The IFS is originally a numerical weather prediction system dedicated to operational meteorological forecasts. It was extended to forecast and assimilate aerosols (Morcrette et al., 2009; Benedetti et al., 2009), greenhouse gases (Engelen et al., 2009; Agustí-Panareda et al., 2014) and reactive trace gases (Flemming et al., 2009, 2015; Huijnen et al., 2019). "IFS-AER" denotes the IFS extended with the bin-bulk aerosol scheme used to provide aerosol products in the CAMS project.

The atmospheric composition component IFS-AER was continually updated within the MACC and CAMS project, with yearly or twice yearly upgrades of the operational forecasting system that followed and included the upgrades of the operational IFS. The code revisions that are integrated into the operational version of IFS-AER must satisfy the two conditions (one qualitative, one quantitative) that they bring the model closer to "physical" reality, i.e. that more processes and/or species are represented, and that they improve the skill scores against observations

Besides its use in the CAMS project, different versions of IFS-AER have been adapted within Météo-France's CNRM climete model system (Michou et al., 2015); it is also part of the MarcoPolo–Panda ensemble dedicated to the forecast of air quality in Eastern China (Brasseur et al., 2019), of which ECMWF is a member.

Various versions of IFS-AER have been tested and used to improve meteorological forecasts by ECMWF. This was done first by generating and implementing a 3D aerosol climatology using IFS-AER (Bozzo et al. (2019)). Interactive aerosols are also being experimented in sub-seasonal forecasts with the IFS and were shown to have a significant and positive impact on the skill of these products because of an improved representation of the radiative impacts of dust and carbonaceous aerosols in particular (Benedetti and Vitart (2018).

Morcrette et al. (2009), hereafter denoted as M09, and Benedetti et al. (2009) describe the aerosol modelling and data assimilation aspects, respectively, in the cycle 32R2 of the IFS. This paper focuses on the updates in the forward model since 2009; the data assimilation aspects and the optical properties used are only briefly described.

The paper is organized as follows. Section 2 presents a general description of IFS-AER and how it is implemented in the IFS and interacts with other components of the forecasting system. Section 3 describes the model configuration used in the operational Near-Real-Time (NRT) simulations. . Section 4 focuses on the dynamical and prescribed aerosol emissions and production processes. Section 5 details the aerosol sink processes: dry and wet deposition and sedimentation. The aerosol optical properties and PM formulae are presented in Section 6. Section 7 presents simulation results and budgets; Section 8 is dedicated to a preliminary evaluation of simulations against Aerosol Optical Depth (AOD) observations from the AERONET network (Holben et al., 1998) and against European and North-American PM observations.

**2  General description of IFS and IFS-AER**

**2.1  Atmospheric composition forecasts with the Integrated Forecasting System (IFS)**

General aspects of the IFS and how they relate to atmospheric composition modelling are described in Flemming et al. (2015); a more detailed technical and scientific documentation of the cycle 45R1 release of the IFS can be found at https:

//www.ecmwf.int/en/forecasts/documentation-and-support/evolution-ifs/cycles/summary-cycle-45r1 (last accessed on 9th of May 2019). The IFS is a Numerical Weather Prediction (NWP) model operated by the ECMWF to provide operational weather forecasts with extensions to represent tropospheric aerosols, chemically-interactive gases and greenhouse gases. This integrated atmospheric composition forecasting system forms the core of the global system of the Copernicus Atmosphere Monitoring
5    Service ; it is also used at a much higher resolution to provide operational meteorological forecasts. At the start of the time step, the three-dimensional advection of the tracer mass mixing ratios is simulated using a semi-Lagrangian (SL) method as described in Temperton et al. (2001) and Hortal (2002). Mass conservation of the transported tracers (aerosols and trace gases) can be an issue because the SL scheme is not formally mass conservative. Similarly to what is practiced for trace gases (Flemming et al., 2015; Diamantakis and Flemming, 2014) and for greenhouse gases (Agusti-Panareda et al., 2017), a
10   proportional mass fixer is used in order to ensure that the total global mass of aerosol tracers is conserved during advection.

The aerosol tracers are mixed vertically by the turbulent diffusion scheme (Beljaars and Viterbo, 1998), which also simulates the injection of emissions at the surface and the application of the surface dry deposition flux as boundary conditions. The dry deposition velocity is estimated by IFS-AER depending on the land surface and meteorological conditions as outlined in section 4. The aerosol tracers are further transported and mixed vertically by the shallow and deep convection fluxes (Bechtold et al.,
15   2014).

Since cycle 43R3, a new radiation package is in use operationally in the IFS and is described in Hogan and Bozzo (2018). The shortwave and longwave aerosol-radiation interactions (ARI) can be computed using an aerosol climatology based on the CAMS interim Reanalysis (Bozzo et al., 2019). Optionally, the prognostic aerosol mass mixing ratio from IFS-AER can be used to compute dynamically the ARI; this option is used in the operational context since cycle 45R1. The impact of using
20   prognostic aerosols in the radiation scheme is generally small on simulated aerosol fields. There can occasionally be a large impact on surface temperature and on aerosol loading itself (e.g. Rémy et al., 2015) when the aerosol loading is particularly high. The use of interactive aerosols in the radiation scheme can also indirectly impact chemical species when running coupled with CB05, since photolysis rates are usually dependent on temperature. There is currently no representation of aerosol-cloud interactions (ACI). Introducing a representation of ACI in IFS-AER is planned in the future..

25   The aerosol tracers and related processes are represented only in grid-point space. The horizontal grid can be either a reduced Gaussian grid (Hortal and Simmons, 1991) or a cubic octahedral grid. The vertical distribution uses a hybrid sigma-pressure coordinate with 60 or 137 levels. In this paper, a horizontal spectral resolution of $T_L 511$ (equivalent to a grid box size of about 40 km) and a vertical resolution of 60 levels were used, which matches the resolution used operationally with cycle 45R1. The resolution used is much coarser than for the operational IFS operated by ECMWF, which currently uses a
30   $T_{CO} 1279 L137$ resolution for its high resolution simulations, because of the numerical cost of the extra aerosol and trace gases (CB05) components. In an operational context there are tight constraints on time for the model to run, which effectively limits the horizontal and vertical resolution used with IFS-AER. .

The aerosol tracers in IFS-AER can either be initialized using the 4D-Var data assimilation of the IFS as described in Benedetti et al. (2009), or by the 3D fields from the previous forecast (in so-called "cycling forecast mode"). In the latter case,
35   the meteorological fields are provided by the ECMWF IFS operational analysis.

**2.2 Atmospheric composition in the IFS**

Tropospheric and stratospheric chemistry is represented in the IFS through the IFS-CB05-BASCOE system (Flemming et al., 2015; Huijnen et al., 2016). Tropospheric chemistry in the IFS is based on a modified version of Carbon Bond 05 (CB05; Yarwood et al., 2005), which represents 55 trace gases interacting through 93 gaseous, 3 heterogeneous and 18 photolysis reactions. IFS-CB05 is described in detail in Flemming et al. (2015). Stratospheric chemistry is based on the Belgian Assimilation System for Chemical ObsErvations (BASCOE; Errera et al., 2008), which was first developed to assimilate satellite observations of stratospheric composition. The BASCOE version as adapted in the IFS includes 58 trace gases interacting through 142 gaseous, 9 heterogeneous and 52 photolysis reactions. The merging of tropospheric and stratospheric chemistry parameterizations is described in detail in Huijnen et al. (2016). The representation of stratospheric chemistry through BASCOE is not used in the operational cycle 45R1. Alternative chemistry schemes, based on IFS-MOZART and IFS-MOCAGE, have also become available recently (Huijnen et al., 2019).

**2.3 Main characteristics of IFS-AER**

IFS-AER is a bulk-bin scheme derived from the LOA/LMDZ model (Boucher et al., 2002; Reddy et al., 2005), using mass mixing ratio as the prognostic variable of the aerosol tracers. The aerosol species and the assumed size distribution are shown in Table 1. The prognostic species are sea-salt, desert dust, Organic Matter (OM), Black Carbon (BC), sulfate and its gas-phase precursor sulphur dioxide. IFS-AER can be run in standalone mode, i.e. without any interaction with the chemistry, or coupled with IFS-CB05. Sea-salt is represented with three bins (radius bin limits at 80% relative humidity are 0.03, 0.5, 5 and 20 microns). As described in Reddy et al. (2005), sea salt emissions as well as sea salt particle radii are expressed at 80% relative humidity. This is different from all the other aerosol species in IFS-AER, which are expressed as dry mixing ratios (0% relative humidity). Users should bring special attention to this when dealing with diagnosed sea-salt aerosol mass mixing ratio, which needs to be divided by a factor of 4.3 to convert to dry mass mixing ratio, in order to account for the hygroscopic growth and change in density. Desert dust is also represented with three bins (radius bin limits are 0.03, 0.55, 0.9, and 20 microns). For both dust and sea-salt, there is no mass transfer between bins. Two components are considered of organic matter and black carbon, hydrophilic and hydrophobic fractions,  with the ageing processes transferring mass from hydrophobic to the hydrophilic OM and BC. Sulphate aerosols and, when not fully coupled to IFS-CB05, its precursor gas sulfur dioxide are represented by two prognostic variables. When running fully coupled with IFS-CB05, which is not the operational configuration with cycle 45R1, sulfur dioxide is represented in CB05 and thus not in IFS-AER. For the optional nitrate species, two prognostic variables represent fine mode nitrate, produced by gas-particle partitioning, and coarse mode nitrate, produced by heterogeneous reactions of dust and sea-salt particles. In all, IFS-AER is thus composed of 12 prognostic variables when running standalone and 14 when fully coupled with IFS-CB05 (including nitrates and ammonium) , which allows for a relatively limited consumption of computing resources, as shown in Table 2.

**Table 1.** Aerosol species and parameters of the size distribution associated to each aerosol type in IFS-AER ($r_{mod}$ =mode radius, $\rho$=particle density, $\sigma$=geometric standard deviation). Values are for the dry aerosol apart from sea salt which is given at 80% RH.

| Aerosol type | Size bin limits (sphere radius, $\mu$m) | $\rho$ (kg m$^{-3}$) | $r_{mod}$ ($\mu$m) | $\sigma$ |
|---|---|---|---|---|
| Sea Salt (80% RH) | 0.03-0.5 0.5-5.0 5.0-20 | 1183 | 0.1992,1.992 | 1.9,2.0 |
| Dust | 0.03-0.55 0.55-0.9 0.9-20 | 2610 | 0.29 | 2.0 |
| Black carbon | 0.005-0.5 | 1000 | 0.0118 | 2.0 |
| Sulfates | 0.005-20 | 1760 | 0.0355 | 2.0 |
| Organic matter | 0.005-20 | 2000 | 0.021 | 2.24 |

**Table 2.** System Billing Unit (SBU) consumption of a 24hour forecast at T$_L$511L60. SBU is a unit of CPU consumption used at ECMWF; its precise definition can be found at https://confluence.ecmwf.int/display/UDOC/HPC+accounting

| Configuration | SBU used |
|---|---|
| IFS (NWP) | 483 |
| IFS-AER (standalone) | 704 |
| IFS-AER-CB05 (coupled) | 1030 |

**2.4 Coupling to the chemistry**

IFS-AER can run coupled with the tropospheric chemistry scheme included in the IFS, CB05. The coupling is two-way and consists, on the chemistry side, in the use of aerosols in heterogeneous chemical reactions and in the computation of the photolysis rates, which is operational since cycle 43R3. On the aerosol side, the coupling is not used operationally and consists in the use of the gaseous precursors $HNO_3$ and $NH_3$ from IFS-CB05 for the production of nitrate and ammonium aerosols through gas-partitioning and heterogeneous reactions on dust and sea-salt particles, as described in Section 4. The updated concentrations of the precursors gases are passed back to IFS-CB05. Production rates of sulphate aerosols as estimated by IFS-CB05 can also be used in IFS-AER (this option is also not used operationally).

**3  Operational configuration**

IFS-AER cycle 45R1 has been operated by ECMWF to provide operational Near-Real-Time aerosol products in the framework of the Copernicus Atmospheric Monitoring Services until July 2019 when it has been upgraded to cycle 46R1. The model is run in assimilation mode, using AOD observations from MODIS collection 6 (Levy et al., 2013) and from the Polar Multi Angle Product (Popp et al., 2016). Before cycle 45R1, only MODIS AOD was assimilated. IFS-AER cycle 36R1, 40R2 and 42R1 were used in assimilation to produce the MACC reanalysis (Inness et al., 2013), the CAMS interim reanalysis (Flemming et al., 2017) and the CAMS reanalysis (Inness et al., 2019). The operational configuration and the changes brought by successive cycles are presented in https://atmosphere.copernicus.eu/node/326. A summary of the operational configurations of the latest versions of the NRT system during the CAMS and MACC projects, as well as the three reanalysis is shown in table 12.

The horizontal resolution was updated in June 2016, increasing from $T_L 255$ (approximately 80 km grid size) to $T_L 511$ (40 km). The vertical resolution has increased from 60 to 137 levels in the upgrade to cycle 46R1 on 9[th] of July 2019. Also, the CAMS reanalysis as well as the operational cycle 45R1 are run with interactive aerosols as an input of the radiative scheme to compute aerosol radiative interaction. The specific treatment of $SO_2$ emissions over outgassing volcanoes has been introduced in cycle 45R1 also. The oceanic DMS source of sulfur dioxide was implemented in cycle 37R3 in April 2013. In the cycle 45R1 operational configuration, IFS-AER was run in standalone mode, not coupled with the chemistry. In the newly operational cycle 46R1, IFS-AER is now running coupled with the chemistry. A summary of the changes brought by the new cycle 46R1, not described in this article, can be found at https://atmosphere.copernicus.eu/node/472. Also, biomassburning injection heights are not used in the operational configuration of cycle 45R1, but are now used operationally in cycle 46R1.

**4  Aerosol sources**

In IFS-AER, the sea-salt and dust emissions are computed dynamically using prognostic variables from the meteorological model. Conversion of sulfur dioxide into sulfate aerosol as well as nitrate production also uses input from the meteorological model. The other aerosol species use external emissions datasets such as the MACCity (Granier et al. (2011)), the CMIP6 (Gidden et al. (2019)) or the CAMS_GLOB datasets . Aerosol emissions are released at the surface, except for emissions from biomass burning which can optionally be released at an injection height, and $SO_2$ emissions from outgassing volcanoes which can optionally be released at the altitude of the volcano. In the operational 45R1 context, emissions from biomass burning are released at the surface while $SO_2$ emissions from outgassing volcanoes are released at the altitude of the volcano. The use of injection heights for biomass burning emissions is not used operationally in cycle 45R1 because its impact has not yet been sufficiently validated for trace gases: in an operational context it is important that biomass burning emissions of aerosols and trace gases are treated in the same way.

**Table 3.** IFS-AER cycles and options used operationally for Near-Real-Time global CAMS products. MF stands for Mass fixer, DDEP for dry deposition and SCON for sulfate conversion. G01bis is for the Ginoux et al. (2001) dust emission scheme with modified distribution of the emissions into the dust bins. R05bis is for the updated simple sulphate conversion scheme with temperature and relative humidity dependency. Cycle 46R1 also includes new developments not described in this article.

| Model Version | Date | Resolution | Sea-salt | Dust | OM | BC | SO2 | MF | DDEP | SCON |
|---|---|---|---|---|---|---|---|---|---|---|
| | | | | | Emissions | | | | | |
| CY37R3 | 04/2013 | T255L60 | M86 | G01 | EDGAR | EDGAR | EDGAR | No | R05 | R05 |
| CY40R2 | 09/2014 | T255L60 | M86 | G01 | EDGAR | EDGAR | EDGAR | No | R05 | R05 |
| CY41R1 | 09/2015 | T255L60 | M86 | G01 | EDGAR | EDGAR | EDGAR | No | R05 | R05 |
| CY41R1 | 06/2016 | T511L60 | M86 | G01 | EDGAR | EDGAR | EDGAR | No | R05 | R05 |
| CY43R1 | 01/2017 | T511L60 | M86 | G01bis | MACCity +SOA | MACCity | MACCity | Yes | R05 | R05 |
| CY43R3 | 09/2017 | T511L60 | M86 | G01bis | MACCity +SOA | MACCity | MACCity | Yes | R05+SO$_2$ | R05bis |
| CY45R1 | 06/2018 | T511L60 | G14 | G01bis | MACCity +SOA | MACCity | MACCity | Yes | ZH01+SO$_2$ | R05bis |
| CY46R1 | 07/2019 | T511L137 | G14 | G01bis | MACCity +SOA | MACCity | MACCity | Yes | ZH01+SO$_2$ | R05bis |
| MACCRA | 2013 | T255L60 | M86 | G01 | EDGAR | EDGAR | EDGAR | No | R05 | R05 |
| CAMSiRA | 2016 | T159L60 | M86 | G01 | EDGAR | EDGAR | EDGAR | No | R05 | R05 |
| CAMSRA | 2018 | T255L60 | M86 | G01bis | MACCity +SOA | MACCity | MACCity | Yes | R05 | R05bis |

**4.1 Organic Matter and Black Carbon**

The anthropogenic (non biomass-burning) sources of OM and BC can be taken from the MACCity (Granier et al., 2011) or the more recent CMIP6 (Gidden et al., 2019) emissions datasets; for the operational cycle 45R1 analysis and forecasts emissions from MACCity are used. These emissions inventories provide monthly emissions, updated from year to year for MACCity.

5 MACCity emissions of black carbon are distributed by 20% into the hydrophilic and the remaining 80% into the hydrophobic black carbon tracers as in Reddy et al. (2005). MACCity emissions provide only organic carbon emissions rather than organic matter emissions. To translate these organic carbon emissions into OM emissions a OM:OC ratio of 1.8 is used. This is in the middle range of the OM:OC ratio provided by Canagaratna et al. (2015) and Philip et al. (2014). The OM emissions are then divided evenly between hydrophilic and hydrophobic OM. Table 4 reports the average yearly global anthropogenic emissions

10 for the year 2014 from the three inventories. Biomass-burning emissions from the Global Fire Assimilation System (GFAS) are also shown. The sulfur dioxide emissions are remarkably consistent between the three two datasets. This is less the case for OM and BC.

Biomass burning sources of OM and BC are provided by GFAS (Kaiser et al., 2012), which estimates these emissions (along with those of trace gases) using active fire product from the Moderate Resolution Imaging Spectroradiometer (MODIS)

**Table 4.** Global emissions in 2014 of Organic Matter, Black Carbon and Sulfur dioxide in Tg yr$^{-1}$. Anthropogenic (non biomass burning) sources from MACCity and CMIP6 and biomass burning sources from GFAS are shown.

| Species | Anthropogenic | | Biomass-burning |
| --- | --- | --- | --- |
| | MACCity | CMIP6 | GFAS |
| Organic matter | 21.3 | 29.7 | 29.8  |
| Black carbon | 4.97 | 7.97 | 6.57 |
| Sulfur dioxide | 108.7 | 111.1 | 2.3 |

instrument onboard the Aqua and Terra satellites. Kaiser et al. (2012) compared cycling forecast simulations of biomass burning aerosols with simulations using data assimilation and concluded that a scaling factor of 3.4 should be applied to GFAS biomass-burning sources when used in the IFS so as to minimize error as compared to MODIS AOD. This means that the "perceived biomass-burning emissions" of the model needed to fit observations are estimated based on GFAS data. The same method

5 was used in Rémy et al. (2017) to derive distinct scaling factors for the OM and BC species; with scaling factors varying from 2.7 to 5 with an average of 3.2 for the former, and from 4.9 to 7 with a 6.1 average for the latter. The use of scaling factors for biomass burning emissions is frequent; for example a value of 1.7 is used in the Met Office Unified Model limited area model configuration over South America that was used for the South American Biomass Burning Analysis (SAMBBA) campaign (Kolusu et al., 2015); values of 1.8 to 4.5 are used in GEOS-5 (Colarco, 2011). Some models such as CAM5 (Tosca

10 et al., 2013) also use regional scaling factors (Lynch et al., 2016). The reasons why scaling factors are required are not fully elucidated.  In the operational cycle, the 3.4 scaling factor of Kaiser et al. (2012) is used.

  Biomass burning emissions are by default released at the surface. This can be unrealistic: a large fraction of fires release smoke constituents in the planetary boundary layer (PBL) and a minority of very large fires emit large quantities of aerosols

15 and trace gases in the free troposphere and even, for extreme cases, in the stratosphere (Freitas et al., 2005). The fraction of fires that emit aerosols and trace gases in the free troposphere was evaluated at 5–15% by various authors (Kahn et al., 2008; Val Martin et al., 2010; Sofiev et al., 2012). The GFAS dataset also includes daily injection heights that are computed using two different methods: the IS4FIRE approach (Sofiev et al., 2013) and the Plume Rise Model (PRM; Freitas et al., 2010) approach. Injection heights from GFAS as estimated using the PRM can optionally be used for biomass burning emissions. Biomass

20 burning aerosols are emitted at the mean height of maximum injection, which is defined as the average of the plume heights at which detrainment is above half the maximum value. The daily injection heights in GFAS are representative of the maximum value reached during daytime (see Rémy et al., 2017), so using these at night when the atmosphere is stable could lead to errors in the vertical distribution and transport of biomass burning aerosol plumes. To prevent this, injection heights are used only if the mean height of maximum injection is above 200 m and if the diagnosed PBL height is above 1500 m. Otherwise, the smoke

25 constituents are released in the first three model levels above surface.

Secondary Organic Aerosols (SOA) are formed from a variety of anthropogenic and biogenic gaseous and liquid precursors (Hallquist et al. (2009)). A commonly used approach to represent these processes is the Volatility Basis Set (VBS) scheme, used in GEOS-CHEM (Jo et al. (2013), Hodzic et al. (2016)) and in the ORACLE module of the EMAC model (Tsimpidi et al. (2014)).
[revised manuscript text omitted]

[Figure]

**Figure 2.** 2017 total emissions of sea-salt aerosol at 80% relative humidity from M86 (top) and G14 (bottom), in $kg\,m^{-2}\,yr^{-1}$.

ranging from 0.1 to 0.4 $kg\,m^{-2}\,yr^{-1}$. G14 stands out however more for the large increase of sea-salt production in the tropics, caused by the newly introduced dependency on sea surface temperature (SST). Production in the tropics range from 0.001 to 0.01 $kg\,m^{-2}\,yr^{-1}$ for M86 and from 0.05 to 0.2 $kg\,m^{-2}\,yr^{-1}$ for G14. It should be noted that over the Great Lakes area production of sea-salt aerosol is not zero for all schemes, which is clearly an artifact of the land-sea mask. This was corrected
5  in later cycles.

Figure 4 shows the bias in 2017 of total AOD simulated with cycle 45R1 IFS-AER using the M86 and G14 schemes as well as the observed and simulated AOD at the AERONET station of Ragged Point in the Antillas, which is one of the few stations that is mostly impacted by sea-salt. Transatlantic transport of dust emitted in the Sahara also occasionally reaches the station. The G14 scheme increases simulated AOD to values that are generally closer to AERONET observations except in May-
10  June and October/November . IFS-AER with M86 generally underestimates AOD over oceans. As compared to MODIS/Aqua Collection 6.1 AOD (Levy et al. (2013)) at 550nm, the global bias is reduced from -0.058 with M86 to -0.038 with G14. The 2017 average of daily RMSE vs MODIS AOD is slightly reduced from 0.083 to 0.08.

[Figure]

**Figure 3.** 2017 bias of simulated total AOD at 550nm against MODIS/Aqua collection 6.1 AOD; M86 (left) and G14 (right). Bottom pannel, May - Decembre 2016 daily AOD at 500 nm at the Ragged Point AERONET station, observations from L2.0 AERONET (blue points), simulated by cycling forecast only IFS-AER using the M86 scheme (violet) and by cycling forecast IFS-AER using the G14 scheme (black)

**Figure 4.** 2014 daily AOD at 500 nm at the Ragged Point AERONET station, observations from L2.0 AERONET (blue points), simulated by cycling forecast only IFS-AER using the M86 scheme (violet) and by cycling forecast IFS-AER using the G14 scheme (black)

**4.3   Dust**

The parameterization of dust emissions has been left unchanged since M09. Only the distribution of dust emissions into the three dust bins has been modified. The formulation of Ginoux et al. (2001) is used. The areas likely to produce dust are first diagnosed using a combination of masks: potential dust producing grid-cells must satisfy the following criteria:

5     – Surface albedo is under 0.52,

- The grid cell is entirely composed of land,

- The snow cover is null,

- The fraction of bare soil is above 0.1,

- There is no ice and no wet skin,

- The fraction of low vegetation is under 0.5,

- There is no high vegetation,

- The standard deviation of subgrid orography is under 50 m.

For a potential dust producing grid-cell, the total dust flux is computed by

$$F(U_{10\,\text{gust}}) = SU_{10\,\text{gust}}^2(U_{10\,\text{gust}} - U_t) \text{ if } U_{10\,\text{gust}} > U_t \tag{8}$$
$$= 0 \text{ otherwise}$$

Where $U_t$ is the lifting threshold speed, $S$ is a dust source function and $U_{10\,\text{gust}}$ are the 3 second wind gusts computed using the mean wind including gustiness effect $U_{10}$ from Eq. 1 (Bechtold and Bidlot, 2009):

$$U_{10\,\text{gust}} = U_{10} + 7.71\,u_*\left(1 + f(\frac{z}{L})\right) \tag{9}$$

Where $z$ is the PBL height, taken as 1000 m here, $u_*$ is the surface friction velocity and $L$ is the Monin-Obukhov length-scale defined as a function of surface fluxes of sensible and latent heat. This follows the parameterization of wind gusts in the IFS until cycle 33R1. The function $f$ can be expressed as:

$$f\left(\frac{z}{L}\right) = 1 + \left(\frac{0.5}{12}\frac{z}{L}\right)^{1/3} \tag{10}$$

Estimating the lifting threshold speed is a key part of any dust emission scheme; it depends on soil wetness, soil roughness on dust characteristics and mineralogy, and the size of the dust particles that are being lifted. In IFS-AER, a simple approach is used to estimate $U_t$:

$$U_t = U_{t0}\,D_{p0}^{0.25}\,(1.2 + log(w)) \tag{11}$$

Where $U_{t0}$ and $D_{p0}$ are "climatological" lifting threshold speed and dust particle radius at emission; the former varying between 3.5 m/s over the Taklimakan to 6 m/s over the Sahara and the latter being set constant at 5 μm. w is the prognostic surface volumetric soil moisture. . The lifting threshold speed is similar for each dust bin and is shown in Figure 5. Values are highest over the Sahara, above 5 $\text{m s}^{-1}$, while areas of very low values (0.1–1 $\text{m s}^{-1}$) can be found in some boreal regions. The relatively low values over the Taklimakan and Gobi deserts can explain the high dust emissions over these regions.

[Figure]

**Figure 5.** 2017 lifting threshold speed in $\mathrm{m\,s^{-1}}$.

The dust source function $S$ is proportional to surface albedo. Equation 8 provides an estimate of the total emitted dust flux, which has to be distributed into the three dust bins. Until cycle 43R1, the distribution was 8% of emissions into fine dust, 31% into coarse dust and 61% into super-coarse dust. Comparing these values to the observed size distribution of dust aerosols at emission provided by Kok (2011) showed that the relative fraction of super-coarse particles was too low, and the relative

5    fraction of fine particles too high. In the CAMS reanalysis (Inness et al., 2019) and in the operational cycles 43R1 onward, the distribution of total emissions into the dust bins was revised as follows: 5% into fine dust, 12% into coarse dust and 83% into super-coarse dust. Even though the total emissions are left unchanged, this change of distribution led to a significant decrease in the simulated burden and AOD of dust aerosols because the lifetime of super-coarse dust is shorter than for the other two bins as it is subject to a large sedimentation rate.

10    Figure 6 shows the 2017 emissions of total dust, ie the sum of the three bins. The highest emissions, at 0.2–0.3 $\mathrm{kg\,m^{-1}\,yr^{-1}}$ occur in the Gobi and Taklimakan deserts, which were impacted by severe dust storms in particular in May 2017. The Sahara, the Arabian Peninsula and parts of Iran and of Turkestan are also prominent. The emissions are very widespread in these regions, which is probably not realistic. Maps of  frequency of occurrence of  dust AOD as retrieved using MODIS deep blue information (see Ginoux et al. (2012) for more details on the method) being above different

15    thresholds were computed using data kindly provided by Paul Ginoux (personal communication), and can serve as dust source functions. These show much higher maxima and lower minima in the Sahara and Arabian peninsula, which confirms that the current operational approach could be refined.

[Figure]

**Figure 6.** 2017 total emissions of dust aerosol in $\mathrm{kg\,m^{-2}\,yr^{-1}}$.

**4.4 Sulfur dioxide and sulfate**

[revised manuscript text omitted]

$$S_r = \frac{1}{3u_*(E_{\mathrm{B}} + E_{\mathrm{IM}} + E_{\mathrm{IN}})} \tag{29}$$

where $E_{\mathrm{B}}$, $E_{\mathrm{IM}}$ and $E_{\mathrm{IN}}$ are the collection efficiencies for Brownian diffusion, impaction and interception, respectively.

5 $$E_{\mathrm{B}} = Sc^{-Y_{\mathrm{R}}} \tag{30}$$

where $Sc$ is the particle Schmidt number computed by $\frac{\nu}{D}$ where $\nu$ is the kinematic viscosity of air and $D$ is the particle diffusion coefficient, $Y_{\mathrm{R}}$ is a surface-dependent constant with values provided in Table 3 of Zhang et al. (2001).

$$E_{\mathrm{IM}} = \left(\frac{St}{\alpha + St}\right)^2 \tag{31}$$

where $St$ is the Stokes number for smooth and rough flow regime:

10 $$St = V_{\mathrm{g}}\frac{u_*^2}{D_{\mathrm{visc}}} \quad \text{smooth surface:} z_0 < 1mm, \tag{32}$$

$$St = V_{\mathrm{g}}\frac{u_*}{(gC_{\mathrm{R}})} \quad \text{rough surface:} z_0 > 1mm \tag{33}$$

where $D_{visc}$ is the dynamic viscosity of air, computed as a function of temperature only, and $V_{\mathrm{g}}$ is the gravitational velocity computed as

$$V_{\mathrm{g}} = 2\rho\frac{D_{\mathrm{p}}^2 gC_{\mathrm{F}}}{(18D_{\mathrm{visc}})} \tag{34}$$

$C_F$ is the Cunningham slip correction to account for the viscosity dependency on air pressure and temperature. $\rho$ and $D_p$ are the particle density and diameter respectively. For $D_p$, the mass median diameter (MMD) of each aerosol prognostic variable is used, and hygroscopic growth is accounted for the relevant species. The Cunningham slip correction is defined differently from the original Zhang et al. (2001) implementation:

$$C_F = \exp(16\,\sigma) + 1.246 \exp(3.5 \ln(2\sigma)) \times 2\frac{\lambda}{D_p} \tag{35}$$

[revised manuscript text omitted]

where $D_i$ is the fraction of aerosol $i$ that is included in cloud droplets and $f_k$ is the cloud fraction at level $k$. The value of the parameter $D_i$ is from Reddy et al. (2005); it is indicated in Table 10. Following Giorgi and Chameides (1986), $\beta_k$ is the rate of conversion of cloud water to rain water, computed by comparing the precipitation flux at levels $k$ and $k+1$ is written as follows:

$$\beta_k = \frac{P_{k+1} - P_k}{\rho_k \, \Delta z_k \, f_k \, q_k} \tag{41}$$

where $P_k$ is the sum of rain and snow precipitation fluxes at level $k$, $q_k$ the sum of the liquid and ice mass mixing ratio and $\Delta z_k$ is the layer thickness at level $k$. This means that, as in M09, no distinction is made between rain and snow.

**5.3.2 Below-cloud wet deposition (washout)**

The below cloud scavenging rate at model $k$ of an aerosol $i$ is given by:

$$W_{i,k}^{B} = \frac{3}{4}\left( \frac{P_{k\mathrm{l}}\,\alpha_{\mathrm{l}}}{R_{\mathrm{l}}\,\rho_{\mathrm{l}}} + \frac{P_{k\mathrm{i}}\,\alpha_{\mathrm{i}}}{R_{\mathrm{i}}\,\rho_{\mathrm{i}}} \right) \tag{42}$$

[revised manuscript text omitted]

**8 Evaluation**

In this section, a short evaluation of the simulated AOD against observations from the Aerosol Robotic Network (AERONET; Holben et al., 1998) is shown, and of PM$_{2.5}$ and PM$_{10}$ against observations from the Airnow and Airbase networks in the UnitedStates and Europe, respectively. The simulations evaluated here consist of 24h cycling forecasts with meteorolog-

5   ical initial conditions provided by an analysis, and aerosol and chemical initial conditions provided by the previous forecast. No aerosol data assimilation was used in the simulations that have been evaluated in this section.An evaluation of such a simulation with cycle 45R1 using the same configuration and resolution (T$_L$511L60) as the operational forecasts is presented first. A comparison of the skill scores between standalone and coupled with IFS-CB05 simulations with cycle 45R1 is then made. Finally, the skill scores of simulations with cycle 40R2 and 45R1 are compared, with simulations using similar resolution

10  (T$_L$159L60) and emissions. Cycle 40R2 was chosen because it was used in the CAMS interim Reanalysis and because the changes between cycle 32R2, described in Morcrette et al. (2009), and cycle 40R2, are limited as far as aerosols are concerned. Because of upgrades in the ECMWF high performance computing facility, it is not possible anymore to run simulations of the

original Cycle 32R2. This is not intended as a full evaluation, which would require a much more thorough validation of the output of IFS-AER, but rather to show that the model performs relatively well for the headline CAMS products.

**8.1 Summary**

Table 14 shows a summary of global and regional skill scores for AOD at 500 nm and PM for a year of simulation for the four experiments described above. The Modified Normalized Mean Bias (MNMB) and Fractional Gross Error (FGE) are shown, so that the skill of the model in simulating relatively low AOD values is not overlooked. MNMB varies between -2 and B and is defined by the following equation, for a population of N forecasts $f_i$ and observations $o_i$:

$$MNMB = \frac{2}{N} \sum_i \frac{f_i - o_i}{f_i + o_i} \tag{43}$$

FGE varies between 0 (best) and 2 (worst), and is defined as:

$$FGE = \frac{2}{N} \sum_i \left| \frac{f_i - o_i}{f_i + o_i} \right| \tag{44}$$

For all regions and for AOD at 500 nm, $PM_{2.5}$ and $PM_{10}$, the FGE is improved by cycle 45R1 as compared to 40R2 at a similar resolution, sometimes by a large margin: global FGE on AOD at 500 nm is decreased by more than 10%. The only exception is over Europe where mean FGE for AOD is nearly similar between the two cycle. Bias, as measured by MNMB, is improved nearly everywhere except over Europe and Africa for AOD and North America for $PM_{2.5}$. Interestingly, the 45R1 simulation using the operational resolution of $T_L511L60$ shows an improved MNMB for AOD and more markedly for $PM_{2.5}$ as compared to the 45R1 simulation at $T_L159L60$. Simulation with cycle 45R1 coupled with IFS-CB05 shows a small improvement as compared to standalone CY45R1 at a global scale. However, regional AOD scores are notably improved with the coupled simulation, especially over Europe, where FGE is reduced from 0.53 to 0.38, and where the negative bias is nearly eliminated.

**8.2 Evaluation against AERONET**

Figures 12 gives an indication of how the model compared to AERONET observations for daily AOD forecasts, globally and over Europe, in its standalone and coupled to the chemistry (including nitrates) configurations . When coupled to the chemistry, sulfate oxidation rates are provided by IFS-CB05, and are generally lower than the rates computed by the simple scheme of IFS-AER. Lower sulfate burden and surface concentration are compensated by contributions from nitrate and ammonium. The global MNMB is generally negative; slightly less so for the coupled version, with values usually between -0.1 and -0.2. Over Europe, MNMB with the standalone cycle 45R1 is more negative, from -0.2 to -0.6; the improvement is significant with the coupled configuration, with MNMB only slightly negative in general. Global FGE is comprised between 0.5 and 0.7 generally; slightly less with the coupled configuration. European FGE is much improved by the coupled configuration, decreasing from 0.5-0.7 on average to 0.3-0.5. The Root Mean Square Error (RMSE) on weekly AOD is in the range of 0.1 to 0.2 on average, while the bias is generally lower than 0.05. The global correlation factor between simulated and observed instantaneous AOD

**Table 14.** Average over the 1/5/2016 to 1/5/2017 period of the Modified Normalized Mean Bias (MNMB)/Fractional Gross Error (FGE) of daily AOD at 500nm and PM from the experiments described in this section. AOD observations are from AERONET level 2; European PM observations are from 65 $PM_{2.5}$ and 138 $PM_{10}$ background rural Airbase stations; North American PM observations are from 1006 $PM_{2.5}$ stations and 336 $PM_{10}$ stations.

| Experiment | Global | Europe | N.America | S.America | Africa | SE Asia |
|---|---|---|---|---|---|---|
| AOD 40R2 ($T_L159L60$) | −0.24 / 0.66 | −0.16 / 0.51 | −0.23 / 0.54 | −0.46 / 0.68 | 0.027 / 0.53 | −0.43 / 0.71 |
| AOD 45R1 ($T_L159L60$) | −0.12 / 0.59 | −0.28 / 0.52 | −0.11 / 0.52 | −0.28 / 0.56 | 0.12 / 0.45 | −0.11 / 0.48 |
| AOD 45R1 ($T_L511L60$) | −0.16 / 0.60 | −0.34 / 0.53 | −0.08 / 0.50 | −0.29 / 0.57 | 0.12 / 0.46 | −0.058 / 0.27 |
| AOD 45R1 coupled ($T_L511L60$) | −0.13 / 0.56 | −0.03 / 0.38 | −0.02 / 0.47 | −0.30 / 0.53 | 0.13 / 0.43 | −0.081 / 0.27 |
| $PM_{2.5}$ 40R2 ($T_L159L60$) | — | 0.17 / 0.75 | −0.06 / 0.67 | — | — | — |
| $PM_{2.5}$ 45R1 ($T_L159L60$) | — | 0.22 / 0.51 | 0.32 / 0.59 | — | — | — |
| $PM_{2.5}$ 45R1 ($T_L511L60$) | — | 0.08 / 0.49 | 0.31 / 0.59 | — | — | — |
| $PM_{2.5}$ 45R1 coupled ($T_L511L60$) | — | 0.27 / 0.51 | 0.34 / 0.59 | — | — | — |
| $PM_{10}$ 40R2 ($T_L159L60$) | — | 0.12 / 0.75 | 0.6 / 0.90 | — | — | — |
| $PM_{10}$ 45R1 ($T_L159L60$) | — | 0.05 / 0.44 | −0.23 / 0.58 | — | — | — |
| $PM_{10}$ CY45R1 ($T_L511L60$) | — | −0.09 / 0.46 | −0.18 / 0.60 | — | — | — |
| $PM_{10}$ 45R1 coupled ($T_L511L60$) | — | 0.12 / 0.44 | −0.17 / 0.58 | — | — | — |

**8.3 Evaluation against PM observations**

In this section, simulations of near-surface $PM_{2.5}$ and $PM_{10}$ are evaluated against observations from two regional networks: the Airnow network which gathers observations mostly over the United States and Canada, and the AirBase, the European Air Quality Database operated by the European Environment Agency (EEA), which gathers observations over Europe. As metadata was not available for Airnow observations, data from all sites (1036 for $PM_{2.5}$ and 354 for $PM_{10}$ in 2017) are used, which means that urban, suburban and rural stations are included in the scores. Similarly, the observations may be representative of background pollution or closer to traffic sources. For the Airbase observations, for which site information was available, it was chosen to focus on scores vs background rural stations (62 stations in all). Because of the relatively coarse horizontal resolution of the simulation (about 40 km), urban areas cannot be realistically represented and this will lead to a low bias as compared to observations, caused not from issues in the model or in emissions, but mostly from resolution. Figures 13 and 14 show the result of the evaluation against the Airnow and the AirBase observations respectively, for the standalone and the coupled configurations. .

[Figure]

**Figure 12.** Global (left) and European (right) Modified Normalized Mean Bias (MNMB, top) and Fractional Gross Error (FGE, bottom) of daily AOD at 500 nm simulated by standalone (black) and coupled (violet) IFS-AER cycle 45R1 against observations from AERONET.

Observed $PM_{2.5}$ at all Airnow stations show higher values during the summer months, reaching 30 µg m$^{-3}$ in early September. These high values ware generally caused by fire events. IFS-AER generally overestimates $PM_{2.5}$ significantly, with a MNMB between 0.2 and 0.6 except during the winter of 2016/2017. This overestimation is mainly caused by the SOA emissions at surface, which contribute to a large fraction of $PM_{2.5}$, and to the fact that emissions from biomass-burning are released

5    at the surface, which can increase $PM_{2.5}$ forecasts to very high values during fire events. The FGE against $PM_{2.5}$ observations over North America stands at 0.5-0.7, with relatively little variations.Simulated $PM_{1}0$ is generally biased low, in contract to $PM_{2.5}$, and the FGE is slightly higher also, with values of 0.5-0.8. Both MNMB and FGE are relatively similar for the standalone and coupled configurations.

As compared to AirBase observations (Figure 14) , $PM_{2.5}$ and $PM_{10}$ show a negative MNMB during most of January and

10    February 2017 by -0.4 to -0.8 for the standalone configuration . For the rest of the year, MNMB is positive for $PM_{2.5}$ but with value hovering around 0.2 for standalone and 0.4 when running coupled with the chemistry The MNMB of PM10 simulated with the standalone configuration is slightly negative from May to September 2016, with values between 0 and -0.2, positive during autumn, and very negative during January-Februay 2017, possibly caused by the underestimation of surface concentration of anthropogenic aerosols during the

[Figure]

**Figure 13.** MNMB (top) and FGE (bottom) of daily near-surface PM$_{2.5}$ (left) and PM$_{10}$ (right) in $\mu g\,m^{-3}$ simulated by standalone (black) and coupled (violet) cycle 45R1 against North American observations from the Airnow network.

mostly anticyclonic conditions of this period. The MNMB of PM10 is lifted upwards by 0.3 on average, and becomes generally positive when running coupled with the chemistry.

The FGE of simulated PM$_{2.5}$ and PM$_{10}$ are quite close and is generally comprised between 0.4 and 0.7, with higher values during parts of January and February 2017 probably associated with pollution events. Interestingly, the FGE is generally higher when coupled with the chemistry, but lower during the January-February 2017 spikes, showing that these pollution events are better simulated when the coupling with the chemistry is used.

The error is on average significantly larger for PM$_{10}$ (between 5 and 15 $\mu g\,m^{-3}$, excluding larger spikes) than for PM$_{2.5}$ (generally below 5 $\mu g\,m^{-3}$, except for spikes). Biomass-burning events are associated with spikes in RMS, which otherwise is generally be Similarly to results against Airnow observations, European PM$_{10}$ is generally underestimated by IFS-AER, with a RMS significantly higher than for PM$_{2.5}$, at 5-10 $\mu g\,m^{-3}$ generally (higher in January-February 2017) against less than 5 $\mu g\,m^{-3}$ for PM$_{2.5}$.

In this section, the AOD and European PM forecasts of cycle 45R1 IFS-AER in its standalone and coupled to the chemistry configurations are compared. When coupled to the chemistry, sulfate oxidation rates are provided by IFS-CB05, and are generally lower than the rates computed by the simple scheme of IFS-AER. Lower sulfate burden is compensated by contribution

[Figure]

**Figure 14.** MNMB (top) and FGE (bottom) of daily near-surface PM$_{2.5}$ (left) and PM$_{10}$ (right) in $\mu$g m$^{-3}$ simulated by standalone (black) and coupled (violet) cycle 45R1 against European observations from the Airbase network.

from nitrate and ammonium. The simulations were carried out at the resolution used operationally: T$_L$511 with 60 levels. It should be noted that the results presented here are only valid for cycle 45R1: since IFS-CB05 is constantly evolving, both the sulfur dioxide oxidation rates and the concentration of the nitric acid and ammonia precursor gases of nitrate and ammonium can vary from cycle to cycle. A selection of skill scores against observations of Global and European AOD are shown in Figure

5   ?? and against European observations of PM$_{2.5}$ and PM$_{10}$ in Figure ??. The coupled IFS-AER show a slightly degraded global bias, as the lower sulfate is not entirely compensated by nitrate and ammonium. However, the global error is on average lower with the coupled system. Over Europe, the impact of the coupling with IFS-CB05 is much more positive, with a clear decrease in both bias (by up to 0.05) and error (by up to 25%).

The impact on PM forecasts is however generally negative over Europe. The coupling with IFS-CB05 appear to have mostly

10   negative impact on PM$_{2.5}$, with an increase by 2–4 $\mu$g m$^{-3}$ of the positive bias, and a small increase of RMSE. The difference is slightly more important for PM$_{10}$, which is simulated 3–5 $\mu$g m$^{-3}$ higher when running coupled with IFS-CB05. This slightly larger difference could be caused by coarse mode nitrate formed on sea-salt particles, which is taken into account in PM$_{10}$ but contributes only 25% of its mass to PM$_{2.5}$. RMSE of simulated PM$_{10}$ is very close for the two simulations, slightly degraded for the coupled run.

**8.4 Comparison with IFS-AER cycle 40R2**

This section aims to give an initial evaluation of the impact of the model upgrades between cycles 40R2 and 45R1. To achieve this, the control run (i.e., without data assimilation) of the CAMS interim reanalysis Flemming et al. (2017) is used, and a simulation with cycle 45R1 in standalone mode, using the same fixed emissions (except for the new SOA emissions) and the same horizontal resolution ($T_L159$) was carried out. The latter simulation covers the May–December 2016 period. Skill scores of daily AOD vs AERONET observations, and of daily near-surface $PM_{2.5}$ and $PM_{10}$ vs observations from the AirBase and Airnow networks were then computed and are shown in Figures 15, 16 and 17. As compared to cycle 40R2, 45R1 brings an important improvement in the skill of AOD forecasts, as measured by MNMB, FGE and correlation. MNMB is generally above -0.1 for 45R1. The global FGE is significantly reduced and RMSE (not shown) is at times more than halved by 45R1 as compared to 40R2. The spatial correlation is also generally higher with cycle 45R1 as compared to 40R2. The development that had the most impact on skill scores against AERONET is the implementation of a new SOA source in cycle 43R1, which led to a significant improvement of both bias and RMSE. Using the ZH01 dry deposition scheme also improved scores. The new G14 sea-salt aerosol scheme had little impact on AOD scores against AERONET or on PM skill scores, but improved notable the bias and RMSE versus MODIS AOD

Skill scores of the global PM simulations are also generally improved, except for $PM_{2.5}$ over North America, for which the MNMB as well as the FGE during summer is degraded by cycle 45R1 as compared to 40R2. The correlation is nonetheless improved. This degradation is caused by the new source of SOA, which is associated with high PM values over North America and an unrealistic diurnal cycle. $PM_{10}$ is generally underestimated over North America, but less so with 45R1. This is associated with a lower error and a much higher correlation of simulations with observations. PM forecasts over Europe are significantly improved by cycle 45R1 as compared to 40R2. The improvement is especially marked for $PM_{10}$, for which the RMSE (not shown) is divided by more than three.

**9 Conclusions**

IFS-AER is a simple and low-cost scheme that aims to represent the major atmospheric aerosol species. Because of this simplicity, many processes such as internal mixing, coagulation and nucleation are not explicitly represented. Despite this, IFS-AER achieves a reasonable skill in forecasting AOD and PM. The improvement in skill for the headline products of the operational CAMS system, AOD and PM, were improved considerably (except for $PM_{2.5}$ over North America) from cycle 40R2 to 45R1. The tendency has been towards a system that is more integrated with the other components of the IFS and particularly atmospheric chemistry, with the possibility to run with integrated sulfur and ammonia cycles. The coupling with IFS-CB05 generally improves the  error of the forecasts of IFS-AER,  except for European PM simulations. This also makes the skill of IFS-AER more dependent on the evolution of IFS-CB05.

As compared to the original implementation as described in Morcrette et al. (2009), many components of IFS-AER have been reviewed, and two new species (nitrate and ammonium) were added. Some parameterizations on the other hand such as dust production and wet scavenging are essentially similar in cycle 45R1 as in M09.

[Figure]

**Figure 15.** MNMB (left), FGE (right) and spatial correlation (bottom) of daily AOD at 500-550 nm simulated by cycles 40R2 (green) and 45R1 (red) against global observations from AERONET.

  Future upgrades will aim to address some of the shortcomings noted in the model evaluation. The currently operational cycle 46R1, for which a description of the changes can be found at https://atmosphere.copernicus.eu/node/472, includes an upgrade of the dust emission and of the scavenging schemes as well

5    as the use of biomass burning injection heights for aerosols and trace gases fire emissions. IFS-AER is running in cycle 46R1 coupled with the chemistry, with the new nitrate and ammonium species. The treatment of secondary organics is also very simplistic and lead to unrealistic diurnal cycle in PM simulations regionally. In the longer term, the possibility to use SOA production rates provided by the chemistry scheme will be given.

*Code availability.*

10    Model codes developed at ECMWF are the intellectual property of ECMWF and its member states, and therefore the IFS code is not publicly available. ECMWF member-state weather services and their approved partners will get access granted.

[Figure]

**Figure 16.** MNMB (top), FGE (middle) and spatial correlation (bottom) of daily near-surface PM$_{2.5}$ (left) and PM$_{10}$ (right) in $\mu$g m$^{-3}$ simulated by cycles 40R2 (green) and 45R1 (red) against observations in the United States from the Airnow network.

[revised manuscript text omitted]